# Species-specific metabolic reprogramming in human and mouse microglia during inflammatory pathway induction

Angélica María Sabogal-Guáqueta[1,8], Alejandro Marmolejo-Garza[1,2,8], Marina Trombetta-Lima[1,2], Asmaa Oun [1], Jasmijn Hunneman[1], Tingting Chen[1], Jari Koistinaho[3,4], Sarka Lehtonen [3], Arjan Kortholt [5,6], Justina C. Wolters [7], Barbara M. Bakker [7], Bart J. L. Eggen [2], Erik Boddeke [2] & Amalia Dolga [1] ✉

Metabolic reprogramming is a hallmark of the immune cells in response to inflammatory stimuli. This metabolic process involves a switch from oxidative phosphorylation (OXPHOS) to glycolysis or alterations in other metabolic pathways. However, most of the experimental findings have been acquired in murine immune cells, and little is known about the metabolic reprogramming of human microglia. In this study, we investigate the transcriptomic, proteomic, and metabolic profiles of mouse and iPSC-derived human microglia challenged with the TLR4 agonist LPS. We demonstrate that both species display a metabolic shift and an overall increased glycolytic gene signature in response to LPS treatment. The metabolic reprogramming is characterized by the upregulation of hexokinases in mouse microglia and phosphofructokinases in human microglia. This study provides a direct comparison of metabolism between mouse and human microglia, highlighting the species-specific pathways involved in immunometabolism and the importance of considering these differences in translational research.

Microglia are the resident innate immune cells of the central nervous system (CNS) and are involved in the immune response to pathogens or alteration to the CNS microenvironment. Microglia are also required for neurodevelopment, neuroplasticity, and tissue repair[1–3]. Dysregulation of microglial function has been well established in pathologies linked to neurodegeneration, including Alzheimer's disease (AD)[4–6], Parkinson's disease[7], multiple sclerosis,[8,9] and Huntington's disease[10]. The lack of disease-modifying therapies for these conditions demonstrates the need to better delineate mechanisms that govern microglial function and to study the modulation of key mediators of such mechanisms for potential therapies.

Microglia acquire an inflammatory phenotype comprising production of pro-inflammatory cytokines, such as interleukin-1β (IL-1β) and tumor necrosis factor-α (TNF-α) in response to pathogenic stimuli. Besides the inflammatory phenotype, microglia can adopt several phenotypes to clear pathogenic substances, eliminate cellular/synaptic debris, clear metabolic waste, regulate synaptic pruning, neuronal maturation, and support neuro-regeneration. During

[1]Department of Molecular Pharmacology, Faculty of Science and Engineering, Groningen Research Institute of Pharmacy, Behavioral and Cognitive Neurosciences (BCN), University of Groningen, Groningen, The Netherlands. [2]Department of Biomedical Sciences of Cells & Systems, section Molecular Neurobiology, Faculty of Medical Sciences, University of Groningen, University Medical Center Groningen, Groningen, The Netherlands. [3]A.I. Virtanen Institute for Molecular Sciences, University of Eastern Finland, P.O. Box 1627, 70211 Kuopio, Finland. [4]Neuroscience Center, Helsinki Institute for Life Science, University of Helsinki, Haartmaninkatu 8, 00290 Helsinki, Finland. [5]Department of Cell Biochemistry, University of Groningen, Groningen, The Netherlands. [6]YETEM-Innovative Technologies Application and Research Centre Suleyman Demirel University, Isparta, Turkey. [7]Laboratory of Pediatrics, Section Systems Medicine of Metabolism and Signaling, Faculty of Medical Sciences, University of Groningen, University Medical Center Groningen, Groningen, The Netherlands. [8]These authors contributed equally: Angélica María Sabogal-Guáqueta, Alejandro Marmolejo-Garza. ✉e-mail: a.m.dolga@rug.nl

phenotypic transition, metabolic reprogramming is considered a hallmark of inflammatory murine macrophages/microglia[11,12]. Metabolic reprogramming or a metabolic switch encompasses a variety of cellular alterations in bioenergetic pathways to adapt to the cellular metabolic needs. The changes in metabolic pathways include several processes, such as oxidative phosphorylation (OXPHOS), tricyclic acid cycle (TCA cycle), glycolysis, the pentose phosphate pathway, amino acid metabolism, and fatty acid oxidation. Recent studies have shown that mouse primary microglia are able to switch their cell metabolism from mainly mitochondrial OXPHOS to glycolysis[12] in response to pro-inflammatory stimuli, such as lipopolysaccharides (LPS), and Amyloid-β (Aβ)[13] in acute and chronic manners, pointing at a role of metabolism in trained innate immunity in microglia. Similarly, this phenomenon of metabolic switch to glycolysis is also present in mouse macrophages, dendritic cells, NK cells, B cells and effector T cells[14,15].

Under physiological conditions, immune cells, including microglia and macrophages, primarily rely on oxidative phosphorylation[16]. Under inflammatory conditions, an accumulation of citrate and succinate has been reported in macrophages, which contributes to an increase in reactive oxygen species (ROS) and nitric oxide production, leading to a switch from anti- to pro-inflammatory phenotype in the immune cells[14,17]. These findings were followed by an exponential surge of interest in reprogramming metabolic pathways and the term of "immunometabolism" was coined in 2011 by Mathis & Schoelson[18]. However, the vast majority of these experimental studies have been performed in murine immune cells and not much information is available on metabolic reprogramming in human immune cells, specifically innate brain immune cells, microglia. For instance, species-specific differences in LPS-treated macrophages have been reported in mouse and human systems[11] indicating that murine findings shall be interpreted with caution and highlighting the need to establish more adequate human model systems.

Studies of human brain microglia have been performed on isolated microglia from fresh post-mortem samples from potentially neuropathologically affected individuals, which might be hindered by a high interindividual variation. Alternatively, robust differentiation protocols of iPSC-derived human microglia could provide a possibility to study metabolic profiles. First studies on iPSC-derived microglia were documented in 2016 and mainly focused on the characterization of the microglial phenotype in terms of differentiation and maturation. Hence, the goal of our study was to investigate the metabolic reprogramming in human iPSC-derived microglia compared with mouse microglial in vitro and in vivo models in response to the prototypical stimulus LPS.

In this study, we show dysregulation of metabolic pathways concomitant with upregulation of inflammatory pathways in both in vivo and in vitro treated mouse and human iPSC-derived microglia supported by multi-omics analyses. Functional measurements demonstrated glycolytic upregulation in both species but discrepant changes in oxidative metabolism. This report provides evidence on species-specific differences in metabolic reprogramming in microglia.

## Results

### LPS-stimulation of mouse microglia induces an inflammatory gene signature

LPS is commonly used to investigate the stimulation of immune cells and their response to toll-like receptor 4 (TLR4) activation, as a model to mimic inflammatory processes. To study how microglia respond to LPS at the transcriptome level, we performed RNA-seq of LPS-treated primary mouse microglia (Fig. 1A). We analyzed the RNA-seq data following LPS challenge for a short period (4 h) and a longer period (24 h) of mouse microglia to assess acute and late responses to LPS stimulus. We used LPS 250 ng/ml based on previous studies[19,20] and

data showed in Supplementary Fig. 1A−C. The analysis of these data revealed sets of LPS-induced genes (Supplementary Dataset 1-4) and robust transcriptomic changes in mouse microglia (Fig. 1B). LPS-treated microglia, regardless of the treatment duration segregated from control microglia in the first principal component. Transcriptomic differences between LPS treatments for 4 h and 24 h segregated microglia in the second component (Fig. 1B).

Treatment with LPS for short time (4 h) induced a transcriptomic response mainly characterized by the upregulation of inflammation-related genes, such as IL1/chemokine signaling pathways (*Il-1a, Nfkbia, Ccl4, Ccl5*), ribonuclease activity (*Oasl2, Oasl1*), or immune cell response (*Cd69, Gbp3, Isg15, Igtp, Mtmr14, H2-M2*) among others. Short-term LPS treatment led to the downregulation of several genes related to immune cell differentiation, (such as *Inpp5d, Lhfpl2, Cebpa, Rab7b, Nfic, Kif21b, IL16ra, Tmem86a, Tmem104, Fblim1, S1pr1*) (Fig. 1C, Supplementary Dataset 1) when compared to control cells. Treatment with LPS for longer time (24 h) induced the expression of inflammatory genes *Il-1a, Nlrp3*, interferon-induced genes *Gbp3, Cd69, Oasl1*, immune activation genes *H2-M2, Ddx58, Socs3, Vnn3, Igtp*, and the mitochondrially-localized *Sod2*, and decreased the expression of microglial markers *Pla2g15, Cx3cr1, Cd300lb*, and of genes related to Alzheimer's Disease, such as *Pald1, Igf1, Plau, Flt1*, and the immune cell differentiation gene *Cebpa* (Fig. 1D, Supplementary Dataset 2) when compared to the control cells. Hierarchical clustering of the top 50 (Fig. 1E) and 100 (Supplementary Fig. 1D) most changed genes following 4 h of LPS treatment showcased a module of upregulated genes, including many genes involved in the immune response, such as *Irf1, Mtmr14, Ifih1, Nfkbie, and Tnfaip2*. This cluster was downregulated following 24 h compared to 4 h LPS treatment. Another cluster of genes involved in the immune response, including *H2-M2, IL1a, Sod2, Ccl5 and Socs3* was upregulated at a later stage after 24 h LPS challenge. Differential expression analysis between LPS 24 h and 4 h showcases downregulation of *Irf1, Mtmr14, Il15* among others (Supplementary Fig. 1E and Supplementary Dataset 4), demonstrating the temporal regulation of LPS responses.

Gene set enrichment analysis (GSEA) of statistically significant dysregulated genes following LPS application for 4 and 24 h demonstrated shared enrichment of inflammatory pathways, such as NF-κB signaling pathway, inflammatory response, and TNF signaling pathway (Fig. 1F). We next applied Molecular Complex Detection (MCODE) algorithm on the Protein-Protein interaction (PPi) analysis that resulted from the input of DEGs of LPS-treated cells. This analysis resulted in a network characterized by top enriched clusters, such as cytokine signaling in immune system, ISG15 antiviral mechanism, extrinsic apoptotic signaling pathway, antigen processing and presentation, interleukin-1 processing, among others (Fig. 1G), depicting important cluster overlaps in the top enriched pathways both at 4 h and 24 h. Because the majority of the DEGs with largest fold changes were positively regulated upon LPS, we aimed to investigate the negative regulatory effect of LPS on gene expression. Accordingly, we performed GSEA of downregulated genes in microglia challenged with LPS and observed the highest shared enrichment in processes that comprise lysosomal metabolism (i.e. *Abca2, Lamp1, Cd63, Cd68, Hexa, Npl*), carbohydrate catabolism linked to glycolysis (i.e. *Pfkfb2, Akt1, Gpi1, Slc2a8*), pentose phosphate pathway (i.e. *H6pd*) TCA cycle (i.e. *Mdh1*), lipid catabolism (i.e. *Abcd1, Cpt1a, Gpx4, Acox3*), fatty acid synthesis (i.e. *Fasn*), autophagy (i.e. *Ulk1, Atg13, Atg7, Ubqln2*) and response to starvation (i.e. *Pparg, Apoe, Cd68, Foxo1, Foxo3, Jun*) (Fig. 1H).

Collectively, these results strongly suggest that upon LPS challenge, irrespective of the treatment duration, mouse microglia upregulate a battery of genes that are related to inflammation and immune activation, which is paralleled by downregulation of catabolic processes that aim to provide energy from substrates such as lipids, fatty acids, and cell organelles.

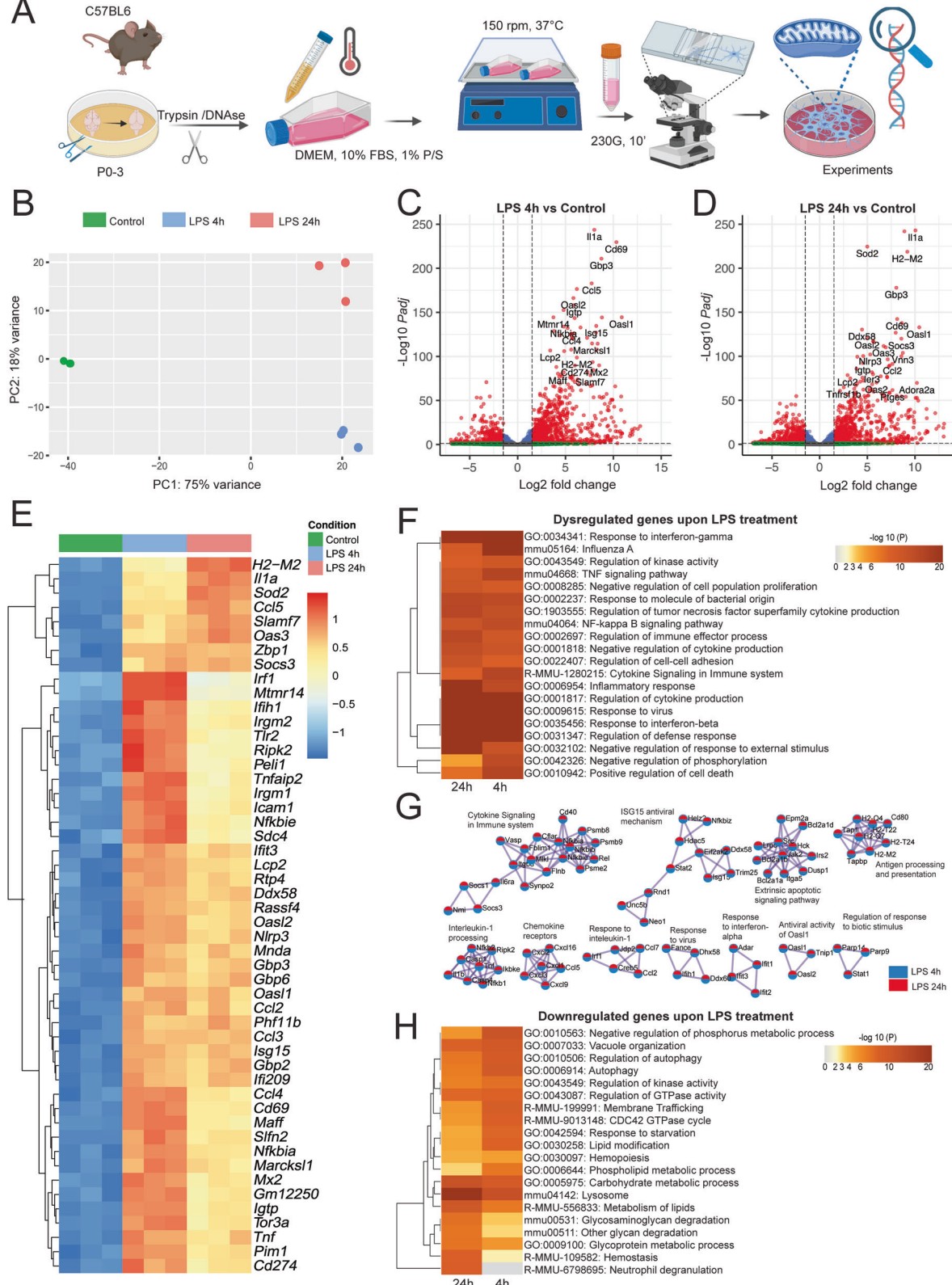

## LPS challenge of mouse microglia in vitro and in vivo induces transcriptomic signatures consistent with glycolytic upregulation

Our observations indicated that catabolic pathways such as lysosomal lipid degradation, pentose phosphate pathways and autophagy were downregulated in in vitro LPS-treated microglia. Based on these observations, we hypothesized that the metabolic reprogramming

induced by LPS in microglia preferentially involves a shift towards glycolysis as an upregulated metabolic pathway to support immune activation. To assess whether the LPS-induced transcriptomic changes in cultured microglia reflect the in vivo microglia response to LPS, we compared the gene expression profiles of LPS-treated cultured microglia with microglia isolated from mice, 3 h after an intraperitoneal (i.p.) injection with LPS[21], with a particular focus on genes of the

**Fig. 1 | Transcriptomic analysis of LPS-treated primary mouse microglia.**
**A** Experimental workflow of primary mouse microglia challenge. **B** Principal
Component Analysis (PCA) plot depicting each RNA-seq sample transcriptome,
colored by experimental group. Volcano plots depicting fold changes and -Log2 of
the adjusted *p*-value per gene comparing 4 (**C**) or 24 (**D**) h LPS treatment and the
untreated control (24 h). **E** Heatmap depicting the top 50 most significant differ-
entially expressed genes of 4 h LPS treatment compared with the control. Color key
corresponds to row Z-score. **F** Gene set enrichment analysis (GSEA) from the DEGs
of LPS application of 4 and 24 h, depicted as a hierarchically clustered heatmap.
Color key corresponds to the -Log10 of the *p*-value for enrichment score. **G** Protein-
protein interaction network analysis: proteins from the lists of DEGs of cells

challenged with LPS for 4 and 24 h, where Molecular Complex Detection (MCODE)
algorithm was applied to identify densely connected network neighborhoods,
where such neighborhood component is more likely to be associated with a par-
ticular complex or functional unit than the rest of the network. Legend denotes
whether the components were found in the DEG lists of either comparison. **H** GSEA
from the downregulated genes in mouse microglia challenged with LPS for 4 and
24 h, depicted as a hierarchically clustered heatmap. Color key corresponds to the
-Log10 of the *p*-value for enrichment score. *N* = 3 different animals per condition.
For all comparisons, *p*-value was determined by Wald–test with multiple-testing
correction by DESEQ2. Figure 1A was made with Biorender.

glycolytic and TCA pathways (Fig. 2). The in vivo data contained 1740
DEGs, while our in vitro data contained 1750 DEGs, of which 406 were
shared across both data sets (Fig. 2A). The overlap between in vitro and
in vivo gene sets (Fig. 2B, top plot) and the functional overlap between
gene sets (Fig. 2B: bottom plot) are shown by circos plots. GSEA
showed shared enrichment in inflammatory response pathways
(Fig. 2C). In depth analysis of the gene expression involved in glycolysis
demonstrated an upregulation of *Hk2, Hk3, and Pfkp*, and a down-
regulation of *Pfkfb2* and *Pfkfb4* following 4 h of LPS challenge. *Gapdh*
appeared lower at 4 h and increased at 24 h in vitro, while in vivo it was
significantly increased. *Hk2* continued to be upregulated following
24 h of LPS treatment, while *Pfkfb3, Pfkfb4, Aldh2, Ldhb, Aldh9a1,* and
*Gpi1* were downregulated (Fig. 2D, E). The analysis of glycolysis-related
genes showed consistent downregulation of *Pfkfb2, Pfkfb1, Pfkfb4,
Eno3* and consistent upregulation of *Hk2* and *Ldha* both in vivo and
in vitro (Fig. 2F). Hexokinases are critical enzymes in glycolysis as they
phosphorylate glucose and initiate glycolysis *Hk3* was downregulated
in vivo but not in vitro. On the other hand, *Pfkfb3* was upregulated
in vivo but downregulated in vitro (Fig. 2D, E).

Glucose transporters (GLUTs) are key players in glucose meta-
bolism and are not classically contained in the glycolytic gene sets (e.g.
Gene Ontology or Kyoto Encyclopedia of Genes and Genomes). We
hypothesized that GLUTs expression would be enhanced by LPS
treatment in mouse microglia. We observed an upregulation of the
transcripts that code for GLUT6 and GLUT8 (*Slc2a6,* and *Slc2a8,*
respectively) upon LPS 4 h and 24 h treatment, and GLUT1 (*Slc2a1*) only
at 4 h after LPS treatment (Supplementary Datasets 1–3).

All in all, mouse microglia in vitro and in vivo exhibit shared
dysregulation of glycolysis-related and other central metabolic gene
transcription following LPS challenge consistent with glycolytic upre-
gulation by increased GLUTs and HKs expression.

### LPS challenge of mouse microglia in vitro and in vivo induces transcriptomic signatures consistent with TCA dysregulation

Immune activation of macrophages and microglia has been linked to
metabolic changes and adaptation in glycolysis coupled to the TCA
cycle[11,13,22]. The final product of glycolysis is pyruvate, which can give
rise to either lactate via lactate dehydrogenase (LDH) or to acetyl-CoA
via pyruvate dehydrogenase (PDH). Acetyl-CoA enters the tri-
carboxylic acid (TCA) cycle generating NADH and FADH, which can
subsequently fuel the electron transport chain. We hypothesized that
the TCA cycle and PDH would be downregulated coupled to an
increase of glycolysis to promote preferential glycolysis and aerobic
fermentation into lactate. Indeed, we observed downregulation of
specific genes involved in TCA cycle such as *Idh1* and *Ogdh1*, but
upregulation of *Dld* following 4 h of LPS treatment. 24 h of LPS treat-
ment downregulated *Idh1, Mdh1, Idh2, Idh3g, Suclg2* and upregulated
*Dld* and *Dlst* (Fig. 2G).

Analysis of in vivo LPS-treated microglia showed downregulation
of TCA cycle genes *Idh1, Mdh1, Sdhb, Idh2, Pck2, Aco1,* among others
and upregulation of *Dld, Suclg2, Dlst, Idh3a,* and *Idh3b* (Fig. 2H). When
comparing the transcriptomic changes of microglia upon LPS treat-
ment in vivo and in vitro, we observed that the strongest up- and

downregulated shared genes of the PDH complex and the TCA cycle
were Dihydrolipoamide dehydrogenase (*Dld*, upregulated) and Iso-
citrate dehydrogenase (*Idh*, downregulated), respectively, (Fig. 2I). *Idh*
is the first of four oxidative steps within the TCA cycle, and *Dld* codes
for a mitochondrially-localized protein which serves as a NAD+ oxi-
doreductase in the pyruvate dehydrogenase multienzyme complex
(*Pdhc; Pdc; Pdh*). This complex is an associated set of three enzymes
that ultimately converts pyruvate to acetyl coenzyme A (acetyl-CoA).
Collectively, changes in the expression of the TCA genes strongly
indicate a decrease in TCA cycle activity following LPS challenge,
indicating that in mouse microglia, TCA genes are regulated during
early (4 h) inflammation.

### LPS challenge of human microglia induces an inflammatory gene signature

It has been proposed previously that mouse and human macrophage
responses to TLR4 activation could lead to differential outcomes in
terms of metabolic reprogramming[11]. To further investigate whether
this phenomenon occurs in human microglia, as well, we generated
iPSC-derived microglia-like cells (iMGLs), challenged them with LPS,
and assessed their gene expression with RNA-seq profiling. We gen-
erated iMGLs from control iPSCs and differentiated them by using a
modified method described initially by Fatorelli and colleagues[23].
Microglial progenitors were further differentiated with IL-34, TFG-β,
GM-CSF, CD200, and CX3CL1 to assure a full maturation profile.
Following 14 days of maturation, human iMGLs expressed microglial-
specific markers, TMEM119 and IBA1. In addition, qPCR indicated
upregulation of *P2YR12, TMEM119, CD11c* and low expression of
*Nanog* and *NeuN*, markers for pluripotency and neurons, respec-
tively, demonstrating the features and markers of mature microglia
(Fig. 3A). Differentiated human iMGLs exhibited ramified morphol-
ogies with a small round cell body and possessed dynamic surveying
movements. To further assess whether the gene profile of the dif-
ferentiated human iMGLs resembles the adult or fetal human brain
microglia, we compared their transcriptome and demonstrated that
the differentiated iMGLs were similar to fetal/adult brain
microglia[24,25] and iMGLs[26] generated with other available protocols.
Additionally, the transcriptome of iPSC-derived microglia clustered
apart from CD14+/CD16- and CD14-/CD16+ monocytes and dendritic
cells (Fig. 3B). We observed that in the microglial differentiation
protocol, there was a substantial collection-dependent effect on the
transcriptomes of the iMGLs (Supplementary Fig. 2A, B). Accord-
ingly, the collection number was included as a covariate in our ana-
lysis. However, it was evident that LPS elicited robust transcriptomic
changes in these iMGLs clustering separately.

In order to further validate the LPS-transcriptomic signature of
our human microglia-like cells, we performed a meta-analysis of pre-
viously published transcriptomes of LPS-treated human microglial
cells[27,28]. Although strikingly different in length (Supplementary
Fig. 3A–C), the functional overlap of biological processes that these
lists enrich for is shared for most of the detected biological processes
(Supplementary Fig. 3C, D). The top dysregulated processes that the
enrichment analysis detected comprised "inflammatory response",

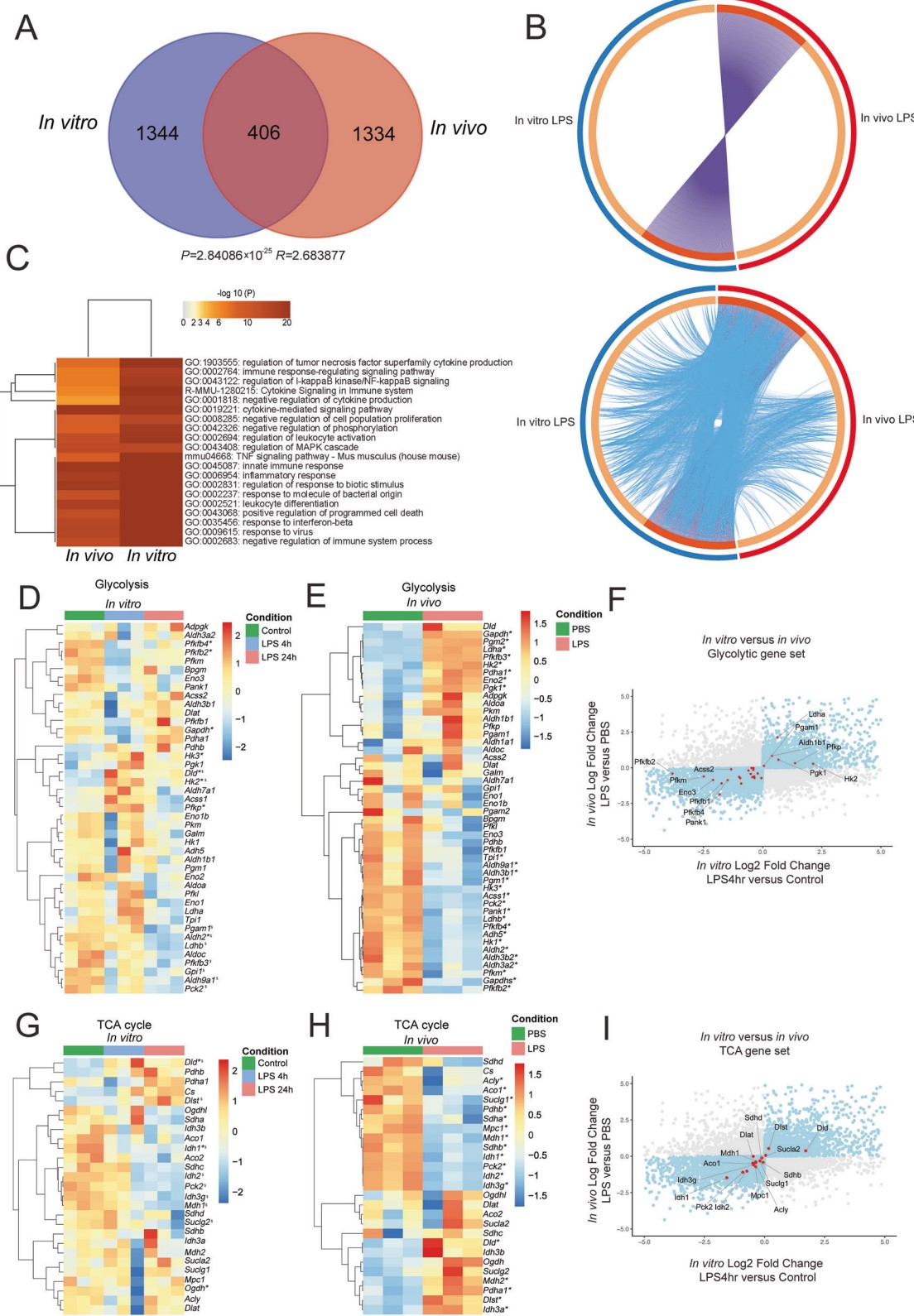

"cell activation", regulation of inflammatory response", "innate immune response", "cytokine signaling in immune system", (Supplementary Fig. 3D). Similarly, by examining the top 100 dysregulated processes (Supplementary Fig. 3E) we observed a major overlap in enrichment between these DEG lists.

These data showcase the reproducibility of microglial-like cell differentiation from iPSC and their response to LPS.

Comparing the effects of LPS at different time points following LPS challenge, we observed that LPS-induced genes *USP12*, *SOCS3*, *IL23A*, *CD274*, *FGF2*, *CCM2L*, *IL23A*, *IL6* were strongly upregulated after 4 h. *USP12* codes for a deubiquitinating enzyme, *SOCS3* codes for a protein member of the STAT-induced STAT inhibitor (SSI). *LPAR6*, *ADORA3*, *RREB1*, were downregulated after 4 h following LPS challenge (Fig. 3D). *LPAR6* codes for a Purinergic receptor from the P2Y family,

**Fig. 2 | Transcriptomic changes in LPS-treated primary mouse microglia and in vivo LPS-treated microglia. A** Overlap of DEGs in vitro primary neonatal microglia at 4 h after LPS stimulation and in vivo treated adult microglia 3 h after intraperitoneal LPS injection. The transcriptome analysis of the TCA/glycolytic genes of in vivo microglia was performed using recently published data[21]. Hypergeometric *p*-value and enrichment level are for such overlap are depicted. **B** Circos plots depicting the overlap in the lists of DEGs. On the outside, each arc represents the identity of each gene list, using the following color code: Blue, In vitro *primary microglia*; red, in vivo-*treated* microglia. On the inside, each arc represents a gene list, where each gene member of that list is assigned a spot on the arc. Dark orange color represents the genes that are shared by multiple lists and light orange color represents genes that are unique to that gene list. Purple lines link the same gene that are shared by multiple gene lists. For the plot in the bottom, blue lines link the genes that, although different, fall under the same ontology term. Blue lines indicate the amount of functional overlap among the input gene lists. **C** Heatmap depicting the top 20 statistically enriched terms (GO/KEGG, canonical pathways,

etc.) hierarchically clustered into a tree based on Kappa-statistical similarities among their gene memberships. The term with the best *p*-value within each cluster is shown as its representative term in the dendrogram. Heatmap cells are colored by their *p*-values. Hierarchically clustered heatmaps depicting gene expression changes in glycolytic pathways in vitro primary microglia (**D**) and in in vivo-*treated* microglia (**E**). Color key corresponds to row Z-score. **F** Scatterplot that graphically represents the fold changes of genes following LPS stimulation in vitro versus in vivo. Dots in blue depict genes that are increased or decreased in both models (in vitro and in vivo). Dots in red represent glycolytic genes. **G, H** Hierarchically clustered heatmaps depicting gene expression changes of TCA in vitro and in vivo-treated microglia. Color key corresponds to row Z-score. For the heatmaps, * denotes *padj* < 0.05 LPS 4 h versus Control, $ denotes *padj* < 0.05 LPS 24 h versus Control. These *padj* values can be found in Supplementary Datasets. **I** Scatterplot that graphically represents the fold changes of LPS stimulation in vitro versus in vivo. Dots in blue depict genes that are increased or decreased in both models. Dots in red represent TCA genes. *N* = 3 different animals per condition.

*ADORA3* codes for the Adenosine receptor A3, *RREB1* codes for a zinc finger transcription factor that binds to RAS-responsive elements (RREs). After LPS 24 h, *CCM2L, SEMA4A, CLECE, CCL5*, among others, were upregulated (Fig. 3E, Supplementary Fig. 2C, D and Supplementary Datasets 5–7). *CCM2L* codes for the Cerebral Cavernous Malformations 2 Protein-like, which has been reported to play a role in cell-adhesion and positive regulation of fibroblast growth factor (FGF) production, *CCL5* codes for a the chemoattract protein chemokine 5 ligand, *SEMA4A* codes for a member of the semaphoring family that has been implicated in nervous system development, *CLEC4E* codes for a member of the lectin-like superfamily, which are downstream targets of C-binding protein signaling. Genes such as *ITGA4* and *ITGA9* were downregulated upon LPS 24 h (Fig. 3E). *ITGA4* and *ITGA9* code for protein members of the integrins, proteins that mediate cell surface adhesion. We identified a cluster of upregulated genes after 4 h, but not after 24 h, such as *IL6, CCM2L, FGF2, MAP3K8, REL, TNFAIP2, SLAMF7, IL23A, SOCS3* (Fig. 3F). *IL6, IL23A, REL* code for key mediators in inflammatory responses. This battery of genes normalizes their gene expression in iMGLs after 24 h.

Similar to the mouse microglia, GSEA of statistically significant differentially expressed genes following LPS challenge of human iMGLs demonstrated shared enrichment of inflammatory pathways such as NF-κB signaling pathway, prostaglandin signaling, and immune response (Fig. 3G). For the iMGLs, the number of DEGs after LPS 24 h was considerably lower than upon LPS 4 h treatment (Supplementary Fig. 3E). To further illustrate this, we investigated the potential functional implications on the cells by predicting protein-protein interaction (PPi) maps from the DEGs for each comparison. Accordingly, we employed the MCODE algorithm on the PPi analysis resulting in a network characterized by top enriched clusters such as leukocyte chemotaxis, PI3K-Akt signaling, regulation of cysteine-type endopeptidase activity involved in apoptotic process, chemokine signaling, NF-kB signaling and positive regulation of cytosolic calcium ion concentration. This analysis demonstrated incomplete cluster overlaps in the top enriched pathways both at 4 h and 24 h, with 24 h being the lowest enrichment. Data showcased a substantial decrease of the potential PPis after 24 h, suggesting that the upregulated inflammatory pathways are dampened following 24 h of challenge compared to short-term exposure (4 h) (Fig. 3H).

Next, we investigated the effects of LPS on human microglia gene expression of downregulated genes by performing GSEA. We observed a marked shared enrichment of the biological processes linked to the regulation of small GTPase-mediated signal transduction (i.e. *CSF1, FBP1, IGF1, P2YR8, NOTCH1*), regulation of kinase activity (i.e. *CD86, MAP3K5, MAP3K5, ABCA7, CD4, CD74, TGFB1, TLR3*), glycerolipid metabolic process (i.e. *ABCD1, HEXA, HEXB, FAXDC2, ACSS2*), and the Tyrobp causal network in microglia (i.e. *RUNX3, CD4, GPX1, ITGAM, LYL1, MAF, RGS1, TGFBR1, TYROBP, LOXL3*) (Fig. 3I). Taken together, these data

indicate that human microglia derived from iPSCs present classical LPS-related transcriptomic responses coupled to downregulations of lipid synthesis, catabolism, and phosphate-based metabolism.

## Temporal transcriptomic responses to LPS are different in mouse and human models

We analyzed the overlap of DEGs between 4- and 24 h time points in mouse and human microglia observing that in mouse microglia, 402 genes are shared and ~670 genes are unique per condition. Furthermore, we performed GSEA and observed that the 402 (Supplementary Fig. 4A, B) genes that are shared across both conditions enrich for processes such as innate immune response, response to lipopolysaccharide, and interferon signaling (Supplementary Fig. 4B). Similarly, we performed the same approach to the iMGLs observing a distinct pattern than murine microglia. We found that the number of DEGs at 4 h LPS was 481 and for 24 h was 124, sharing 59 genes (Supplementary Fig. 4E). 65 genes were dysregulated 24 h after LPS and not after acute treatment with LPS for 4 h. This list of genes contains *STEAP1B, IGFBP4, FYN, TREML3P, PLD4, SLC6A7, PTGES, FAXDC2, ITGA9, CCL7, IGF1, ENOX1*, among others (Supplementary Fig. 4E). *IFBP4* codes for a protein member of the insulin-like growth factor binding protein (IGFBP) family of proteins, which prolong the half-life of the IGFs. *IGF1* was upregulated at this timepoint. *SLC6A7* codes for a brain-specific L-proline transporter. *TREML3P* codes for a pseudogene in the *TREM* locus that has been suggested to play roles in TREM-dependent responses[29]. Notably, *ITGA9* codes for an alpha integrin, crucial for cell-cell and cell-matrix adhesion.

For the 59 overlapping genes. we observed enrichment of processes such as leukocyte migration, inflammatory response, and mononuclear cell migration among others (Supplementary Fig. 4F). We found that these data support our conclusions on how the human microglia-like cells have a strong TLR4 activation at 4 hours but decreased its enrichment at 24 h. However, we did not see such an effect in mouse microglia, where TLR4 signaling maintained the transcriptomic enrichment of inflammatory pathways across time points. Because we observed a larger number of DEGs upon LPS 4 h and 24 h in mouse microglia, and a modest number of DEGs upon the LPS 24 h in human iMGLs, we compared their transcriptomic responses. We investigated the overlap in DEGs and the associated biological processes and observed a higher overlap in mouse microglia than in iMGLs (Supplementary Fig. 4C, G). Next, we calculated the slopes of the Log Fold Changes (LogFC) of DEGs that occurred at 4 h and 24 h by fitting them into a linear model. We observed a slope of *m* = 0.93 for mouse microglia (Supplementary Fig. 4D), and a slope of *m* = 0.7729 for iMGLs (Supplementary Fig. 4H). We surveyed the *Tlr4* and *TLR4* transcript abundance, and observed lower expression in mouse microglia compared to human (Supplementary Fig. 4I). These data showcase the relation between transcriptomic responses of 4 h and 24 h is almost

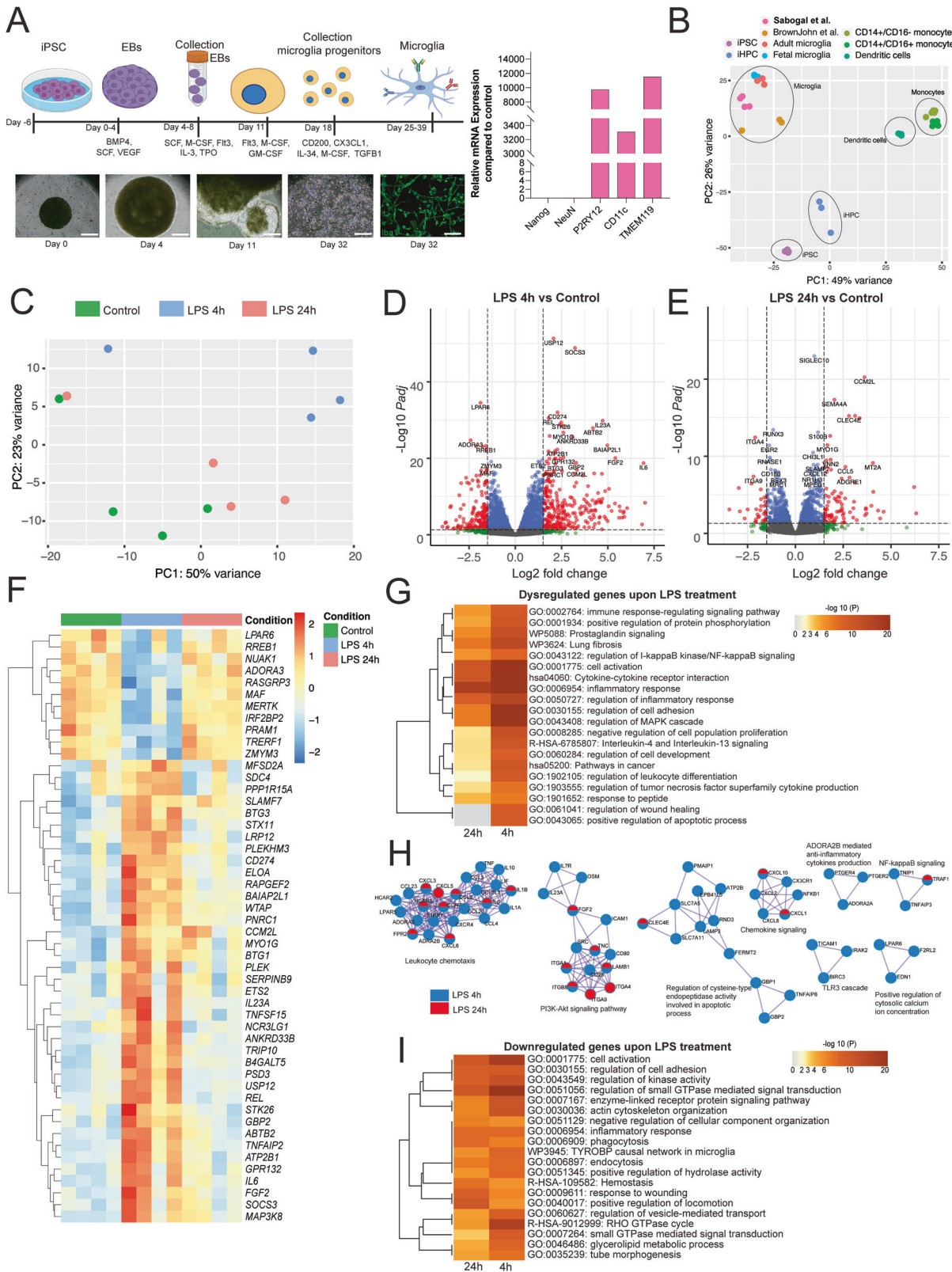

linear for mouse microglia, while it is stronger for 4 h and somewhat attenuated at 24 h in human iMGLs with different basal *TLR4* expression.

**LPS dysregulates the transcriptomic signature of glycolytic and TCA pathways in human iMGLs**

To follow-up on the transcriptomic dysregulations that LPS elicits on human iMGLs, we hypothesized that the genes that code for enzymes

of glycolysis would be dysregulated. We surveyed genes that code for components of this pathway and observed an upregulation of *ALDH1B1* and *PFKFB3*, and a downregulation of *PFKFB2*, *PFKFB4*, *ALDH3A2*, *HK1*, and *PFKM* at LPS 4 h compared to the control. Conversely, we observed upregulation of *PFKFB3*, *PKM*, *TPI1*, and *ENO3*, and a downregulation of gluconeogenesis genes (*FBP1*), or other genes (*GALM, ALDH7A1, ALDH3A2*) at LPS 24 h. Genes such as *PFKFB4, HK1*

**Fig. 3 | Transcriptomic analysis of LPS-challenged human iPSC-derived microglia. A** Differentiation protocol to generate iMGLs from human iPSCs. Representative micrographs of 3 independent experiments with similar results in various stages of the protocol. Bright-field images Day 0- Day 32 Scale bar: 250 µm. Immunofluorescence of Iba-1. Day 32. Scale bar: 50 µm. Representative plots 5 independent experiments with similar results of relative mRNA expression of Nanog, NeuN, P2RY12, CD11Cc, TMEM119 in iMGL. **B** Principal Component Analysis (PCA) depicting transcriptomes of various cell types and previously published protocols for iPSC-derived microglia. **C** PCA of untreated iPSC-derived microglia and LPS-stimulated microglia for 4 and 24 h. Volcano plots depicting fold changes and Log2 of the adjusted *p*-value per gene by comparing 4 (**D**) or 24 (**E**) h of LPS challenge and the untreated control (24 h). **F** Heatmap depicting the top 50 most significant differentially expressed genes of 4 h LPS treatment compared with the control cells. Color key corresponds to raw Z-score. **G** Gene set enrichment analysis (GSEA) from the DEGs upon LPS 4 and 24 h depicted as a hierarchically clustered heatmap. Color key corresponds to the -Log10 of the *p*-value for enrichment score. **H** Protein-protein interaction network analysis: proteins from the lists of DEGs of cells challenged with LPS for 4 and 24 h, where Molecular Complex Detection (MCODE) algorithm was applied to identify densely connected network neighborhoods, where such neighborhood component is more likely to be associated with a particular complex or functional unit than the rest of the network. Legend denotes whether the components were found in the DEG lists of either comparison. **I** GSEA of the downregulated genes following LPS 4 and 24 h challenge depicted as a hierarchically clustered heatmap. Color key corresponds to the -Log10 of the *p*-value for enrichment. *N* = 4 independent experiments per condition. For all comparisons, *p*-value was determined by Wald–test with multiple-testing correction by DESEQ2. Figure 3A was made with Biorender.

and *PFKM* which were downregulated 4 h post-LPS treatment, were normalized at 24 h post-LPS treatment (Supplementary Fig. 5A). We then surveyed for the genes that code for GLUT transporters in the human iMGLs. We observed transient upregulation of GLUT3 and GLUT6 (*SLC2A3*, *SLC2A6*, respectively) only at 4 h, and returning to control values at 24 h (Supplementary Datasets 5–7). These data strongly suggest that glycolytic flux of LPS-treated iMGLs may increase due to increased glucose uptake via GLUT transporters 3 and 6, with upregulation of *PFKFB3*.

For the TCA gene set, we observed a shared upregulation of genes that code for Aconitase 1 (*ACO1*) and downregulation of isocitrate dehydrogenase (IDH) genes *IDH1* (4 h) and *IDH2* at 4 h and 24 h LPS. Fumarate hydratase (*FH*) and Citrate Synthase genes (*CS*) were only upregulated at 4 h and not at 24 h (Supplementary Fig. 5B, Supplementary Datasets 5–7). These data highlight the upregulation of the genes whose protein expression would increase the TCA flux at 4 h and 24 h, with a downregulation of IDH.

## Glucose metabolism is differentially altered in mouse and human microglia challenged with LPS

To investigate how transcriptomic signatures are dysregulated across species upon the TLR4 agonist LPS, we analyzed one-to-one orthologues between mouse and human genomes and compared their fold-changes following LPS treatment. Our analysis demonstrated a shared downregulation of *PFKFB4* and *PFKM* and an upregulation of *DLD* in the in vitro cultured microglia following 4 h of LPS challenge (Fig. 4A). This gene signature (*PFKFB4* and *PFKM*) is characteristic for the glycolytic pathway. We observed differential induction of hexokinase isozymes: mouse microglia upregulated both *Hk2* and *Hk3*, while in human microglia *HK1* was downregulated at 4 h and *HK3* had a tendency towards downregulation after 4 h LPS treatment (Fig. 4B, D).

After 24 h, mouse microglia continued upregulating *Hk2* and downregulating *Gpi1*, *Pfkfb3*, and *Ldhb*, while human microglia did not exhibit the same pattern (Fig. 4C). *PFKFB3* was upregulated at 4 h and 24 h and *PFKB4* and *PFKFB2* were downregulated after 4 h and no other isoform of PFKFB was dysregulated (Fig. 4D) in the iMGLs. The upregulation of hexokinase transcripts in mouse microglia suggests that there may be an increased glycolytic flux. Additionally, human microglia upregulated *PFKFB3*, an important regulator of glycolysis, since increased PFKFB3 activity increases the rate of glycolysis. To further investigate the effects of LPS on mouse microglia and human microglia-like cells at the protein level, we performed label-free (LFQ) and targeted proteomics by mass spectrometry. Next, we compared the transcriptomic data with the proteomic data, focusing on the key enzymes of glycolysis: Hexokinases (HKs), Phosphofructokinases (PFKs), 6-phosphofructo-2-kinase/fructose-2,6-biphosphatases (PFKFBs) and Phosphoenolpyruvate kinases (PKs). HKs, PFKs and PKs are classically regarded as the "rate-limiting" steps of glycolysis. PFKFBs are bifunctional enzymes that regulate glycolysis by modulating the levels of fructose 2,6 biphosphate (F2,6BP). PFKFB3, among the PFKFB isoenzymes, has a higher kinase activity compared to phosphatase activity, resulting in increased concentrations of F2,6BP. Notably, F2,6BP serves as allosteric activator of PFKs.

We assessed the species-specific transcriptomic and proteomic signatures of these enzymes and observed that Hexokinase isoform gene and protein abundance was species-specific: *Hk3* transcripts were more abundant than other isoforms in mouse microglia (Fig. 5A), while *HK1* and *HK2* were more abundant than *HK3* transcripts in iMGLs (Fig. 5B). LFQ proteomic data demonstrated that HK2 was the most abundant isoform in mouse microglia (Fig. 5C) while HK1 was the most abundant isoform in iMGLs (Fig. 5D). Upon LPS treatment, mouse cells upregulated *Hk2* and *Hk3* transcripts (Fig. 5A) but proteomic differences were not striking (Fig. 5C, E). Human cells downregulated *HK1* transcripts in response to the LPS challenge (Fig. 5B), however, this was not captured by the proteomic analysis (Fig. 5D, F).

At the transcriptional level, it was clear that both mouse and human cells exhibited *Pfkl* and *PFKL* transcripts, respectively, as the most abundant isoforms (Fig. 5G, H). Similarly, we observed that phosphofructokinase liver (PFKL) was the most abundant isoform of the three forms of PFKs in mice and human microglia at the gene and protein level (Fig. 5I, J). In mouse microglia, PFKs gene and protein abundance did not significantly change following LPS stimulation (Fig. 5I, K). In human microglia, LPS stimulation increased the abundance of PFKL protein (Fig. 5J, L) but not the *PFKL* transcripts (Fig. 5H). Additionally, we observed an increased abundance of the isoforms PFKM and PFKP by our targeted approach in human but not mouse microglia (Fig. 5K, L). We observed transcriptomic changes of the phosphoenolpyruvate kinase isoform M (*PKM*) in human microglia only at 24 h after LPS stimulation (Supplementary Dataset 6). Furthermore, we performed targeted proteomic analysis for pentose-phosphate pathway (PPP) enzymes following LPS stimulation and observed a significant increase in Phosphogluconate dehydrogenase (PGD) in human microglia but not in mouse microglia (Supplementary Fig. 7). These data strongly suggest that glucose is shunted towards PPP in human microglia.

The allosteric activator of PFK is fructose-2,6-bisphosphate (Fructose 2,6BP), which is a product of PFKFB3. Conversely, PFKFB2 and PFKFB4 catalyze the degradation of Fructose 2,6BP to Fructose 6P. *Pfkfb4* and *Pfkfb2* were downregulated upon LPS in mouse microglia (Fig. 5M). *PFKFB3* was upregulated upon LPS in human microglia while *PFKFB2* and *PFKFB4* were decreased (Fig. 5N). Based on this analysis, we observed that mouse transcripts, such as *Hk2*, *Hk3*, and *Pfkp* increased in inflammatory conditions, while proteins did not exhibit striking alterations upon LPS challenge, as evidenced by the targeted proteomic analysis (Fig. 5O). The iMGLs exhibited a clearer effect on PFKs, and PFKBs (PFKFB1 and PFKFB3) in conditions of LPS challenge both at the transcript and protein levels (Fig. 5N, P).

To further study the transcriptomic dysregulations elicited by LPS in the iMGLs, we surveyed for key genes of interest for glycolysis in differential expression data as well as in published data from LPS-stimulated iMGLs from Alasoo[27] and Hasselmann[28]. Regarding

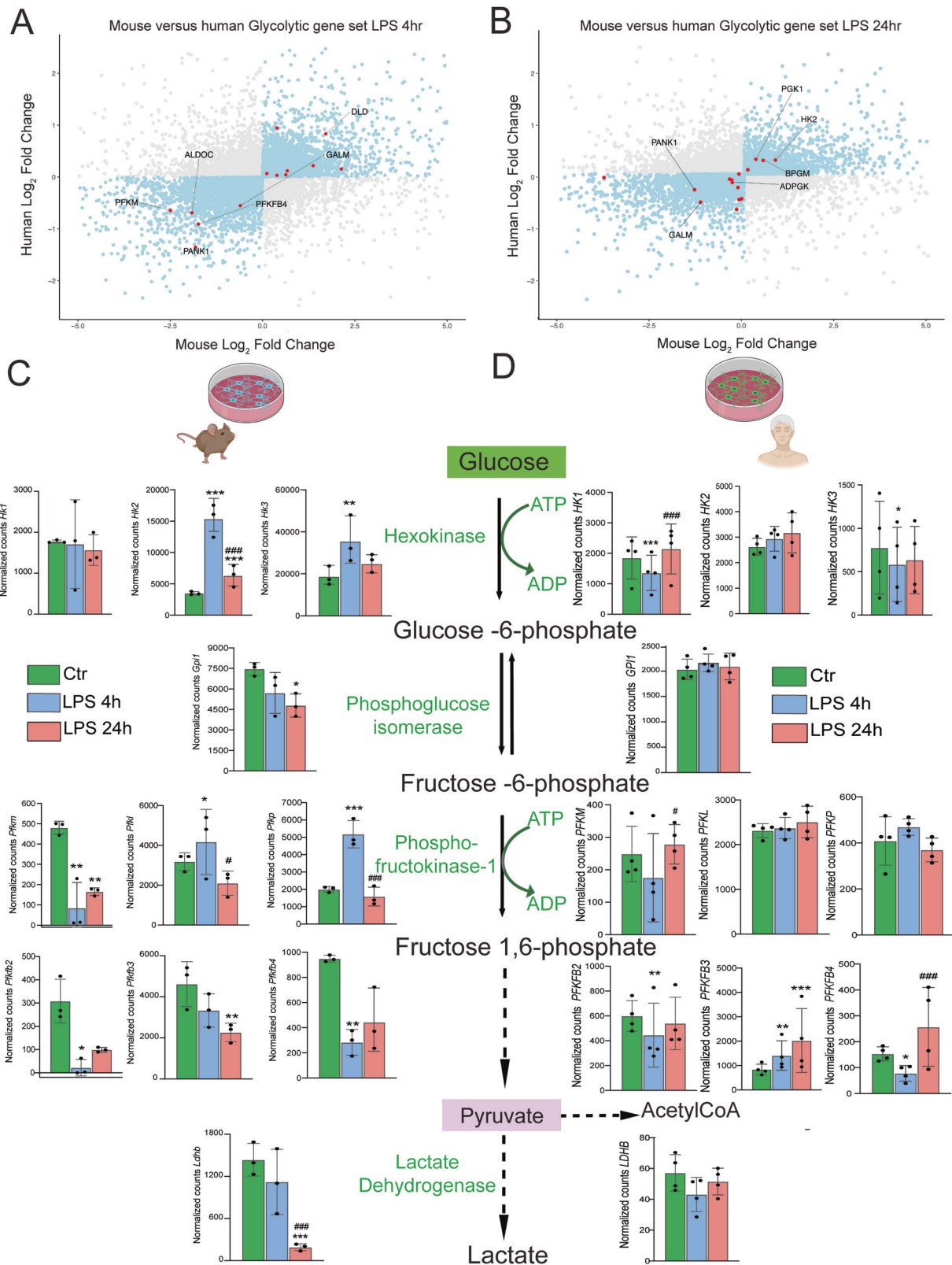

glycolysis, we observed overlap on significant upregulation of *PFKFB3* across models with discrepant changes on genes that code for hexokinases or phosphofructokinases (Supplementary Table 1).

Taken together, it is evident that the metabolic switch and the increase in glycolytic activity in microglia challenged with LPS is conserved across species. However, the enzymes that drive this upregulation of glycolysis are modulated in a species-specific manner.

Moreover, the baseline abundance of glycolytic enzymes, such as Hexokinases are different across species.

## Main enzymes of the TCA cycle are dysregulated in mouse and human microglia after LPS challenge

The final product of glycolysis is pyruvate, which can be converted to lactate or acetyl-CoA. In fact, the enzyme Dihydrolipoamide

**Fig. 4 | Transcriptomic comparison of mouse and human glycolytic genes.**
**A**, **B** Scatterplots that graphically represent the fold changes of LPS stimulation in mouse and human microglia. Dots in blue depict genes that are increased or decreased in both models. In **A** and **B**, dots in red represent glycolytic genes at 4 and 24 h, respectively. In (**C**) and (**D**) Control is represented in green bars, LPS 4 h is represented in blue bars and LPS 24 h is represented in light red bars. In **C** we observed several of the main glycolytic genes regulated in mouse and in **D**, in human. In **C**: *Hk2* (Ctr vs LPS 4 h: $p = 3.53E^{-17}$; Ctr vs LPS 24 h: $p = 0.0027$; LPS 4 h vs LPS 24 h: $p = 3.54E^{-6}$), *Hk3* (Ctr vs LPS 4 h: $p = 0.0038$), *Gpi1* (Ctr vs LPS 24 h: $p = 0.035$), *Pfkm* (Ctr vs LPS 4 h: $p = 0.0016$; Ctr vs LPS 24 h: $p = 0.0053$), *Pfkl* (LPS 4 h vs LPS 24 h: $p = 0.030$), *Pfkp* (Ctr vs LPS 4 h: $p = 5.99E^{-6}$; LPS 4 h vs LPS 24 h: $p = 2.36E^{-8}$), *Pfkfb2* (Ctr 4 h vs LPS 4 h: $p = 0.011$), *Pfkfb3* (Ctr 4 h vs LPS 24 h: $p = 0.00202$), *Pfkfb4* (Ctr 4 h vs LPS 4 h: $p = 0.00207$), *Ldhb* (Ctr 4 h vs LPS 24 h:

$p = 2.36E^{-12}$; LPS 4 h vs LPS 24 h: $p = 4.55E^{-12}$). **D**: *HK1* (Ctr vs LPS 4 h: $p = 0.00077$; LPS 4 h vs LPS 24 h: $p = 3.34E^{-12}$), *HK3* (Ctr vs LPS 4 h: $p = 0.035$), *PKFM* (LPS 4 h vs LPS 24 h: $p = 0.014$), *PKFFB2* (LPS 4 h vs LPS 24 h: $p = 0.0063$), *PKFFB3* (Ctr vs LPS 4 h: $p = 0.0022$; Ctr vs LPS 4 h: $p = 5.44E^{-7}$), *PKFFB4* (Ctr vs LPS 4 h: $p = 0.038$; LPS 4 h vs LPS 24 h: $p = 1.32E^{-4}$.) For the pathway scheme, dotted lines denote intermediate steps in the pathway. Data are presented as mean ± SD. For panel 4C, $N = 3$ animals per condition. For panel 4D $N = 4$ independent experiments per condition. For all comparisons, *p*-value was determined by Wald-test with multiple-testing correction by DESEQ2 except for *Pfkm* where *p* value was determined by one-way ANOVA (See supporting information) $^*p < 0.05$, $^{**}p < 0.01$, $^{***}p < 0.001$ (compared with ctr), $^#p < 0.05$, $^{####}p < 0.001$ (LPS 4 h vs LPS 24 h). Figure 4C and D were made with Biorender.

dehydrogenase (DLD) is one enzymatic component of the mitochondrial-based pyruvate dehydrogenase multienzyme complex in charge of the conversion of pyruvate to acetyl coenzyme A. We observed in human and mouse microglia an increase of *DLD* after 4 h, followed by a reduction after 24 h, as shown in the scatterplots and in the representative genes of the TCA pathway (Fig. 6A–D). At the same time, the isocitrate dehydrogenase enzymes (*IDH1* and *IDH2*) that catalyze the oxidative decarboxylation of citrate, resulting in 2-oxoglutarate, were downregulated in both species after 4 h and continued to be downregulated after 24 h (Fig. 6D). Likewise, Succinyl-CoA ligase [GDP-forming] subunit beta enzyme, encoding for the *SUCLG2* gene which catalyzes the reversible conversion of Succinyl-CoA to succinate and acetoacetyl CoA was downregulated after 24 h in both species, as shown in the scatterplot comparison and bar graphs (Fig. 6B–D). Transcriptomic data showed key enzymes such as *IDH, DLD, SUCLG2* are altered and their regulation might impact metabolic fluxes in response to LPS-mediated immune activation, thus supporting the metabolic reprogramming hypothesis.

By comparing our data with the data generated by Hasselmann[28] and Alasoo[27], within the TCA cycle gene list, we observed shared downregulation of *IDH1* and *IDH2*, with overlapping upregulations of *ACO1, ACO2* and unique upregulations of *CS* and *DLST* (Supplementary Table 2). All in all, these data point towards the LPS-dependent transcriptomic changes of metabolic pathways in various iPSC-derived microglial models.

Next, we compared the transcriptomic data with the proteomic data, focusing on the key enzymes of the TCA cycle. It was observed that transcripts of *Idh1* and *IDH1* were the most abundant isoforms in mouse (Supplementary Fig. 6A) and human cells (Supplementary Fig. 6B), respectively. With the proteomic analysis, we could determine that IDH2 is the most abundant isoform in mouse (Supplementary Fig. 6C) microglia, while both IDH1 and IDH2 have comparable abundances in human microglia (Supplementary Fig. 6D). The transcriptomic profile of the transcripts of TCA cycle enzymes showed a significant downregulation of *Idh1* for mouse (Supplementary Fig. 6E), upregulation of *ACO1, FH, CS* transcripts, and downregulation of *IDH1* and *IDH2* in human microglia (Supplementary Fig. 6F).

Overall, we observed no significant differences in the TCA cycle enzymes after LPS stimulation by label-free quantification (Supplementary Fig. 6G, H). However, the targeted proteomics approach yielded discrete upregulations that did not reach statistical significance in mouse microglia (Supplementary Fig. 6I), but yielded significant upregulation of ACO2,CS, FH, IDH2, IDH3A and MDH2 (Supplementary Fig. 6J) in human iMGLs. All in all, these results strongly suggest that TCA cycle activity may be enhanced in human cells, but not in mouse cells. Both species exhibit downregulation of *Idhs* and *IDHs* at the transcriptional level. However, proteomic differences become evident for upregulations of TCA cycle enzymes at the 4 h LPS-treatment timepoint.

## Differential response to LPS through time in mouse and human microglia

To assess the potential effect of metabolic reprogramming response on microglial morphology we used the xCELLigence cell impedance system. This system allows continuous monitoring of alterations in cell morphology by real-time cell impedance measurements which are indicated as normalized cell indexes. We observed an increase in the cell impedance in response to LPS in mouse and human microglia in the first 6 h (Supplementary Fig. 8A, B). However, the normalized cell index of human microglia started to decrease after 8 h (Supplementary Fig. 8B) to values similar to the control, which could mean the cells have changed their morphology, decreasing the size and area after a peak response; in the case of the mouse microglia the cell morphology alteration was maintained through the 24 h (Supplementary Fig. 8A). Staining with IBA1 following 24 h LPS stimulation demonstrated that the cell morphology was altered in mouse microglia and the cell shape changed from a resting, elongated to a large, amoeboid shape, considered a classical feature of microglial response to LPS stimulus (Supplementary Fig. 8C). Conversely, the human microglial phenotype after 24 h of stimulation did not appear drastically changed compared to the control iMGLs (Supplementary Fig. 8D), with fewer amoeboid cells compared to the mouse microglia. These morphological alterations probably reflect the major changes in impedance during the first 6 h of LPS challenge detected by the real-time xCELLigence measurements. These measurements showed a clear distinction between LPS response in mouse and human microglia. Subsequently, it was investigated whether microglial phagocytosis activity is comparable in both species. Phagocytosis of *E. coli* particles was increased by mouse and human microglia (Supplementary Fig. 8E, F) and the presence of cytochalasin D, an inhibitor of actin polymerization, blocked phagocytosis in both human and mouse microglia (Supplementary Fig. 8G, H), although phagocytotic activity and morphological changes of microglia presented slightly different behavior in both species.

## Species differences in metabolic reprogramming

To study the functional metabolic alterations in human vs mouse microglia, the bioenergetic profile of unstimulated and LPS-challenged microglia was evaluated using a Seahorse XF-analyzer. The analysis of Oxygen consumption rate (OCR) data of mouse microglia showed that LPS does not affect mitochondrial respiration after a short challenge (4 h), while long treatment (24 h) decreased mitochondrial respiration compared to the control mouse microglia (Fig. 7A). In contrast, the OCR of LPS-treated human microglia is decreased after 4 h and increased after 24 h compared to the control (Fig. 7B). We observed a decrease in the basal respiration of mouse microglia after 24 h compared to 4 h LPS treatment, however, we did not observe significant changes in the measurements of ATP-linked OCR and maximal OCR compared to the control (Fig. 7C–E). Human microglia showed a significant increase in basal respiration and ATP-linked OCR following 24 h LPS treatment compared to the control (Fig. 7F, G). The basal respiration and maximal OCR was decreased after 4 h LPS treatment in

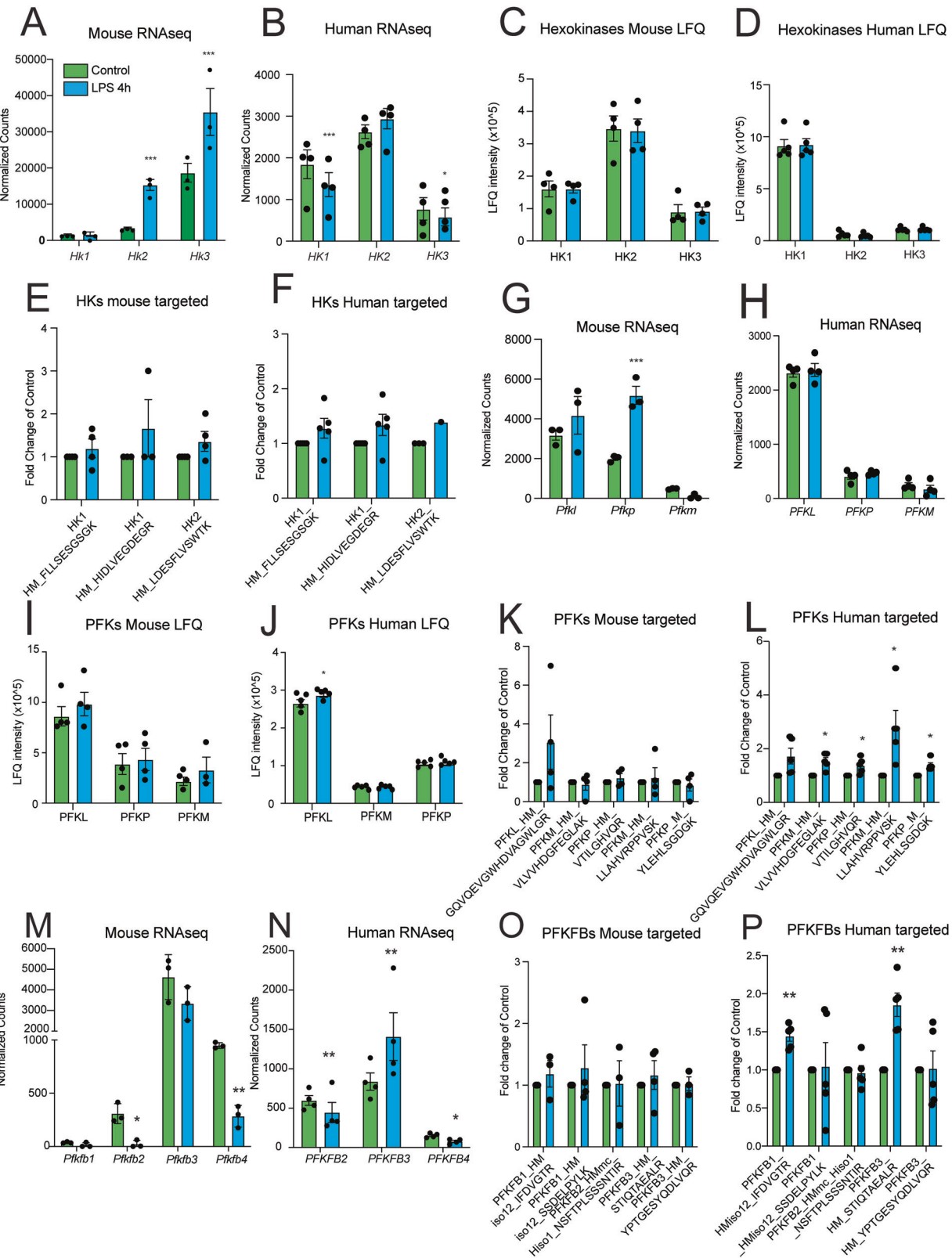

human microglia (Fig. 7F, H) We then measured ATP concentrations in LPS-treated microglia and observed that ATP levels were decreased at all timepoints for mouse microglia (Fig. 7I) and for human iMGLs (Fig. 7J).

The extracellular acidification rate (ECAR), an index of glycolysis feeding into lactate, was significantly increased after 24 h LPS stimulation in both species (Fig. 7K, L). Basal glycolysis and glycolytic

capacity were increased after 24 h LPS stimulation in mouse (Fig. 7M, N) and human microglia (Fig. 7P, Q). However, the glycolytic reserve was increased just in mouse microglia (Fig. 7O, R). To further characterize the glycolytic metabolism of these cells, we performed lactate determinations of the supernatant of LPS-treated cells, observing significant increases of lactate at all timepoints for mouse microglia (Fig. 7S) and for human iMGLs (Fig. 7T). This indicates that

**Fig. 5 | Transcriptomic and proteomic characterization of key glycolytic enzymes in mouse microglia and human microglia-like cells.** Mouse and human microglial cells were in vitro stimulated with LPS and collected 4 hours after stimulation for proteomic analysis by LC-MS/MS. Control is represented in green bars and LPS 4 h is represented in blue bars. **A**, **B** Normalized counts depicting transcript abundance of *Hk1*, *Hk2* ($p = 3.53E{-}17$), and *Hk3* ($p = 0.0038$) and *HK1* ($p = 0.00077$), *HK2*, *HK3* ($p = 0.035$). Label-free quantitation values from proteomic analysis depicting relative abundance of HK1, HK2, and HK2 in mouse (**C**) and human (**D**) cells. **E**, **F** Proteomic abundances of hexokinases measured by LC-MS/MS with internal standards. **E** Depicts plots of relative abundances on mouse microglia and (**F**) depicts human cells. Normalized counts depicting transcript abundance of *Pfkl*, *Pfkp* ($p = 5.95E{^}6$), and *Pfkm* (**G**) and *PFKL*, *PFKP*, and *PFKM* (**H**). Label-free quantitation values depicting relative abundance of PFKL ($p = 0.049$ for human), PFKP, and PFKM in mouse (**I**) and human (**J**) cells. **K**, **L** Proteomic abundances of phosphofructokinases measured by LC-MS/MS with internal standards. **K** Depicts plots of relative abundances on mouse microglia and **L** depicts human cells (*$p < 0.05$). Normalized counts depicting transcript abundance of *Pfkfb1*, *Pfkfb2* ($p = 0.01$), *Pfkfb3*, and *Pfkfb4* ($p = 0.002$) (**M**) and *PFKFB2* ($p = 0.0063$), *PFKFB3* ($p = 0.0022$) and *PFKFB4* ($p = 0.038$) (**N**). **O**, **P** Proteomic abundances of PFKFBs measured by LC-MS/MS with internal standards. **O** Depicts relative abundances on mouse microglia and **P** depicts human cells (**$p < 0.01$). For all panels, error bars represent SEM. Every biological replicate is depicted as a dot. For mouse RNAseq $N = 3$ different animals, for human RNAseq $N = 4$ independent experiments, for mouse proteomics $N = 4$ different animals and for human proteomics $N = 5$ independent experiments. For RNAseq data $p$-values were determined by the Wald test and multi testing corrected in DESEQ2. For proteomic data, $p$ values were determined by unpaired two-tailed $t$-test. *$p < 0.05$, **$p < 0.01$, and ***$p < 0.001$.

the capability of microglia to respond to an energetic demand, as well as, how close the glycolytic function is to their maximum capacity depends on the species. These data indicate that LPS stimulation affects mitochondrial respiration in both mouse and human cells, and the overall response in basal and glycolytic capacity is similar between the two species. Importantly, the metabolic reprogramming towards an increased glycolytic activity was demonstrated in both mouse and human microglia.

## Discussion

In this study, we report that mouse and human microglia undergo metabolic reprogramming upon LPS stimulation. Overall, LPS treatment promoted glycolytic metabolism in both species. Furthermore, increases in glycolytic activity are mainly attributed to hexokinase expression in mouse, and phosphofructokinase expression in human microglia. Both species upregulated glucose transporters. However, GLUT6 was only upregulated in human iMGLs at 4 h LPS, while in mouse microglia continued to be upregulated at 24 h. Additionally, oxidative metabolism was suggested to be downregulated in both species by transcriptomic analysis, while functional experiments demonstrated a differential time profile of maximal FCCP stimulated flux: in mouse a slow decrease that becomes evident after 24 h, whereas in human microglia there is a fast, transient decrease (4 h) which is normalized after 24 h. Here, we report cross-species transcriptional and functional responses of microglia challenged by inflammatory stimuli (Fig. 8).

We compared the transcriptomic response of mouse microglia in vitro and in vivo, since in vivo mouse microglia might behave differently compared to cultured acutely-isolated microglia[2,30]. In the studies of Gosselin and colleagues[30], primary microglia were cultured ex vivo by obtaining microglia enriched fractions by percoll gradient from mice that were 8-9 weeks of age. Subsequently, these fractions were purified by cell sorting and selecting for live/DAPI- CD11b+ CD45Low. These microglia were maintained in culture for 6 h, 24 h or 7 days with DMEM/F12 supplemented with 5% FBS (Supplementary Table 3). The aim of their studies was to assess how environmental disruptions modify the transcriptome of macrophage populations. In the current study, primary microglial were cultured in conditioned medium from astrocytes. The study of Bohlen[31] and colleagues showcases the importance of astrocytic support to define microglial phenotypes in vitro. In light of the nature of the acute ex vivo cultures, where the astrocytic support is lost, the strength of our approach is that we employ astrocyte-derived conditioned media that supports microglial growth and homeostatic phenotype. It is crucial to note that the study of Bohlen demonstrated that primary cultured microglia with their protocol do not show hallmarks of overt inflammation and it was comparable to freshly isolated microglia. Similarly, LPS- transcriptomic responses of in vitro cultured microglia were comparable between serum-supplemented and serum-deprived cultures as long as the cues from astrocyte conditioned medium are supplied[31].

In the pursuit of robust models for biomedical research, murine models have been commonly used, to ensure the translation of data to human physiology or pathology. However, there are some aspects that remain as central concerns: i) The disparities in metabolic pathways between these two species can potentially impact the reliability of extrapolations. Recent research by Matsuda et al. in 2020, showed that human presomitic mesoderm cells exhibit slower metabolism compared to their murine cells when they evaluated the transcription factor HES7. Similar results have been shown in generation of motor neurons[32] and biochemical reaction speeds of the p53-Mdm2 network[33], highlighting the intricate interplay between species and their metabolic dynamics. ii) These differences are not limited to inter-species disparities alone. Over the past few decades, metabolic differences have been identified within various strains of mice, not only in liver metabolism as noted by Bulfield et al. in 1977[34] but also in brain and other tissues, as described by Burlikowska et al. in 2020[35]. iii) Expanding the scope of the study, intriguing distinctions could emerge even during cellular reprogramming, as it was shown a notable divergence in bioenergetic remodeling between induced pluripotent stem cells and fibroblasts[36]. Collectively, it is fundamental to underline that both inter-species divergence and specific experimental conditions play a significant role over metabolic processes.

Cultivation and differentiation of primary murine microglia cultures provided crucial cues on the importance of experimental conditions, such as medium composition, or serum presence. Serum deprivation in primary microglia led to a significant cell death, nuclei condensation[37,38] and cell death markers, such as p-p38 and p-ERK[39]. Therefore, several (growth) factors tested to replace the use of serum, such as M-CSF[40], M-CSF, GMCSF, NGF and CCL-2[41]) could lead to viable microglia with primary and secondary processes. More recently, Bohlen et al., 2017 proposed the addition of three important factors: CSF-1/IL-34, TGF-β2, and cholesterol to prevent microglial cell death[31]. These cytokines have been used for murine primary microglia cultures[42,43] and human primary microglia obtained from brain patients[44]. Extra addition of CD200, CX3CL1 was also proposed to support survival and human microglial response[45]. However, it has been shown that the use of serum further enhances survival and proliferation[46]. While uncertainties persist regarding the optimal culture models for recapitulating the properties and functions of microglia in the intact central nervous system (CNS), we have chosen the most used protocol for generating primary microglia, which involves the inclusion of 10% Fetal Bovine Serum (FBS). This approach allows us to establish a basis for comparison with a majority of prior studies. Although, Dorion et al. 2022[45] have suggested that the media formulation exerts only a minor influence on culture-induced alterations in the human microglial transcriptome; we recognize that the addition of serum or other reagents within the medium could potentially give rise to significant changes in the metabolic profile.

Current literature on LPS responses in humans and mice suggests that LPS sensitivity is different across species. One may come to this

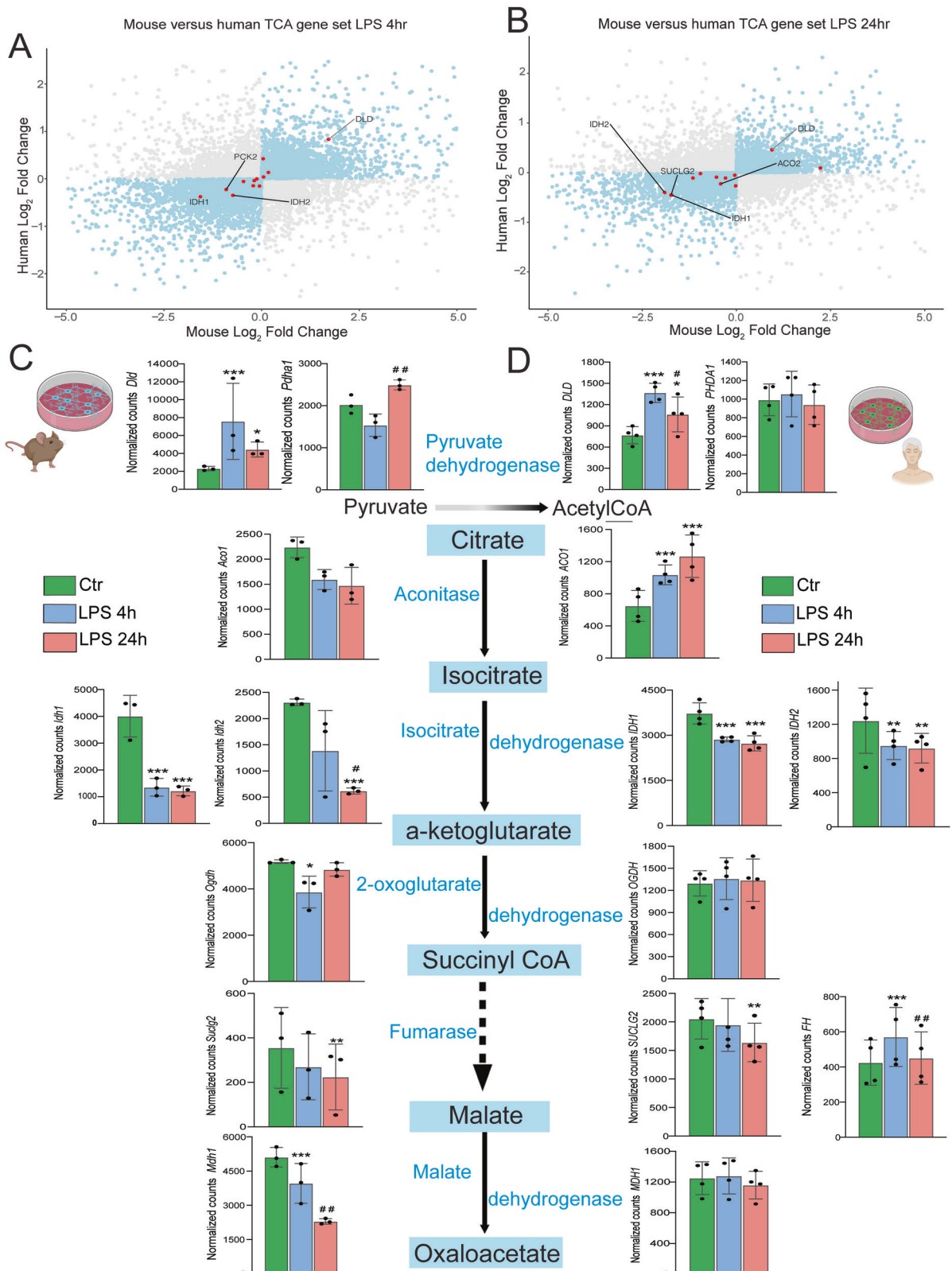

conclusion due to the range of working concentrations and dosing between humans and mice. However, several considerations must be noted: i) In vivo LPS administration in humans (intravenous) and mice (intraperitoneal) follows comparable temporal evolution, being TNF, and IL-6 main readouts in the 2–6 h following administration. However, ii) available literature raises further questions such as the arbitrary working doses for LPS administration (0.4–2.0 ng/kg body weight in

humans and 100–800 ug/kg body weight in mice). It is important to mention that iii) the dosing in mice had the main objective of inducing sickness behavior and that iv) intraperitoneal administration yields lower plasma concentrations (~10 times) of LPS than intravenous. v) Intraperitoneal administration is slower than intravenous. This discrepancy in dosing does not strictly mean that mice have lower sensitivity, but it highlights that we lack proper translational approaches

**Fig. 6 | Transcriptomic comparison of mouse and human TCA cycle genes.**
**A, B** Scatterplots represent the fold changes of LPS stimulation in mouse and human microglia. Dots in blue depict genes that are increased or decreased in both models. In **A** and **B**, dots in red represent TCA genes at 4 and 24 h, respectively. Several of the main TCA cycle genes are regulated in mouse (**C**) and in human (**D**) microglia. Dotted lines denote intermediate steps in the pathway. Enzymes are depicted in blue font. In (**C**) and (**D**) Control is represented in green bars, LPS 4 h is represented in blue bars and LPS 24 h is represented in light red bars. In C: *Dld* (Ctr vs LPS 4 h: $p = 1.11E^-5$; Ctr vs LPS 24 h: $p = 0.03$), *Pdha1* (LPS 4 h vs LPS 24 h: $p = 0.010$), *Idh1* (Ctr vs LPS 4 h: $p = 1.39E^-7$), *Idh2* (Ctr vs LPS 24 h: $p = 7.82E^-5$; LPS 4 h vs LPS 24 h: $p = 0.040$), *Suclg2* (Ctr vs LPS 24 h: $p = 0.0071$; LPS 4 h vs LPS 24 h: $p = 0.015$), *Mdh1* (Ctr vs LPS 24 h: $p = 7.57E-7$; LPS 4 h vs LPS 24 h: $p = 0.0025$). In D: *DLD* (Ctr vs LPS 4 h: $p = 1.92E-7$; Ctr vs LPS 24 h: $p = 0.020$; LPS 4 h vs LPS 24 h: $p = 0.044$), *ACO1* (Ctr vs LPS 4 h: $p = 0.00065$; Ctr vs LPS 24 h: $p = 1.89E^-6$), *IDH1* (Ctr vs LPS 4 h: $p = 0.00039$; Ctr vs LPS 24 h: $p = 5.09E^-5$), *IDH2* (Ctr vs LPS 4 h: $p = 0.025$; Ctr vs LPS 24 h: $p = 0.019$) *SUCLG2* (Ctr vs LPS 24 h: $p = 0.0163$), *FH* (Ctr vs LPS 4 h: $p = 0.00038$; LPS 4 h vs LPS 24 h: $p = 0.0061$). Data are presented as mean ± SD. For panel 6C, $N = 3$ animals per condition. For panel 6D $N = 4$ independent experiments per condition. For all comparisons, $p$-value was determined by Wald−test with multiple-testing correction by DESEQ2. $^*p < 0.05$, $^{**}p < 0.01$, $^{***}p < 0.001$ (compared with ctr); $^#p < 0.05$, $^{##}p < 0.01$ (LPS 4 h vs LPS 24 h). Figure 6C and D were made with Biorender.

to study LPS responses across species. In our study, we have employed the same LPS concentration to stimulate human and mouse microglia in vitro models as a starting point to objectively assess LPS-sensitivity. However, further characterization of TLR4 signaling and epigenomic landscapes that modulate transcriptional responses between species remains to be investigated.

Overall, the induction of the glycolytic genes *Hk2*, *Pfkp* and *Pkm* in the in vivo-treated microglia was robust, and *Hk2* and *Pfkp* induction was recapitulated in vitro cultured cells, as well. Both models did not show significant dysregulation of *ldha* while a decrease of *ldhb* was detected only in vitro cultured cells. LDHA has a higher affinity for pyruvate, and depending on the metabolite concentrations preferentially converts pyruvate to lactate, and NADH to NAD+ under anaerobic conditions, whereas LDHB possess a higher affinity for lactate, preferentially converting lactate to pyruvate, and NAD+ to NADH[47]. The so-called "Warburg effect", which was first described in cancer cells and later identified in immune cells, comprises a metabolic switch from oxidative metabolism towards glycolysis without a hypoxic environment, namely aerobic glycolysis preference, which is dependent on LDHA induction[48]. Previous studies in other immune cells have supported this notion (Supplementary Table 4). Overall, transcriptomic analysis and lactate determination of both microglia indicated an induction of glycolytic metabolism without hypoxic conditions.

The differentiated human iMGL cells resembled human microglia by both transcriptomic and functional analyses. Whole-transcriptome analysis of our iMGLs and other cell types demonstrates a clear differentiation course of iPSCs to iHPCs and then microglia, clustering far from monocytes and dendritic cells. Traditionally, microglial differentiation strategies have been criticized and described to provide monocytic-like in vitro phenotypes, similar to cultured primary human microglia[1]. As demonstrated by the study of Gosselin and colleagues[2], brain environment signals, such as TGF-β1 are necessary for the maintenance of the in vivo phenotype of microglia. Indeed, we have matured microglial progenitors in the presence of IL-34, the tissue-specific ligand of CSF-1 receptor (CSF-1R)[49], and TGF-β1. Our experiments provide functional microglia-like cells that can be used to study mechanisms of disease and effects of brain microenvironment in the future.

The essential steps of glycolysis involving hexokinases, phosphofructokinase-1 were dysregulated in the transcriptomic datasets of both mouse and human microglia. When any fast (~ 4 h) transcriptomic dysregulation occurred, no changes in metabolism were observed. In contrast, long LPS stimulation (24 h) elicited robust metabolic changes but subtle transcriptomic changes in all models. This discrepancy highlights the timeline between transcriptomic, translational, and functional responses in our model. These data, coupled to lactate measurements suggest that 4 h LPS could represent a timepoint to assess transcriptomic induction of glycolytic genes, while functional responses may be detected.

It is generally well-accepted that macrophages undergo metabolic reprogramming upon LPS treatment, a vital process for inflammatory responses[50–55]. Experiments with murine microglia and human

microglial cell lines[56,57] recapitulate key hallmarks of previous macrophage studies, such as the upregulation of glycolysis activity. Indeed, we observed an overall promotion of glycolysis, coupled to mitochondrial dysfunction in mouse microglia, supported by transcriptomic and functional assays. Our findings are in line with a recent study by Geric and colleagues[22] which reported increased glycolytic flux upon LPS stimulation in mouse microglia with attenuation of oxidative metabolism.

Immune tolerance is a host-protective mechanism in which cells become unresponsive to subsequent stimulation and has been proposed to underlie pathophysiology of neurodegenerative diseases. LPS-induced maternal inflammation during late gestation was shown to differentially affect microglial responsiveness during adulthood between hippocampi microglia and total brain microglia[58], hinting to tolerogenic mechanisms. It has been reported that tolerogenic murine microglia tend to normalize their glycolytic metabolism and increase their oxidative metabolism[13], rendering them dysfunctional with poor phagocytosis capacity, chemotaxis, and inflammatory responses. Treatment with IFN-γ boosted glycolytic metabolism and reversed the defective microglial responses. However, human studies on the context of immune tolerance and metabolic reprogramming will be useful to assess whether boosting glycolysis in the chronic setting or inhibiting glycolysis in the acute setting will prove beneficial for therapy when microglial is reactive in a disease context. A recent study reported that mature dendritic cells exhibit a metabolic signature consistent with upregulation of OXPHOS protein complexes, higher oxygen consumption ratio compared to immature dendritic cells[59]. Our data show that human iPSC-microglia exhibit these features. Whether human mature innate immune cells exhibit this metabolic signature remains to be further investigated.

Most transcriptomic changes in glycolysis occurred in hexokinase and phosphofructokinase in mouse microglia, while transcriptomic and proteomic changes occurred in the PFKs and PFKFBs in human iMGLs. The direct interrogation of mRNA and protein changes is a topic that remains to be further investigated as a plethora of mechanisms which are species-specific govern transcription, translation, and protein homeostasis. Reports on human microglial phenotypes and subpopulations that are associated with disease hallmarks, aging or metabolic states have demonstrated that transcription of HKs or PFKs are differentially regulated in association with Aβ responses similarly to inflammatory microglia, while tau responses were comparable to homeostatic microglia[4,60]. Some studies have proposed glucose transporters[61], HK2, and PFKs as therapeutic targets in neurodegenerative diseases with altered microglial functions aiming to attenuate glycolytic fluxes. However, we speculate that this will be beneficial only depending on the specific metabolic signatures of microglia. Given the discrepant relation between decreased OCR and ATP with gene and protein expression of TCA cycle enzymes for human and mouse cells at LPS 4 h, we hypothesize that the decreased OCR in human cells may be confounded by biochemical properties such as reaction speeds, half-lives, cellular environments, feedback loops and degradation rates of TCA cycle enzymes at this time point. Such changes have been recently reported during development[62].

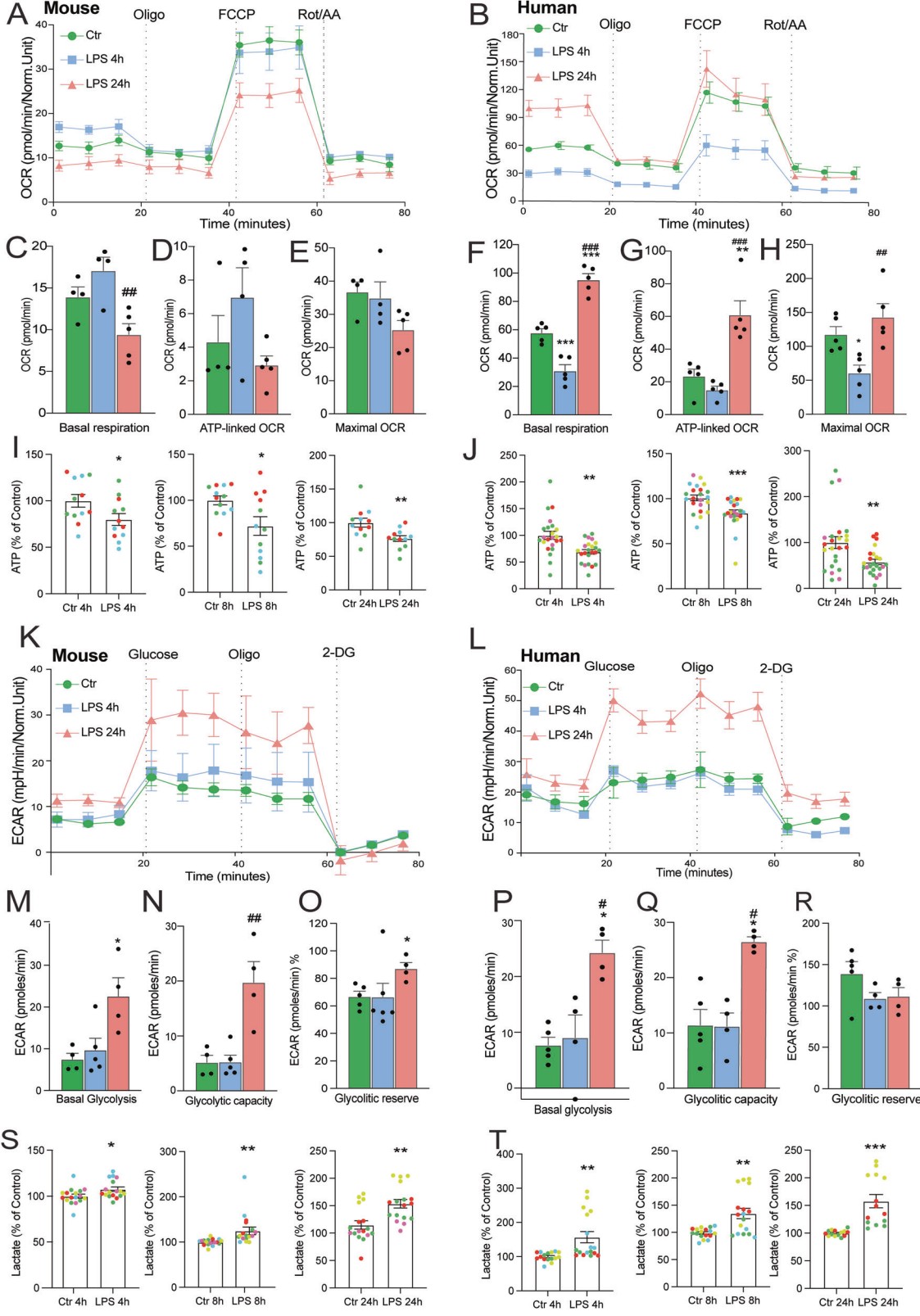

A difference in such parameters would mechanistically imply that upon an increased gene and protein expression, OCR and ATP would remain unchanged or decreased but not increased in human cells at 4 h. Such biochemical characterizations and their relation to species-specificity remain to be investigated.

Mouse iPSC-derived microglia were described and characterized in 2017[63], concluding that their gene expression is consistent with primary neonatal microglia. Although in this study, iPSC-derived mouse microglia were not studied, we compared the LPS responses of primary neonatal microglia with the human iMGL cells, which may also exhibit young phenotypes.

Our results should be interpreted in the light of the wide plethora of the experimental setups of primary microglia and iMGLs that prevail in the field (i.e. FBS presence and LPS concentrations). While mouse

**Fig. 7 | Mitochondrial respiration analysis of mouse and human microglia.**
Representative experiments showing OCR (oxygen consumption rate) in mouse (**A**) and human (**B**) microglial cells treated with LPS for 4 h and 24 h. For panels **A**, **C**–**E**: $N = 4$ Control and LPS 4 h; $N = 5$ LPS 24 h technical replicates of a representative experiment that was repeated 4 times with similar results. **C**–**H** Bar graph showing different mitochondrial respiration parameters represented as mean ± SEM of the OCR values in mouse (in (**C**), LPS 4 h vs LPS 24 h, $p = 0.006$) and human (**F**–**H**). For panels **B**, **F**–**H**: $N = 5$ Control, LPS4h and LPS 24 h technical replicates of a representative experiment that was repeated 4 times with similar results. (In (**F**), Ctr vs LPS 4 h, $p = 0.001$, Ctrl vs LPS 24 h, $p = 4.74E\text{-}5$, LPS 4 h vs LPS 24 h, $p = 1.66E^{-7}$; in (**G**), Ctr vs LPS 24 h, $p = 0.0015$, LPS 4 h vs LPS 24 h, $p = 0.0003$; in (**H**), Ctr vs LPS 4 h, $p = 0.044$, LPS 4 h vs LPS 24 h, $p = 0.005$). ATP measurements of LPS-treated mouse microglia (**I**) or human iMGLs (**J**) for several timepoints colored by experiment. For (**I**) $N = 12$ technical replicates across 4 experiments (LPS 4 h, $p = 0.042$, LPS 8 h, $p = 0.02$, LPS 24 h, $p = 0.0068$) and for (**J**) $N = 23$ Ctr 4 h, LPS 4 h, $N = 22$ Ctr 8 h, LPS 8 h, $N = 23$ Ctr 24 h and LPS 24 h technical replicates across 4 experiments (LPS 4 h, $p = 0.001$, LPS 8 h, $p = 0.0009$, LPS 24 h vs Ctrl 24 h, $p = 0.0047$). **K**, **L** Representative experiments depicting extracellular acidification rates (ECAR) in mouse and human microglia cells treated with LPS for 4 h and 24 h. For panels

**K**, **M**–**O**: $N = 4$ Control; $N = 5$ LPS 4 h; $N = 4$ LPS 24 h technical replicates of a representative experiment that was repeated 4 times with similar results. Bar graphs showing different acidification parameters represented as mean ± SEM of the ECAR values in mouse (in (**M**), Ctr vs LPS 24 h, $p = 0.018$, LPS 4 h vs LPS 24 h, $p = 0.032$; in (**N**), Ctr vs LPS 24 h, $p = 0.0036$, LPS 4 h vs LPS 24 h, $p = 0.0026$; in (**O**), Ctrl vs LPS 24 h, $p = 0.042$) and in human (in (**P**), Ctr vs LPS 24 h, $p = 0.0027$, LPS 4 h and LPS 24 h, $p = 0.0067$, in (**Q**), Ctr vs LPS 24 h, $p = 0.0024$, LPS 4 h vs LPS 24 h, $p = 0.0032$). For panels **L**, **P**–**R**: $N = 5$ Control; $N = 4$ LPS 4 h and LPS 24 h technical replicates of a representative experiment that was repeated 4 times with similar results. Lactate measurements of the supernatants of mouse microglia (**S**) and human iMGLs (**T**) treated with LPS for 4, 8 and 24 h expressed as a percentage of the untreated control for the same timepoint colored by experiment. Data are shown as mean ± SEM. For (**S**) $N = 17$ technical replicates across 4 experiments. For (T) $N = 16$ Ctr 4 h, $N = 17$ LPS 4 h, Ctr 8 h, LPS 8 h, $N = 14$ Ctr 24 h, LPS 24 h. (LPS 4 h, $p = 0.003$, LPS 8 h, $p = 0.001$, LPS 24 h, $p = 0.0003$). $p$-value was determined by one-way ANOVA for OCR and ECAR, and unpaired two-tailed $t$-test for ATP and lactate. $^*p < 0.05$, $^{**}p < 0.01$, $^{***}p < 0.0001$ (compared with ctr), $^\#p < 0.05$, $^{\#\#}p < 0.01$(LPS 4 h vs LPS 24 h).

microglia could be LPS-stimulated in vivo and we could study their transcriptome profile, human microglia with a strong inflammatory phenotype analogous to LPS, such as sepsis or bacterial encephalitis could prove useful to survey microglial phenotypes and responses in vivo. Such RNAseq studies have not been reported yet. However, single-cell transcriptomic datasets that are valuable resources of microglial gene signatures under different conditions, such as the study of Gerrits and colleagues[4] have demonstrated that microglial subpopulations that react to Aβ have a transcriptomic profile similar to inflammatory microglia. For instance, *PFKFB3* was downregulated in amyloid-related microglia, but not in tau-related microglia. This biological proxy lets us survey metabolic genes that are dysregulated in AD, PD[64], and MS[65].

Overall, the aim of our study was to further characterize a human iPSC-derived microglia model system. We propose further investigation on species-specific mediators that may govern oxidative metabolism such as enhancer-promoter interaction landscapes and comparison of enzymatic activity of key glycolytic and oxidative enzymes across models to learn more about human-specific regulation to advance disease modeling in neurodegeneration.

In conclusion, we have shown a systematic comparison between human and mouse responses to the TLR4 agonist LPS. Despite the fact that human and mouse immune system share extensive similarities, as we observed here with the glycolytic upregulation in mouse microglia (in vitro and in vivo) and human iMGLs, many discrepancies have been observed as well. For example, several enzymes such as PFKL, PFKP, PFKFB3, ACO1, MDH1 that are involved in glucose and TCA metabolism, are differentially regulated after LPS stimulation in mouse and human microglial cells. The increases in glycolytic activity are mainly attributed to hexokinase activity in mouse, and phosphofructokinase activity in human microglia. Considering that previous studies have not directly compared the energy metabolism between iMGLs and mouse microglia, this study presents the transcriptome profile and functional data on mouse and human microglia delineating their capacity to undergo metabolic reprogramming and confirming their relevance in translational research, which could provide a platform for targeting strategies for conditions associated with inflammation.

## Methods
### Animals
C57BL/6 J mice from the central animal laboratory at University of Groningen were housed and handled in accordance to Dutch standards and guidelines (Protocol 171224-01-003). All experiments were

approved by the University of Groningen Committee for Animal Experimentation.

### Primary microglia culture
Microglia cultures were prepared as previously described[20]. Briefly, brains were removed from 0- to 3-day-old C57Bl/6 pups, minced, dissociated for 25 min in 0.25% Trypsin, 1xDNAse, and then cells were cultured in Dulbecco's modified Eagle containing (DMEM, Gibco, 42430025), supplemented with 10% fetal bovine serum (FBS, Hyclone, SV30160), 1 mM Sodium Pyruvate (ThermoFisher, 15070063), and 100 U/mL pen/strep (ThermoFisher, 11360070) in the incubator at 37 °C and 5% $CO_2$. After 2 days of in vitro cultivation, the growth medium was completely replaced by fresh medium. After 10–14 days, flasks were mechanically shaken for 60 min, 150 rpm to yield microglia in the supernatant, which were sub-cultured into uncoated well plates according to the experiment. They were kept in 50% astrocyte conditioned medium and 50% fresh DMEM supplemented with 10% FBS, 100 U/mL pen/strep (Table 1). For all experiments, primary microglial cells were used only for the first and second passage.

### In vivo LPS treatment
The transcriptome analysis of the TCA/glycolytic genes of in vivo microglia was performed using our recently published data (Zhang et al., 2022)[21]. In the study of Zhang et al., 2022 mice were given an intraperitoneal (ip) injection of 1 mg/kg LPS (Sigma-Aldrich, Escherichia coli 011:B4,L4391) dissolved in DPBS (Lonza, BE17512F). Control mice received a respective volume of DPBS. After 3 h, animals were sacrificed under anesthesia and the brain was collected. Mouse microglia isolation from mouse were isolated from adult mouse brain using the protocol as described before[66]. Briefly, microglia-enriched percoll fractions were sorted by gating the DAPI[neg]CD11b[high]CD45[int]Ly6c[neg] cells. 200.000-300.000 microglia per brain were sorted for RNAseq experiments.

### Maintenance human induced-Pluripotent Stem Cells
The induced pluripotent stem cell (iPSC) cell line Ctr8.2 (UEF-2B) was obtained from Virtanen Institute for Molecular Sciences-University of Eastern Finland[67]. Another iPSC line purchased from Gibco was also used for the functional assays (Seahorse measurements). The iPSC colonies were cultured and expanded onto Matrigel (Corning, 354277) in 6 well plates and mTeSR8 medium (Gibco, A1516901) supplemented with 100 U/mL penicillin/Streptomycin. Once the cells were confluent, iPSC colonies were passaged every 3-4 days using enzymatic detachment with EDTA 0,5 mM for 5 min and re-plated in mTeSR8 medium

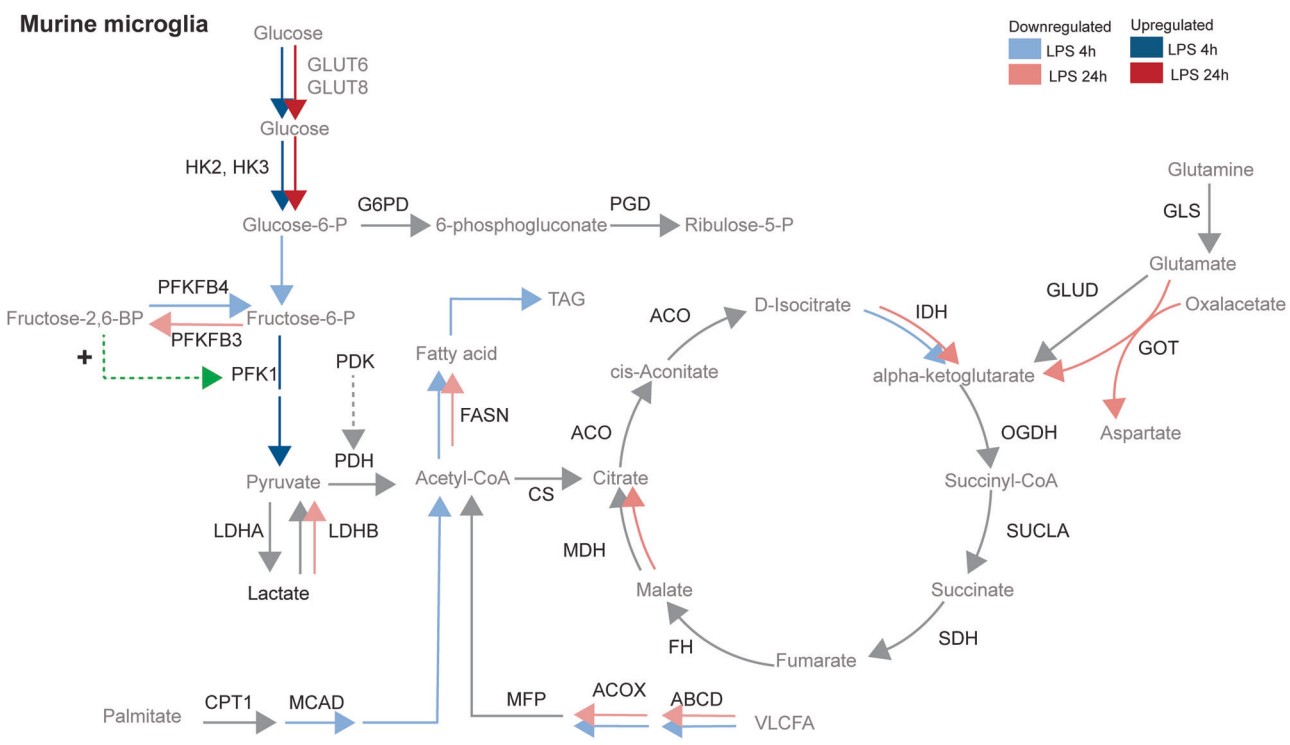

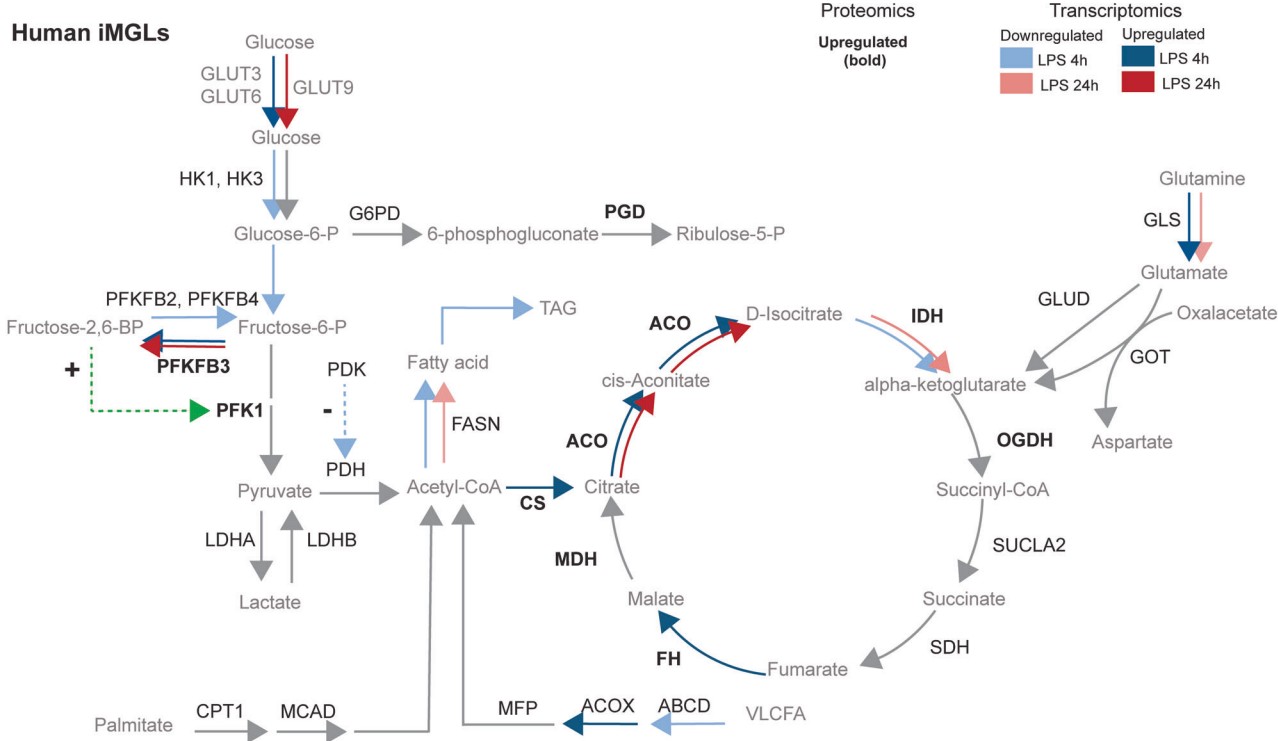

with RevitaCell (Gibco, A2644501). The pluripotency of iPSCs was tested regularly by staining for the pluripotency-marker OCT4. In addition, the cells were examined regularly for the presence of mycoplasma.

### Microglia differentiation protocol from iPSCs
Microglia differentiation from iPSCs was based on modified protocols[23]. Day 0 of differentiation was started when the cells reached 70 to 80% confluency. Cells were detached with accutase (Sigma Aldrich, A6964) and incubated at 37 °C for 5 minutes. Detached cells were transferred to a Falcon tube and centrifuged for 5 min 300rcf. Cells were resuspended in differentiation medium mTers1 supplemented with fresh cytokines with 50 ng/mL BMP-4, 50 ng/mL VEGF, 20 ng/mL SCF (differentiation medium 1) and 10 µM Rock Inhibitor. 10.000–15.000 cells per well were seeded in a 96-well U-bottom ultra-low adherence plate (Corning, 7007) and the plate was

**Fig. 8 | Similarities and divergences in mouse and human microglial metabolism.** Upper panel: Upon LPS treatment mouse microglia upregulate genes that code for glucose transporters GLUT6 and GLUT8, HK2 and HK3 robustly at 4 h and 24 h with a shared decrease of downstream pathway components and downregulation of isocitrate dehydrogenase. A sustained downregulation of genes that encode key fatty acid import mediators, such as ATP-binding cassette transporter subfamily D (ABCD) and peroxisomal acyl-coenzyme A oxidase (ACOX) is concomitant with a downregulation of fatty acid synthesis key enzyme fatty acid synthase (FASN). Lower panel: Upon LPS treatment human iMGL cells robustly upregulate genes that code for glucose transporters GLUT3 and GLUT6 at 4 h, and GLUT9 at 24 h. Human iMGLs only upregulate hexokinases 1 and 3 at 4 h, not at 24 h. Downstream elements of the glycolytic pathway such as PFKFB3 are upregulated. An increase in Fructose-2,6 biphosphate promotes PFK1 activity. In our analyses, the upregulation of GLS only occurred at 4 h in human iMGLs and not in mouse microglia. Within the TCA cycle, citrate synthase (CS) and aconitase (ACO) were upregulated both at 4 h and 24 h, while isocitrate dehydrogenases (IDH1 and IDH2) were downregulated both at 4 h and 24 h. Downregulation of genes that encode fatty acid import and catabolism, such as ABCD and ACOX occurs concomitantly with a sustained downregulation of FASN upon LPS. Grey lines denote unchanged gene expression. Dotted green lines denote promotion of enzyme activity. Dotted blue lines denote negative feedback on enzyme activity. Dark-colored arrows denote upregulation of gene expression at 4 h (blue) and 24 h (red). Light-colored arrows denote downregulation of gene expression at 4 h (blue) and 24 h (red). Upregulation of protein abundance is denoted in bold letters. Data that supports this figure is found in Supplementary material.

centrifuged at 100 rcf for 5 min to ensure clustering of the cells at the bottom of the well. The plate was incubated at 37 °C, 5% $CO_2$.

From day 1 to day 3, 75% of medium was changed using differentiation medium 1. At day 4, all the embryoid bodies (EBs) were collected and medium was removed. The EBs were resuspended in differentiation medium 2 (XVIVO 15 medium + 2 mM Glutamax, 100 U/mL Pen-Strep and 0.055 mM 2-mercaptoethanol with 50 ng/mL SCF, 50 ng/mL M-CSF, 50 ng/mL IL3, 50 ng/mL FLT3 and 5 ng/mL TPO). Approximately 20 EBs were plated per well of a 6-well plate. At day 8, medium was changed by removing all the medium in the wells and adding new differentiation medium 2. At day 11, the old medium was removed and the EBs were resuspended in 20 mL of differentiation medium 3 (XVIVO 15 + 2 mM Glutamax, 100 U/mL Pen-Strep and 0.055 mM 2-mercaptoethanol with 50 ng/mL FLT-3, 50 ng/mL M-CSF and 25 ng/mL of GM-CSF). At day 18, the supernatant was collected and passed through a 37 μm cell strainer and spun down at 300 rcf for 5 min. Microglial progenitors were resuspended in maturation medium (Advanced DMEM/F12 supplemented with 5 μg/mL N-acetylcysteine, 400 μM 1-Thioglycerol, 1 μg/mL heparan sulfate, 1% Glutamax, 1% NEAA, 1% Pen-Strep, 2% B27, and 0.5% N2. Growth factors: 100 ng/mL interleukin-34, 25 ng/mL M-CSF, 25 ng/mL CX3CR1, 25 ng/mL TGF-β-1 and 50 ng/mL CD200) and used for experiments and termed throughout the manuscript as iPSC-derived microglia-like (iMGL) cells.

## LPS stimulation
Mouse microglia and hiMGLs microglia were treated with LPS (250 ng/m) for different time points (as indicated in the figure legends). We have demonstrated that LPS concentrations from 10–1000 ng/mL present similar robust microglial responses in terms of Normalized Cell Index with and without FBS in murine microglia (Supplementary Fig. 1A–C). However, we observed a significant decrease in viability when serum was added to human microglia-like cell cultures (Supplementary Fig. 9).

## IncuCyte phagocytic assay
For the Phagocytic assay, 15,000 cells were seeded in a 96-well flat bottom plate in DMEM (4.5 g/L) or maturation medium, for mouse or iMGL, respectively. The cells were incubated for 24 h at 37 °C and 5% $CO_2$. 4 h before treatment, the plate was placed inside the IncuCyte S3 to obtain a blank measurement of the cells. The cells were treated with Cytochalasin D (Sigma Aldrich, C8273, 10 μM) in microglia medium with pHrodo red *E.coli* bioparticles 1 ug/0.1 ml (Thermo Fischer, p35361) and monitored for a minimum of 24 h. The data was obtained using the IncuCyte S3.

## Immunofluorescent staining
Cells were fixed using paraformaldehyde (PFA) 4% for 20 min and permeabilized using Triton X-100 0.1% for 10 min. Unspecific binding was blocked by incubation with 5% BSA for 1 h at room temperature. Incubation with primary antibody against IBA1, OCT4, was conducted overnight at 4 °C. Secondary anti-rabbit antibodies coupled to Alexa Fluor 488 (Invitrogen, A11034) or 568, (Invitrogen, A10037) were incubated at room temperature. Nuclei were counterstained with DAPI. Images were acquired using a Leica DFC3000 G camera and acquired using Leica Las 4.3 software.

## Real-time cell impedance measurements
Morphological changes in mouse and iMGL cells were measured using a label-free, real-time cell impedance-based system; xCELLigence® RTCA MP system (ACEA BIO). 15,000 cells/well were seeded in a 96-well E-plate (Agilent, 5232368001)[68] containing gold microelectrodes fused to the bottom surface of the well plate[69]. The impedance of the electron flow caused by cell attachment to the well (cell index) was measured every 30 min. The cells were seeded one day before treatment and the measurement of the cell index was performed 24 h following LPS treatment. The cell index was normalized to 1, before the LPS stimulation.

## Mitochondrial respiration measurement
Measurement of oxygen consumption rate (OCR) and extracellular acidification rate (ECAR) was performed using XFe extracellular flux analyzer (Seahorse Bioscience, Billerica, MA) and mitochondrial stress test kit. 50,000–70,000 cells/well of mouse or iMGL microglia cells were seeded in XFe96 cell culture microplate. 4 or 24 h following addition of LPS stimulation, the medium was replaced by Seahorse XF base medium (Agilent, 102353-100) supplemented with 10 mM glucose, 1 mM sodium pyruvate and 2 mM glutamine and incubated for 1 h in $CO_2$ free incubator at 37 °C. For OCR measurements the following substances were injected in order; port A: Oligomycin (2.5 μM), port B: FCCP (0.5 μM) and port C: rotenone (0.5 μM) and antimycin A (0.5 μM). For ECAR measurements were injected in port A: Glucose (17,5 mM), port B: Oligomycin (2, 5 μM) and port C: 2-DG (50 mM). Three baseline OCR measurements (3 min mix, 0 min delay, 3 min measure = 3/0/3) were recorded, followed by assessment of mitochondrial metabolism by injection of oligomycin (3/0/3), FCCP (3/0/3), and a combination of rotenone and antimycin A (3/0/3). Same number of cells are seeded per well. ECAR and OCR data are normalized to live cell number in each assay. The following parameters were deduced: basal respiration (OCR values, used to provide ATP under baseline conditions), ATP-linked respiration (following oligomycin injection, a reduction in OCR values represents the part of basal respiration used to produce ATP), maximal respiration (the maximal OCR values following FCCP injection)[70,71]. Basal glycolysis was calculated from the subtraction between the maximum measurement before oligomycin injection and the last measurement before glucose injection. Glycolytic capacity was calculated by subtracting the maximum measurement after oligomycin injection and the last measurement before glucose injection. Glycolytic reserve was calculated by diving glycolysis capacity by basal glycolysis and it is represented as a percentage. At least 3 independent experiments were performed. Two-way ANOVA test was used to determine statistical significance for the different mitochondrial parameters.

**Table 1 | Key resources table**

| | Brand | Product | Cat No. | Concentration |
|---|---|---|---|---|
| IMGL generation | | | | |
| Medium and reagents | Gibco™ | Essential 8™ Basal Medium *(500 mL)* | A1516901 | NA |
| | Gibco™ | Essential 8™ Supplement (50X) *(10 mL)* | A1517101 | 1x |
| | Stemcell™ | mTeSR™1 Basal Medium *(400 mL)* | #85851 | NA |
| | Stemcell™ | mTeSR™1 Supplement (5X) *(100 mL)* | #85852 | 2.0 mL/mL |
| | Gibco™ | RevitaCell™ Supplement (100X) (5 mL) | A2644501 | 1x |
| | Merck | In-solution Rock Inhibitor (500 µg/296 uL) | Y-27632 | 10 µM |
| | Lonza | XVIVO ™ 15 | BE02-060Q | NA |
| | Thermofisher | 2-mercaptoethanol | 31350010 | 0.055 mM |
| | Gibco™ | GlutaMax | 35050061 | 2 mM 1% |
| | Gibco™ | Penicillin/Streptomycin | 15070063 | 100 mg/mL 100 U/mL |
| | Gibco™ | DPBS 1x | 14190250 | NA |
| | Sigma-Aldrich | Accutase Solution | A6963 | NA |
| | Sigma-Aldrich | EDTA (Ethylenediaminetetraacetic acid disodium salt solution | E7889-100mL | 0,5 mM |
| | Gibco™ | Advanced DMEM/F12 1x Reduced Serum medium (1:1) 500 mL 3.15 g/L | 12634-010 | NA |
| Plastic equipment | Greiner Bio-one | 6 Well culture plate | 657 160 | NA |
| | Corning | 96-well Ultra Low attachment plate U-bottom | 7007 | NA |
| | STEMCELL | Reversible strainer 37µm large | 27250 | NA |
| Maturation medium supplements | Sigma | N-acetylcysteine | A0737 | 5 µg/mL |
| | Sigma | 1-Thioglycerol | M6145 | 400 µM |
| | Sigma | Heparan sulfate | H7640 | 1 µg/mL |
| | Gibco™ | NEAA | 11140-050 | 1% |
| | Gibco™ | B27 | 17504-044 | 2% |
| | Gibco™ | N2 | 17502-048 | 0.5% |
| Growth Factors | Peprotech | Recombinant Human BMP-4 50 µg | 125-05 | 50 ng/mL |
| | Peprotech | Recombinant Human VEGF165 50 µg | 100-20 | 50 ng/mL |
| | Peprotech | Recombinant Human SCF 50 µg | 300-07 | 20 ng/mL |
| | Peprotech | Recombinant Human M-CSF (CSF1) 50 µg | 300-25 | 50 ng/mL 25 ng/mL |
| | Peprotech | Recombinant Human IL-3 50 µg | 200-03 | 50 ng/mL |
| | Peprotech | Recombinant Human FLT3-ligand 50 µg | 300-19 | 50 ng/mL |
| | Peprotech | Recombinant HumanTPO 50 µg | 300-18 | 5 ng/mL |
| | Peprotech | Recombinant Human GM-CSF 50 µg | 300-03 | 25 ng/mL |
| | Peprotech | Recombinant Human Interleukin-34 | 200-34 | 100 ng/mL |
| | Peprotech | Recombinant Human CX3CL1 | 300-31 | 25 ng/mL |
| | Peprotech | Recombinant Human TGF-β1 | 100-21 C | 25 ng/mL |
| | R&D systems | CD200 | 2724-CD-050 | 50 ng/mL |
| Medium and reagents | | | | |
| | Gibco™ | DMEM High Glucose 4.5 g/L | 42430025 | NA |
| | Agilent, | Seahorse XF base medium | 102353-100 | |
| | Hyclone | fetal bovine serum (FBS) | SV30160 | 10% |
| | Thermo Fisher | Sodium Pyruvate | 15070063 | 1 mM |
| | Sigma-Aldrich | Lipopolysaccharide (LPS) *Escherichia coli* | L4391 | 1 mg/kg (In vivo) 250 ng/ml (In vitro) |
| | Lonza | DPBS | BE17512F | |
| | Sigma Aldrich | Cytochalasin D | C8273 | 10uM |
| | Thermo Fischer | pHrodo™ Red E. coli BioParticles™ Conjugate for Phagocytosis | p35361 | 1ug/0.1 ml |
| | Sigma Aldrich | Paraformaldehyde (PFA) | P6148 | 4% |
| | Sigma Aldrich | Tritox X-100 | 11332481001 | 0.1% |
| | Sigma Aldrich | Bovine Serum Albumine (BSA) | A7030 | 5% |
| | Lonza | MycoAlert® PLUS Mycoplasma Detection Kit | LT07-710 | NA |
| | Sigma Aldrich | Fluoroshield™ with DAPI | F6057 | NA |

**Table 1 (continued) | Key resources table**

| | Brand | Product | Cat No. | Concentration |
|---|---|---|---|---|
| | Sigma Aldrich | Glucose | G8270 | 10 mM |
| | Thermo Fischer | Glutamine | 25030024 | 2 mM |
| | Sigma Aldrich | Oligomycin | 04876 | 2.5 uM |
| | Sigma Aldrich | FCCP | C2920 | 0.5 uM |
| | Sigma Aldrich | Rotenone | R8875 | 0.5 uM |
| | Sigma Aldrich | Antimycin A | A8674 | 0.5 uM |
| | Sigma Aldrich | β-Nicotinamide adenine dinucleotide hydrate (NAD) | N7004 | 25 mM |
| | Sigma Aldrich | Lactate dehydrogenase | L2500 | 100 U/mL |
| | BIOKE | NucleoSpin RNA Isolation kit | 740955 | |
| Plastic equipment | | | | |
| | Agilent | 96-well E-plate | 5232368001 | |
| | Agilent Technologies | Extracellular Flux Assay Kit | | |
| | Greiner bio-one | Cellstrar tubes 15 ml | 188285 | |
| Antibodies | | | | |
| | Wako | IBA1, Rabbit Lot: PTR2404 | 019-19741 | (1:500) |
| | Invitrogen | Alexa Fluor 488 Lot: 2256732 | A21206 | (1:2000) |
| | Sigma Aldrich | Fluoroshield™ with DAPI | F6057 | NA |
| *Oligonucleotides* | | | | |
| | Biolegio | NANOG F: CAATGGTGTGACGCAGAAGG R: TGCACCAGGTCTGAGTGTTC | | |
| | Biolegio | NeuN F: ACGATCGTAGAGGGACGGAA R: TTTAGCTTCCAGCCGTTGGT | | |
| | Biolegio | P2RY12 F: TCAGATTACAAGAGCACTCAAGACT R: CTGCAGAGTGGCATCTGGTAT | | |
| | Biolegio | CD11C F: GGGGTGCTGTCTACCTGTTT R: GGGGTGCTGTCTACCTGTTT | | |
| | Biolegio | HPRT F: TCATTATGCTGAGGATTTGGAAA R: GGCCTCCCATCTCCTTCATC | | |
| | Biolegio | TMEM119 F: CAAGGAACTGGTCCTGGGG R: CAGGAGCAGCAACAGAAGGA | | |
| | Biolegio | HBMS F: TGGACCTGGTTGTTCACTCCTT R: CAACAGCATCATGAGGGTTTC | | |

Not applicable is shown as NA.

**Lactate measurement**

Cells were seeded in a density of 15,000 cells/well and treated with LPS 250 ng/ml in DMEM or maturation medium, for mouse or iMGL, respectively. Lactate release was measured after 4–8 h of treatments. Medium from microglia cells was collected and diluted 3 times in demineralized water. A calibration curve of eight lactate standards ranging from 0 to 1.2 mM was prepared for quantification purposes. Subsequently, lactate was measured in a 96-well plate using 20 µl medium sample or lactate standard mixed with 225 µl reaction mixture (0.44 M Glycine/ 0.38 M Hydrazine [pH 9.0], 2.8 mM NAD) and 5 units L-lactic dehydrogenase (EC 1.1.1.27) followed by absorbance determination at 340 nm using the 120 SynergyTM H4. All chemicals were purchased from Merck Millipore. Background absorbance of the blank control (0.0 mM lactate standard) was subtracted from all sample readings and medium samples were corrected for dilution. Medium lactate concentrations were determined based on linear regression of the standard curve and the data are shown as percentage of change from the control.

**ATP measurements**

Cells were seeded in a density of 15,000/well in a 96 well plate in 100 µl of DMEM media of maturation medium, for mouse or iMGL, respectively. After 4 h, 8 h and 24 h of LPS stimulation, 50 µl of detergent and 50 µl of substrate solution were added to the cells for ATP analysis following the manufacture's recommended protocol (AbCam,

ab113849). ATP levels were measured using luminescence in a BioTek Synergy H1 Microplate Reader.

**RNA isolation**

Mouse microglia was treated with LPS (250 ng/m) for 4 or 24 h. The control cells were collected following 24 h media change. Isolation of RNA was performed using the Nucleospin® RNA isolation Kit (Macherey-Nagel). The samples were stored at −80 °C until they were sent for transcriptomic analysis.

**RNA-seq library preparation, quality control and analysis**

The quality of the samples, the construction and the sequencing of the RNA-libraries were performed by GenomeScan Bv. A set of standard quality metrics for the raw data set was determined using third-party (FastQC v0.11.9) and in-house (FastQA v 3.1.25) QC tools. Mouse raw reads were aligned to the GRCm38.p6 genome and human raw reads were aligned to the GRCh38.p13 genome. The reads were mapped to the reference sequence using a short read aligner based on Burrows-Wheeler transform (Tophat v2-2.1) with default settings. Reads were counts with HTSeq (v0.11.0). Only unique reads that fall within exon regions were counted.

Raw count matrices were loaded in R. Lowly expressed genes were filtered using a cut off of 0.5 CPMs. Only genes with >0.5 CPMs in at least 3 samples were included in the analysis. The count matrix was normalized with the blinded variance-stabilizing method from DESeq2

with subsequent differential gene expression analysis[72]. For human microglia RNA-seq, batch effect differences were controlled by including the collection number in the design formula. Several comparisons were made, for all we used an absolute log fold change >1.5 and an FDR-adjusted $p$-value < 0.05 to define a differentially expressed gene (DEG). Gene set enrichment analysis (GSEA) for individual comparisons was performed using Metascape. For visual comparison of the changes of Glycolytic and TCA pathways upon LPS treatments between human and mouse microglia, the Log2 Fold Changes of 1-to-1 ortholog genes were plotted as scatter plots. One to one orthologs between mouse and human were obtained from Biomart (https://www.ensembl.org/). Visualizations were made with the packages 'ggplot2' and 'pheatmap'.

### Proteomics sample preparation

Primary microglia lysates were mixed with LDS loading buffer (NuPAGE) and 10 µg total protein/the total provided samples volume was loaded on the gel. The sample was run (100 V) for about 5 min into a precast 4–12% Bis-Tris gels (Thermo Scientific). The protein band was cut from the gel after staining the proteins with Biosafe Coomassie G-250 stain (Biorad). The gel band was sliced into small pieces and washed for 30 min at RT with 30% and 50% v/v acetonitrile in 100 mM ammonium bicarbonate and 100% acetonitrile for 5 min, before drying the gel pieces in an oven at 37 °C. The cysteines in the proteins were reduced for 30 min at 55 °C with 30 µL 10 mM dithiothreitol in 100 mM ammonium bicarbonate) and alkylated for 30 min at RT (in the dark) with 30 µL 55 mM iodoacetamide in 100 mM ammonium bicarbonate). The gel pieces were washed with 100% acetonitrile for 30 min and dried in an oven at 37 °C. The digestion was performed overnight at 37 °C with 30 µL 16.7 ng/ µL trypsin (sequencing grade modified trypsin, Promega). After the overnight digestion, elution of the peptide was done for 20 min at RT with 50 µL 75% v/v acetonitrile plus 5% v/v formic acid. The eluted fractions were dried in the speedvac and resuspended in 25 µL 0.1% v/v formic acid.

### Targeted proteomics analyses

A triple quadrupole mass spectrometer with a nano-electrospray ion source (TSQ Vantage, Thermo Scientific) coupled to a nano-liquid chromatography system (Ultimate 3000, Dionex) was used for the targeted quantitative measurements for the mitochondrial[73] and glycolytic[74] target panels. All peptides for these panels were concatenated in series of synthetic proteins, which were produced containing $^{13}C$-labeled lysines and arginines (QconCATs, Polyquant). The previously optimized transitions for these isotopically-labeled standard peptides were used for the quantification of the endogenous peptides. The digested peptides were separated prior to the MS measurements by liquid chromatography using a nano column (Acclaim PepMapC100 C18, 75 µm x 50 cm, 2 µm, 100 Å, Thermo Scientific). Samples were concentrated on a trap column (µPrecolumn cartridge, Acclaim PepMap100 C18, 5 µm, 100 Å, 300 µm id, 5 mm Thermo Scientific) after injection of 0.5 µg protein/equivalent of 1 µL starting material and isotopically-labeled standards ranging between between 0.1–1 ng (dependent on the abundance of the protein targets). The injection was done using the µL-pickup system from a from a cooled autosampler (5 °C) with 0.1% v/v formic acid as transport liquid. Peptide separation was obtained in 100 min with a linear gradient from 3–60% v/v acetonitrile plus 0.1% v/v formic acid at a flow rate of 300 nL/min. Detection of the peptides in the mass spectrometer was done with the following settings: the spray voltage was set to 1500 V, the capillary temperature was 270 °C, the instrument was scanning in positive mode with a half maximum peak width of 0.7 for Q1 and Q3 a and a cycle time of 1.2 ms. Fragments were created using a collision gas pressure of 1.2 mTorr and optimal collision energies (CE) were predicted using a linear regressions that was previously optimized for the specific instrument (CE = 0.03*m/z precursor ion +2.905 for 2⁺

precursor ions, and CE = 0.03*m/z precursor ion +2.467 for 3⁺ precursor ions The LC-MS peak assignments were manually curated using the Skyline software[75] and the integrated peak areas were used for quantification via the known concentration of the spiked isotopically labeled standard.

### Discovery-based proteomics analyses

Discovery mass spectrometric analyses applying label free quantification (LFQ) were performed on a quadrupole orbitrap mass spectrometer equipped with a nano-electrospray ion source (Orbitrap Exploris 480, Thermo Scientific). Chromatographic separation of the peptides was performed by liquid chromatography (LC) on an Evosep system (Evosep One, Evosep) using a nano-LC column (EV1137 Performance column 15 cm × 150 µm, 1.5 µm, Evosep; buffer A: 0.1% v/v formic acid, dissolved in milliQ-$H_2O$, buffer B: 0.1% v/v formic acid, dissolved in acetonitrile). The peptide digests were injected in the LC-MS with an equivalent 1 µg total protein starting material and the peptides were separated using the 30SPD workflow (Evosep). The mass spectrometer was operated in positive ion mode and data-independent acquisition mode (DIA) using isolation windows of 16 m/z with a precursor mass range of 400–1000, switching the FAIMS between CV-45V and -60V with three scheduled MS1 scans during each screening of the precursor mass range. LC-MS raw data were processed with Spectronaut (version 16.0220606) (Biognosys) using the standard settings of the directDIA workflow except that quantification was performed on MS1, with a human or mouse SwissProt database (www.uniprot.org, 20350 entries for the human samples and 17021 entries for the mouse samples). Amount of identified proteins for mouse microglia: 2901. Amount of identified proteins for human microglia: 3127.

### Statistical analysis and visualization

Statistical analysis was performed using a paired student's $T$ test or ANOVA and Tukey's test for post hoc multiple comparisons of the parametric data in between-group analyses. Non-parametric data were evaluated using Kruskal–Wallis test. Data were analyzed using GraphPad Prism software (version 9.0, GraphPad Software Inc., La Jolla, CA, USA), expressed as mean ± SD for transcriptomic data and SEM for the rest experiments. DESeq2 analysis used Wald test to determine significance. Graphs were visualized using Excel and GraphPad Prism version 9. $P$ values indicating statistical significance differences between mean values are defined as follows: $^*p < 0.05$, $^{**}p < 0.01$, $^{***}p < 0.001$.

The statistical significance of the overlaps between gene lists is calculated based on the hypergeometric distribution. More specifically, given a total of $N$ genes, if gene sets A and B contain $m$ and $n$ genes, respectively, and $k$ of them are in common, then the $p$-value of enrichment is calculated as follows:

$$p = \sum_{i=k}^{\min(m,n)} \binom{m}{i} \binom{N-m}{n-i} / \binom{N}{n}$$

Let us use Fig. 2A overlap as an example: $N = 11852$, A = 1334 + 406, B = 1344 + 406, $k = 406$. The numerical computation of these $P$ values was performed in R. The enrichment level of the overlap is calculated by R= $kN/(mn)$.

### Reporting summary

Further information on research design is available in the Nature Portfolio Reporting Summary linked to this article.

## Data availability

The datasets generated through (Abud et al., 2017) and re-analyzed for this study are available through GEO Series accession number GSE89189. The datasets generated through (Brownjohn et al., 2018) and re-analyzed for this study are available through GEO Series

accession number GSE110952. The datasets generated through (Zhang et al., 2022) and re-analyzed for this study are available through GEO Series accession number GSE175578. We have generated RNAseq data on human and mouse microglia (iPSC-derived microglia and primary microglia) that are available using the accession number GSE221013. The untargeted proteomics data have been deposited to the ProteomeXchange Consortium via the PRIDE partner repository. The dataset identifier is PXD043836. The targeted mass spectrometry proteomics data have been deposited to the ProteomeXchange Consortium via the Peptide Atlas. The dataset identifier PASS05835. Source data are provided with this paper.

## Code availability

The code that supports the findings of this study are available from the authors upon request.

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

## Acknowledgements

A.M.D. is the recipient of a Rosalind Franklin Fellowship co-funded by the European Union and the University of Groningen. A.M.D. and A.K. are recipients of a Parkinson Fonds grant. A.M.G. is financially supported by the GSMS. We thank for kind technical assistance Renzo Mancuso (VIB-UAntwerp Center for Molecular Neurology, Belgium), Nicola Fattorelli and Anna Martinez-Muriana (KU Leuven, Leuven, Belgium).

## Author contributions

A.M.S.G. and A.M.G: Conceptualization, Methodology, Investigation, Visualization, Writing–Original Draft. M.T.L., A.O., T.C., J.C.W. and J.H.: Methodology, Investigation, Visualization, A.K., B.E., E.B., B.M.B., S.L., J.C.W. and J.K.: Resources, Methodology, Writing–Review & Editing, Supervision. AMD: Conceptualization, Methodology, Resources, Writing–Original Draft, Review & Editing, Supervision, Funding acquisition. All authors have read and agreed to the published version of the manuscript. Several figure items were created with Biorender.com.

## Competing interests

The authors declare no competing interests.
