## [Peer Review File · Nature Communications]

Species-specific metabolic reprogramming in human and mouse microglia during inflammatory pathway inductionREVIEWER COMMENTS

Reviewer #1 (Remarks to the Author):

Review: Species-specific metabolic reprogramming in human and mouse microglia during inflammatory pathway induction by Sabogal-Guaqueta and colleagues.

In this interesting and pertinent manuscript, Sabogal-Guaqueta and colleagues compare the metabolic response of human and mouse iPSC derived microglia using the TLR4 agonist LPS. They find a relation between inflammatory gene induction and glycolytic upregulation with marked species-specific differences. Given the interest in the field in microglial specific immune responses and the link to metabolic reprogramming and immune response regulation, comparative studies such as this are important for understanding translatability within the limitations of our models. Of note, the authors find that while glycolytic activity appears to increase in response to LPS, metabolic reprogramming was characterized by upregulation of hexokinase in mouse microglia and phosphatidyl kinase in human microglia. This may be an important distinction to consider when conducting translational research with these novel systems, but more comprehensive evidence is needed to substantiate these claims as outlined herein. Altogether there are several major and minor concerns that should be addressed by the authors, including comparisons of their data with published datasets, species specific dose responses to LPS, and more robust orthogonal validation of their transcriptomic findings. Overall, the authors finding that glycolytic flex occurs in microglia after inflammatory stimulation is interesting and relevant, however further validation is needed.

Major concerns:

1. This study relies heavily on transcriptomic analysis to characterize the proteins responsible for metabolic reprogramming, but mRNA abundance may not be linearly indicative of protein activity especially in a situation of cellular inflammatory stress. The findings in this paper need to be strengthened by direct interrogation of the protein content (e.g., western blotting to validate the differences in hexokinase versus phosphofructokinase or enzymatic inhibition of key enzymes differing between the species in the Seahorse) to validate these claims and reinforce the link between transcriptomic and metabolic findings.
2. In the TCA cycle section, citrate synthase, isocitrate dehydrogenase, and α -ketoglutarate dehydrogenase are the key enzymes. Why do they show differences in both mouse and human microglia, for instance citrate synthase is decreased in the mouse microglia but increased in the human microglia? α -ketoglutarate dehydrogenase is only altered in the mouse microglia but not in the human microglia, how do authors address these different changes due to the LPS treatment? It would be helpful to validate these statistically significant changes in the TCA cycle described in figure 5 by western.
3. How does the LPS response in the author's findings compare to published transcriptomic datasets on microglia and macrophages? Some specific pertinent datasets include PMID 26224331 (iPSC derived macrophages and primary macrophages with and without 2.5 ng/mL LPS), PMID 30382133 (murine microglia with and without 100ng/mL LPS), and PMID 31375314 (iPSC derived microglia in vitro and in vivo, with and without LPS).
4. The dose response to LPS is markedly different between humans and mice, such that mice have a markedly lower (100-1000x) sensitivity to LPS (PMID 32433924). The dose of LPS utilized by the authors study (250ng/mL) is markedly higher than other published datasets. This therefore leads to the question to what degree the findings of differences in metabolic response are dose dependent. This is compounded by the reported differences between in vitro and in vivo microglial LPS responses (PMID 31375314) finding < 25% overlap in differentially expressed genes in vitro versus in vivo with LPS, admittedly however with differences in dosing/timing. These differences should be addressed experimentally and in the discussion.
5. In addition to effects on metabolism, multiple of the pathways and gene regulated have potential or reported additional effects on inflammation, lipid biosynthesis and metabolism. These effects on both model systems should be assessed. For instance, (line 358-9) PFKFB3 and 4 have differentially regulated expression in human microglia, and have opposite effects on glycolysis with PFKFB3 shunting glucose towards glycolysis, whereas PFKFB4 has more FBPase-2 activity redirecting glucose into the pentose phosphate pathway, and providing reducing power for lipid biosynthesis and scavenging reactive oxygen species. Assays of lipid biosynthesis or metabolism,

or reactive oxygen species at the LPS timepoints discussed would significantly strengthen the findings.

6. Similarly, assays of mitochondrial effects such as on NADH/NAD⁺ ratio would provide validation for transcriptomic findings delineated by the authors. Some differential genes such as *Dld* (lines 236-238) encode mitochondrially localized protein, serving as a NAD⁺ oxidoreductases in the pyruvate dehydrogenase multienzyme complex (Pdhc, Pdc, Pdh). Downregulation of NADH/NAD⁺ ratio with LPS would validate their findings.

7. In figure 6 the authors find that human microglia have an elevated OCR after 24 hours of LPS treatment and decreased after 4 hours of LPS. However, in figure 7 they state that the TCA cycle is increased in iMGLs at all timepoints. Measurements of ATP production or other external validation would confirm the Seahorse findings, and discussion of how these disparate findings are unified in their models is needed.

8. L402-408: This study uses lactate determination with an enzymatic lactate assay at a single timepoint (8 hr) to assess the changes in gene expression levels. However, static metabolic measurements do not reliably align with the activity of metabolic pathways. While detailed flux analysis is outside the scope of this study, having measurements of lactate concentration at more timepoints, especially those used in gene expression analysis, could provide stronger assessment of whether gene expression changes were translated into functional data. This may also be helpful by explicitly linking these results to the ECAR data as well. Additionally, cell survival at LPS dosing and timepoints should be assessed.

9. Overlap in the differentially expressed genes between the 4- and 24-hour timepoints and the pathways involved should be presented.

10. A more robust discussion of in vitro versus in vivo findings in murine microglia are needed. Multiple genes in the glycolytic pathway (lines 193-194) differ between environmental conditions. Minor concerns:

1. A header should be added to figures 2 A,B and 2 D,E to delineate which pathway is investigated.

2. Line 247 should read "human microglia" not just microglia.

3. Line 370, please define gene names.

4. The significance of cellular impedance changes in supplementary figure 5 is not well described.

5. L180-182: Do the authors postulate any significance to the downregulation of *Gapdh* in mouse microglia with the 4h of LPS challenge given its key role in glycolysis?

6. Fig 4, E-F: What is the significance of mouse microglia having significantly higher lactate measurements (Ctr:0.3 mM → LPS 0.4 mM) than human microglia (Ctr: 0.02 mM → LPS: 0.06mM).

7. Are ECAR and OCR in Fig 6 normalized to live cell number or ug of protein? Individual data points should be shown in figure 7?

8. Fig 7: The metabolic pathway for human iMGLs also shows robust upregulation of the *GLS* gene at LPS 4h unlike murine microglia, but I do not recall this being discussed. Can this impact TCA dysregulation? Could this have any confounding effects on analysis of metabolic differences with four hours of LPS given glutamine's role as an important alternate energy source and its supplementation in Seahorse base media?

9. Fig 4/Fig 5: Inconsistent capitalization of gene names between mouse and human bar graphs

Reviewer #2 (Remarks to the Author):

The manuscript by Sabogal-Guaqueta, Marmolejo-Garza and colleagues investigates the metabolic reprogramming in mouse primary microglia as well as iPSC-derived microglia (iMLG) in basal conditions after LPS stimulation. The topic is certainly interesting and relevant as microglial metabolic reprogramming has been suggested to be linked responsive states to disease or damage. However, except for the last figure, this manuscript does not explore metabolic alterations in microglia, but uses transcriptomic changes as a proxy for potential metabolic changes. Although I agree this could be a very valid starting, hypothesis generating point, the absence of any confirmation using metabolic analyses makes the conclusions of this work a bit shallow or incomplete. In addition to that, there are a number of other aspects that I believe can be

improved:

- 1) Figure 1 is not novel. Transcriptomic analysis of mouse primary microglia in response to LPS has been done multiple times before. I suggest transferring this to supplemental data, and begin the manuscript with the more exciting data on iMGL
- 2) The authors use neonatal primary microglia for their analysis. How does this relate to adult microglia? I think experiments should be also performed in adult and old mice, as this can be very relevant for age related diseases
- 3) What is the evidence of transcriptome as a proxy for metabolic state? In many cases there is even a poor correlation of RNA sequencing with proteomic data, so relying only on transcriptomics seems insufficient.
- 4) What do the authors mean with "substantial collection-dependent effect on the transcriptomes"? This is not clear
- 5) A key point when assessing differences between species is that both microglia are not cultures in the same conditions. Given that the media (specially serum) can have critical effects on microglia, I do not think the data should be compared.
- 6) How many iPSC lines were used in this study? And how many biological replicates? It seems from the plot it is only one line, n=3 replicates. This does not seem enough, unless orthogonal experiments (e.g., metabolomics) are performed.
- 7) Along the text, many genes are highlighted, but it is unclear to this reviewer why is the reason behind the selection of some genes but not others. Perhaps it could be more relevant to explore pathways first, and then discuss the genes differentially expressed across conditions in relevant biological processes?
- 8) Figure 5 is very interesting. However, as said before, the fact that both in vitro systems use completely different media compositions probably introduce too many sources of variation and bias the results. I suggest this is repeated using the same culturing conditions for both species.

Reviewer #3 (Remarks to the Author):

The manuscript by Sabogal-Guaqueta and colleagues compares the metabolic reprogramming of human and mouse microglial cells upon inflammatory stimuli. The topic is of high interest for the importance of microglial roles in many inflammatory conditions, and the comparison among human- and mice-derived cells is fundamental for the translatability of data.

Importantly, the authors identified different species-specific mechanisms of LPS-induced reprogramming and described different kinetics of gene modulation. They demonstrated that: 1) LPS promotes an enhanced glycolytic metabolism in both species, but this is the result of increased expression of hexokinase in mice and phosphofructokinase in humans; 2) LPS induces a downregulation of oxidative metabolism in both species but associated to a differential time profile; 3) LPS promotes glycolysis coupled to mitochondrial dysfunction only in mouse microglia not in human iMGL.

Some limits of this work are acknowledged by the authors, and also by this reviewer.

Major points

- 1) Comparison of human iMGL with cultured mouse microglia is not straightforward. It would be necessary to compare the effects of LPS on the same kind of cells, i.e. iMGL from humans and mice. In absence of these experiments, the novelty of the results is mitigated by previous demonstration of transcriptional differences among human and rodent microglia (Smith and Dragunow, *Trends in Neurosciences* 2014; Galato et al., *Nat. Neurosci.* 2017; Gosselin et al., *Science* 2017; Burns et al., *Eur. J. Pharmacol.* 2015 and others).
- 2) The authors should consider that human and mouse responsiveness to LPS could differ for a different expression of TLR4 in the two species (Vaure and Liu, *Front. Immunol.* 2014; Parajuli et al., *J. Neuroinflamm.* 2019; Carpentier et al., *BBI* 2008; Jack et al., *J. Immunol* 2005; Casula et al., 2011).
- 3) Limits of iPSC-derived microglia must be also considered in transcriptomics: transcriptomic and phenotypic deficiencies due to the lack of interaction with other brain parenchyma cells (Abud et al., *Neuron* 2017: 3-dimensional brain organoid (BORG) co-cultured with iPSC-derived microglia; Hasselmann et al., *Neuron* 2019: xenotransplantation approaches).

- 4) Fig.5: the authors should discuss the results on aconitase, which are opposite for human and mouse, and at least mention why FH data are only shown for humans.
- 5) Authors should limit the use of "interesting" while presenting their data (used several times in the text), and/or elaborate the implication of the data.
- 6) Fig.6: the authors should better discuss the Seahorse results showing that OCR is different between humans and mice while ECAR is similar.

Minor points:

- 1) Line 258: "human iMGC cells resembles the adult or fetal human brain regarding their transcriptome." How do the authors reach this conclusion? Do they compare TMEM119, P2Y12 receptor and IBA I at protein levels (immunofluorescence) or they compare their PCR analysis with literature data? It is not clear.
Have they also analyzed microglia from a functional point of view in addition to phagocytosis, i.e. patrolling, migration or others?
- 2) Line 459 Cytochalasin D is not a phagocytosis inhibitor.
- 3) In a number of experiments the number of repeats seems too low considering sample variability (i.e. some transcriptomic results in figure 3 and 4)
- 4) The authors state that "Studies of human brain microglia have been performed on isolated microglia from fresh post mortem samples from potentially neuropathologically affected individuals, which might be hindered by a high interindividual variation." However, this limitation also applies to human iMGL cells, as in Fig, 3C, and should be acknowledged.
- 5) Two different animal Protocols are provided, in two paragraphs named "animals". Which one is used? If both, differences should be mentioned
- 6) Line 250-252: "Microglial progenitors were further differentiated with IL-34, TGF- β , GM-CSF, CD200, and CX3CR1 to assure a full maturation profile." CX3CL1 should substitute CX3CR1

Response to Reviewers (*Responses in Italics*)

Thank you very much for the time and for reviewing our manuscript NCOMMS-23-04080. We thank the editor and the reviewers for their valuable comments to improve our manuscript.

The comments of the reviewers were adequate and brought up important aspects on our project, which were supported by additional experiments. We are listing these points and our responses below.

Reviewer #1

In this interesting and pertinent manuscript, Sabogal-Guaqueta and colleagues compare the metabolic response of human and mouse iPSC derived microglia using the TLR4 agonist LPS. They find a relation between inflammatory gene induction and glycolytic upregulation with marked species-specific differences. Given the interest in the field in microglial specific immune responses and the link to metabolic reprogramming and immune response regulation, comparative studies such as this are important for understanding translatability within the limitations of our models. Of note, the authors find that while glycolytic activity appears to increase in response to LPS, metabolic reprogramming was characterized by upregulation of hexokinase in mouse microglia and phosphatidyl kinase in human microglia. This may be an important distinction to consider when conducting translational research with these novel systems, but more comprehensive evidence is needed to substantiate these claims as outlined herein. Altogether there are several major and minor concerns that should be addressed by the authors, including comparisons of their data with published datasets, species specific dose responses to LPS, and more robust orthogonal validation of their transcriptomic findings. Overall, the authors finding that glycolytic flux occurs in microglia after inflammatory stimulation is interesting and relevant, however further validation is needed.

1. This study relies heavily on transcriptomic analysis to characterize the proteins responsible for metabolic reprogramming, but mRNA abundance may not be linearly indicative of protein activity especially in a situation of cellular inflammatory stress. The findings in this paper need to be strengthened by direct interrogation of the protein content (e.g., western blotting to validate the differences in hexokinase versus phosphofruktokinase or enzymatic inhibition of key enzymes differing between the species in the seahorse) to validate these claims and reinforce the link between transcriptomic and metabolic findings.

*We thank the reviewer for this relevant concern. We agree with the reviewer that mRNA and protein abundance may not correlate linearly. For that reason, the findings in this paper have been paralleled by functional assays, that reflect an increase of glycolysis in both species. To address the reviewer's concern regarding the correlation between genes and protein expression, we performed two mass spectrometry proteomic methods: i) label-free (termed LFQ) and ii) targeted TCA/glycolysis proteomics. In addition, we provided the **proteomic signatures of these enzymes both in human and mouse microglia**. We performed LC-MS proteomic analysis instead of semi-quantitative methods such as Western blotting, as the WB technique is limited by antibody-specificity and cross-species reactivity. This information is included in new figures (Figure 5 and supplementary Figure 6) and the text can be found in the results section under the subsection **Glucose metabolism is differentially altered in mouse and human microglia challenged with LPS**:*

“To further investigate the effects of LPS on mouse microglia and human microglia-like cells at the protein levels, we performed label-free (LFQ) and targeted proteomics by mass spectrometry. Next, we compared the transcriptomic data with the proteomic data, focusing on the key enzymes of glycolysis: Hexokinases (HKs), Phosphofruktokinases (PFKs), 6-phosphofrukt-2-

kinase/fructose-2,6-biphosphatases (PFKFBs) and Phosphoenolpyruvate kinases (PKs). HKs, PFKs and PKs are classically regarded as the “rate-limiting” steps of glycolysis. PFKFBs are bifunctional enzymes that regulate glycolysis by modulating the levels of fructose 2,6 biphosphate (F2,6BP). PFKFB3, among the PFKFB isoenzymes, has a higher kinase activity compared to phosphatase activity, resulting in increased concentrations of F2,6BP. Notably, F2,6BP serves as an allosteric activator of PFKs.

We assessed the species-specific transcriptomic and proteomic signatures of these enzymes and observed that Hexokinase isoform gene and protein abundance was species-specific: *Hk3* transcripts were more abundant than other isoforms in mouse microglia (Fig 5A), while *HK1* and *HK2* were more abundant than *HK3* transcripts in human microglia-like cells. (Fig 5B). LFQ proteomic data demonstrated that HK2 was the most abundant isoform in mouse microglia (Fig 5C), while HK1 was the most abundant isoform in human microglia-like cells (Fig 5D). These differences were recapitulated by targeted proteomic analysis for mouse (Fig 5E) and human (Fig 5F) cells. It is worth noting that upon LPS treatment, mouse cells upregulated *Hk2* and *Hk3* transcripts (Fig 5A) but proteomic differences were not striking (Figs 5C, 5E). Human cells downregulated *HK1* transcripts in response to the LPS challenge, however (Fig 5B), this was not captured by the proteomic analysis (Figs 5D, and 5F).

At the transcriptional level, it was clear that both mouse and human cells exhibited *Pfkl* and *PFKL* transcripts, respectively, as the most abundant isoforms (Fig 5G and 5H). Similarly, we observed that phosphofructokinase liver (PFKL) was the most abundant isoform of the three forms of PFKs both in mice and in human microglia at the gene and protein level (Fig 5I and Fig 5J). In mouse microglia, PFKs gene and protein abundance did not significantly change following LPS stimulation (Figs 5I, 5K). In human microglia, LPS stimulation increased the abundance of PFKL protein (Fig 5J and 5L) but not the PFKL transcripts (Fig 5H). Additionally, we observed an increased abundance of the isoforms PFKM and PFKP by our targeted approach in human but not mouse microglia (Fig 5L). We observed transcriptomic changes of the phosphoenolpyruvate kinase isoform M (Pkm) in human microglia at 24 hours after LPS stimulation (Supplementary Table 6).

The allosteric activator of PFK is fructose-2,6-bisphosphate (Fructose 2,6BP), which is a product of PFKFB3. Conversely, PFKFB2 and PFKFB4 catalyze the degradation of Fructose 2,6BP to Fructose 6P. *Pfkfb4* and *Pfkfb2* were downregulated upon LPS at 4 and 24 hours in mouse microglia (Fig 5M). *PFKFB3* was upregulated upon LPS at 4 and 24 hours in human microglia while *PFKFB2* and *PFKFB4* were slightly decreased (Fig 5N). Based on this analysis, we observed that the mouse transcripts, such as *Hk2*, *Hk3* and *Pfcp* increased in inflammatory conditions, while proteins did not exhibit striking alterations upon LPS challenge, as evidenced by the targeted proteomic analysis (Fig 5O). The human microglial cells exhibited a clearer effect on PFKs, and PFKBs (particularly PFKB3) in conditions of LPS challenge both at the transcript and protein levels (Fig 5 J,L,P).

Taken together, it is evident that the metabolic switch and the increase in glycolytic activity in microglia challenged with LPS is conserved across species. However, the enzymes that drive this upregulation of glycolysis are modulated in a species-specific manner. Moreover, the baseline abundance of glycolytic enzymes, such as Hexokinases are different across species.”

Figure 5. Transcriptomic and proteomic characterization of key glycolytic enzymes in mouse microglia and human microglia-like cells. Mouse and human microglial cells were *in vitro* stimulated with LPS and collected 4 hours after stimulation for proteomic analysis by LC-MS/MS. A,B. Normalized counts depicting transcript abundance of Hk1, Hk2, and Hk3 (A) and HK1, HK2, HK3 (B). C,D. Label-free quantitation values from proteomic analysis depicting relative abundance of HK1, HK2, and HK2 in mouse (C) and human (D) cells. E,F. Proteomic abundances of hexokinases measured by LC-MS/MS with internal standards. I. Depicts plots of absolute and relative abundances on mouse microglia and (F) depicts human cells. G,H. Normalized counts depicting transcript abundance of Pfk1, Pfkp, and PfkM (G) and PFKL, PFKP, and PFKM (H). I,J. Label-free quantitation values depicting relative abundance of PFKL, PFKP, and PFKM in mouse (I) and human (J) cells. K,L. Proteomic abundances of phosphofructokinases measured by LC-

MS/MS with internal standards. (K) Depicts plots of relative abundances on mouse microglia and (L) depicts human cells. (M,N) Normalized counts depicting transcript abundance of Pfkfb1, Pfkfb2, Pfkfb3, and Pfkfb4 (M) and PFKFB2, PFKFB3 and PFKFB4 (N). O,P. Proteomic abundances of PFKFBs measured by LC-MS/MS with internal standards. (O) Depicts relative abundances on mouse microglia and (P) depicts human cells. For all panels, error bars represent SEM. Every biological replicate is depicted as a dot. For mouse RNAseq n=3, for human RNAseq n=4, for mouse proteomics n=4 and for human proteomics n=5. N indicates biological independent experiments. For RNAseq data p values were determined by the Wald test and multi testing corrected in DESEQ2. For proteomic data, p values were determined by unpaired two-tailed t-test. *p<0.05, **p<0.01, and ***p<0.001

By adding these data to our manuscript, we have shown that Hexokinase isoform abundance in microglia is species-specific, PFKFB involvement is species-specific at the transcriptomic and proteomic level. Based on these new findings we include the following consideration in our discussion section:

“Most transcriptomic changes in glycolysis occurred in hexokinase and phosphofructokinase in mouse microglia, while transcriptomic and proteomic changes occurred in the PFKs and PFKFBs in human iMGLs. The direct interrogation of mRNA and protein changes is a topic that remains to be further investigated as a plethora of mechanisms which are species-specific govern transcription, translation, and protein homeostasis.”

2. In the TCA cycle section, citrate synthase, isocitrate dehydrogenase, and α -ketoglutarate dehydrogenase are the key enzymes. Why do they show differences in both mouse and human microglia, for instance citrate synthase is decreased in the mouse microglia but increased in the human microglia? α -ketoglutarate dehydrogenase is only altered in the mouse microglia but not in the human microglia, how do authors address these different changes due to the LPS treatment? It would be helpful to validate these statistically significant changes in the TCA cycle described in figure 5 by western.

We thank the reviewer for their relevant insight on the expression of key TCA cycle enzymes. To address this important point, we have performed LC-MS proteomic analysis, revisited our transcriptomic data, and interpreted it in light of the proteomic analysis. This data is included in the manuscript as Supplementary Figure 6.

All in all, these results strongly suggest that a downregulation of gene transcription may not be paralleled by a decreased abundance of protein. This suggests that in order to observe a downregulation in such a short period of time (i.e. 4 hours), protein should not only decrease its transcription/translation, but increase degradation parallelly. Conversely, inducible changes such as upregulation of FH in human but not mouse microglia could be recapitulated with our targeted proteomic approach, strengthening our transcriptomic findings on LPS responses across species.

This issue is also addressed in the discussion section following remarks of the previous comment of the reviewer.

The following text was added in the manuscript in the TCA cycle subsection *Main enzymes of the TCA cycle are dysregulated in mouse and human microglia after LPS challenge* :

“Next, we compared the transcriptomic data with the proteomic data, focusing on the key enzymes of the TCA cycle. It was observed that transcripts of *Idh1* and *IDH1* were the most abundant isoforms in mouse (Fig S6A) and human cells (Fig S6B), respectively. With the proteomic analysis, we could determine that *IDH2* is the most abundant isoform in mouse (Fig S6C, Fig S6D) microglia, while both *IDH1* and *IDH2* have comparable abundances in human microglia. The transcriptomic profile of the transcripts of TCA cycle enzymes showed a significant downregulation of *Idh1* for mouse (Fig S6E) and upregulation of *ACO1*, *FH*, *CS* transcripts with downregulation of *IDH1* and *IDH2*. in human microglia (Fig S6F). Overall, we observed no significant differences in the TCA cycle enzymes after LPS stimulation by label-free proteomic

quantitation (Fig S6G, and Fig S6H). However, our targeted TCA proteomics approach yielded in discrete upregulations that did not reach statistical significance in mouse microglia (Fig S6I), but yielded significant upregulation of ACO2, IDH2, and MDH2 (Fig S6J) in human iMGLs.

All in all, these results strongly suggest that TCA cycle activity may be enhanced in human cells, but not in mouse cells. Both species exhibit downregulation of *Idhs* and *IDHs* at the transcriptional level. However, proteomic differences become evident for upregulations of TCA cycle enzymes at the 4hr LPS-treatment timepoint (Fig S6)."

Supplementary Figure 6. Transcriptomic and proteomic characterization of TCA cycle enzymes in mouse microglia and human iMGLs. Mouse and human microglial cells were in vitro stimulated with LPS 250ng/mL and collected 4 hours after stimulation for proteomic analysis by LC-MS/MS.

A,B. Normalized counts depicting transcript abundance of genes that code for Isocitrate dehydrogenase enzymes in mouse (A) and human (B) cells. C,D. Label-free quantitation values from proteomic analysis depicting relative abundance of IDH1, IDH2, and IDH3A and IDH3B in mouse (C) and human (D) cells. E,F. Normalized counts depicting transcript abundance of genes that code for TCA cycle enzymes in mouse (E) and human (F) cells. G,H. Label-free quantitation values depicting relative abundance of TCA cycle enzymes in mouse (G) and human (H) cells. I,J. Proteomic abundances of TCA cycle enzymes measured by LC-MS/MS with internal standards. (I) Depicts plots of relative abundances on mouse microglia and (J) depicts human cells. For all panels, data are presented as mean \pm S.E.M.. Every biological replicate is depicted as a dot. For mouse RNAseq n=3, for human RNAseq n=4, for mouse proteomics n=4 and for human proteomics n=5. n indicates biological replicates. For RNAseq data p values were determined by

the Wald test and multi testing corrected in DESEQ2. For proteomic data, p values were determined by unpaired two-tailed t-test. * $p < 0.05$, ** $p < 0.01$, and *** $p < 0.001$

3. How does the LPS response in the author's findings compare to published transcriptomic datasets on microglia and macrophages? Some specific pertinent datasets include PMID 26224331 (iPSC derived macrophages and primary macrophages with and without 2.5 ng/mL LPS), PMID 30382133 (murine microglia with and without 100ng/mL LPS), and PMID 31375314 (iPSC derived microglia in vitro and in vivo, with and without LPS).

To address the reviewer's concern, we have compared our gene set enrichment analysis with the ones published by Alasso and colleagues from the Gaffney lab¹ observing similar enrichment in inflammatory pathways such as TNF-signaling, and NF- κ B signaling pathway, response to interferon-gamma among others, concluding that our transcriptomic responses are comparable. The study of Hasselman and colleagues² challenged iPSC-derived microglia with LPS and observed a robust microglial activation, observing enrichment for similar processes. Still, to have a clearer comparison across these relevant studies, we performed a meta-analysis by comparing the lists of DEGs between LPS-treated and control samples from these studies and ours. These data showcase the strong similarities between published microglial models upon LPS treatment.

*We included these data as a supplementary figure 4, describing it in our results subsection **LPS challenge of human microglia induces an inflammatory gene signature**:*

"In order to further validate the LPS-transcriptomic signature of human microglia-like cells, we performed a meta-analysis of previously published transcriptomes of LPS-treated human microglial cells. Although strikingly different in length (Fig S4A and Fig S4B), the functional overlap of biological processes that these lists enrich for is shared for most of the detected biological processes (Fig S4C). The top dysregulated processes that the enrichment analysis detected comprised "inflammatory response", "cell activation", regulation of inflammatory response", "innate immune response", "cytokine signaling in immune system", among others (Fig S6D). Similarly, by examining the top 100 dysregulated processes (Fig S4E) we observed a major overlap in enrichment between these DEG lists. These data showcase the reproducibility of microglial-like cell differentiation from iPSC and their response to LPS."

Supplementary Figure 4. Comparison of the transcriptomes of LPS-treated iPSC-derived human microglia-like cells from this study with other iPSC-derived microglia models.

A. Venn diagram with the overlap of DEGs of short-term LPS-treated iPSC-derived microglia-like cells from the study of Alasoo and colleagues, Hasselman and this study. B. Circos plot depicting the overlap in the lists of DEGs. On the outside, each arc represents the identity of each gene list, using the following color code: Blue, Hasselman; red, Alasoo; green, this study. On the inside, each arc represents a gene list, where each gene member of that list is assigned a spot on the arc. Dark orange color represents the genes that are shared by multiple lists and light orange color represents genes that are unique to that gene list. Purple lines link the same gene that are shared by multiple gene lists. C. Circos plot built in the same way as (B). Blue lines link the genes that, although different, fall under the same ontology term (the term has to be statistically significantly enriched and with size no larger than 100). Blue lines indicate

the amount of functional overlap among the input gene lists. D. Heatmap depicting the top 20 statistically enriched terms (GO/KEGG, canonical pathways, etc..) hierarchically clustered into a tree based on Kappa-statistical similarities among their gene memberships. The term with the best p-value within each cluster is shown as its representative term in the dendrogram. Heatmap cells are colored by their p-values, while cells indicate the lack of enrichment for that term in the corresponding gene list. E. Heatmap depicting the top100 statistically enriched terms in a similar fashion to (D).

These published studies did not assess functional outcomes related to metabolism to further delve into differences or similarities between macrophages and microglia. In order to support the transcriptomic dysregulations of glycolysis and TCA cycle from our study, we have surveyed the glycolytic and TCA genes from the differential expression analyses that the studies of Alasoo and Hasselmann have made available. We have added the following data as supplementary tables. These data are presented in the results subsections for Glycolysis and TCA and is discussed in the discussion section.

The following text and table were added in relation to these points:

Subsection Glucose metabolism is differentially altered in mouse and human microglia challenged with LPS:

*“To further study the transcriptomic dysregulations elicited by LPS in our human microglia-like cells, we surveyed for key genes of interest for glycolysis in our differential expression data as well as in published data from LPS-stimulated iMGLs from Alasoo¹ and Hasselmann². Regarding glycolysis, we observed overlap on significant upregulation of *PFKFB3* across models with discrepant changes on genes that code for hexokinases or phosphofructokinases (Supplementary Table 9). “*

Subsection Main enzymes of the TCA cycle are dysregulated in mouse and human microglia after LPS challenge:

*“By comparing our data with the data generated by Hasselmann² and Alasoo¹, within the TCA cycle gene list, we observed shared downregulation of *IDH1* and *IDH2*, with overlapping upregulations of *ACO1*, *ACO2* and unique upregulations of *CS* and *DLST* (Supplementary Table 10). All in all, these data point towards the LPS-dependent transcriptomic changes of metabolic pathways in various iPSC-derived microglial models.”*

Supplementary Table 9. Comparison of key glycolytic enzyme changes upon LPS-administration in other iPSC-derived microglia-like cells

Gene	Alasoo and colleagues		Hasselmann and colleagues		This study	
	LPS LOG2FC	LPS p adjusted	LPS LOG2FC	LPS p adjusted	LPS LOG2FC	LPS p adjusted
HK1	-0,033945571	0,869757095	0,311869655	0,715240884	-0,449345293	0,000778876
HK2	1,725753088	4,97E-08	0,16974605	0,860490848	0,155577255	0,603020607
HK3	-0,537599882	0,083773302	1,209588858	0,069627517	-0,441352212	0,035084305
PFKL	0,225884443	0,315874198	-0,489771107	0,241800574	0,032290815	0,852953539
PFKM	-0,240184491	0,538907676	0,083985022	0,938397175	-0,64519715	0,065800871
PFKP	0,832655278	0,000560821	0,443239198	0,467758675	0,218589369	0,313727481
PFKFB2	-2,480182269	1,49E-14	-0,172979321	0,805604011	-0,544419049	0,006348326
PFKFB3	3,78475598	2,32E-41	4,205201541	1,89231E-10	0,679523678	0,002267515
PFKFB4	0,662009829	0,240592732	1,936316186	0,004624704	-0,908830721	0,038238826

Supplementary Table 10. Comparison of key TCA cycle enzyme changes upon LPS-administration in other iPSC-derived microglia-like cells

Gene	Alasso and colleagues		Hasselmann and colleagues		This study	
	LPS LOG2FC	LPS p adjusted	LPS LOG2FC	LPS p adjusted	LPS LOG2FC	LPS p adjusted
ACO1	0,421255237	0,025235254	-0,36264084	0,542526946	0,697360145	0,000652458
ACO2	-0,197243328	0,42950605	-0,652235174	0,102594347	0,083759295	0,544608075
CS	0,015316572	0,929757664	-0,570141527	0,092493501	0,291576022	0,047077513
DLST	0,359539115	0,00167459	0,211144398	0,738888198	0,133740006	0,250510975
FH	0,008833245	0,971012849	-0,688039184	0,199720543	0,422484283	0,000382666
IDH1	-1,266030386	2,29E-05	0,00116655	0,998041115	-0,378472565	0,000393245
IDH2	-0,827082955	0,004406717	-2,069535342	3,40E-14	-0,34908629	0,02583341
MDH1	-0,376936268	0,205612346	0,591075912	0,560827374	0,026806508	0,901072063
MDH2	-0,071247416	0,742852071	0,154950073	0,825393354	0,003924168	0,983967589
OGDH	-0,053030142	0,848309813	-0,891771874	0,281191754	0,056926869	0,768809426
SUCLG1	-0,370416189	0,063284257	-0,180815122	0,799951978	-0,03890903	0,871030102
SUCLA2	-0,193489157	0,159857482	-0,732743216	0,054113095	0,038148576	0,901142661

Regarding mouse microglia, *the study of Pulido-Salgado and colleagues³ that the reviewer suggests observed robust changes such as enrichment of TLR receptor signaling, NF-kappa B signaling, and TNF-signaling in primary microglia (reported by the authors). We have already compared two mouse models in our manuscript and observed similar transcriptomic disturbances upon LPS-stimulation (See Figures 1 and 2). Although, we decided to not add more data regarding this, we include more text in our discussion section.*

4. The dose response to LPS is markedly different between humans and mice, such that mice have a markedly lower (100-1000x) sensitivity to LPS (PMID 32433924). The dose of LPS utilized by the authors study (250ng/mL) is markedly higher than other published datasets. This therefore leads to the question to what degree the findings of differences in metabolic response are dose dependent. This is compounded by the reported differences between in vitro and in vivo microglial LPS responses (PMID 31375314) finding < 25% overlap in differentially expressed genes in vitro versus in vivo with LPS, admittedly however with differences in dosing/timing. These differences should be addressed experimentally and in the discussion.

The review by Lasselin and colleagues⁴ mentioned by the reviewer, illustrates that in vivo LPS administration in humans (intravenous) and mice (intraperitoneal) follows comparable temporal evolution, being TNF, and IL-6 main readouts in the 2-6 hours following administration. A discrepancy on IL-1b measurements is explained by the authors due to low assay sensitivity. However, this publication raises further questions such as the arbitrary working doses for LPS administration (0.4-2.0 ng/kg body weight in humans and 100-800 ug/kg body weight in mice). It is important to mention that the dosing in mice had the main objective of inducing sickness behavior and that intraperitoneal administration yields lower plasma concentrations (~10 times) of LPS than intravenous. Intraperitoneal administration is slower than intravenous. This discrepancy in dosing does not strictly mean that mice have lower sensitivity. If anything, it highlights that we lack proper translational approaches to study LPS responses across species. We have decided to include these considerations in the discussion section as follows:

“Current literature on LPS responses in humans and mice suggests that LPS sensitivity is different across species. One may come to this conclusion due to the range of working concentrations and dosing between humans and mice. However, several considerations must be considered: i) *In vivo* LPS administration in humans (intravenous) and mice (intraperitoneal) follows comparable temporal evolution, being TNF, and IL-6 main readouts in the 2-6 hours following administration. However, ii) available literature raises further questions such as the arbitrary working doses for LPS administration (0.4-2.0 ng/kg body weight in humans and 100-800 ug/kg body weight in mice). It is important to mention that iii) the dosing in mice had the main objective of inducing sickness behavior and that iv) intraperitoneal administration yields lower plasma concentrations (~10 times) of LPS than intravenous. v) Intraperitoneal administration is slower than intravenous. This discrepancy in dosing does not strictly mean that mice have lower sensitivity. If anything, it highlights that we lack proper translational approaches to study LPS responses across species. In our study, we have employed the same LPS concentration to stimulate human and mouse microglial *in vitro* models as a starting point to objectively assess LPS-sensitivity. However, further characterization of TLR4 signaling and epigenomic landscapes that modulate transcriptional responses between species remain to be investigated.”

Following up on the reviewer’s comment, we have performed xCELLigence real-time impedance measurements with various LPS concentrations. We have demonstrated that in our cultures, LPS concentrations higher than 125 ng/mL elicit robust microglial responses in terms of Normalized Cell Index, which indicates changes in shape/morphology of cells. We therefore decided to seed and treat mouse and human microglia with the same conditions and assess metabolic outcomes such as oxygen consumption, extracellular acidification rate, lactate production, and ATP measurements.

LPS responsiveness cannot be experimentally determined in a straightforward fashion, as readouts such as Normalized Cell Index may not be comparable across species (microglial shape is different between mouse and human cells). In this manuscript, we propose that the first experiments to be performed using the same culturing conditions, namely seeding density, media volume and LPS concentration. Any species-specific changes in LPS response would then be a result of an interaction between intrinsic species-specific LPS-sensitivity, and downstream TLR4 signaling.

*To provide additional information on factors that may contribute to species-specific sensitivity to TLR4 agonism with LPS, we have investigated the transcript abundance of the *Tlr4* gene in mouse and human cultures by plotting the transcripts per million reads (TPMs), a normalized measure of gene expression across samples. The approach of comparing TPMs across species is a widely-used method to determine species-specific transcriptomes⁵. By considering these data, it is reasonable to think that human microglia may express more TLR4 receptor and that TLR4 agonism may be enhanced in human compared to mouse microglia. However, we observed that TLR4 agonism induces modest responses in human microglia compared to mouse microglia. In the present study, we propose that metabolic shifts drive these downstream TLR4 agonism. We have decided to include these valuable data in our manuscript.*

*The study of Hasselmann and colleagues demonstrated that *in vitro* and *in vivo* treated xenotransplanted microglia overlap in a minority of genes while the majority of genes that are differentially expressed are within the *in vitro* DEGs. To assess these possible differences between *in vitro* and *in vivo* microglia, we performed a similar approach between our *in vitro* and *in vivo* data from Zhang et al⁶. The *in vivo* data contained 1750 DEGs, while our *in vitro* data contained 1741 DEGs, of which 406 were shared across both conditions. The study of Hasselmann focused on human microglia transplanted and engrafted into the mouse brain and stimulated with intraperitoneal LPS. We included this information in a Venn diagram and integrated in Figure 2 and added text in the discussion.*

Figure 2. Transcriptomic changes in LPS-treated primary mouse microglia and *in vivo* LPS-treated microglia.

A. Overlap of DEGs *in vitro* primary neonatal microglia at 4 hours after LPS stimulation and *in vivo* treated adult microglia 3 hours after intraperitoneal LPS injection.

B. Circos plots depicting the overlap in the lists of DEGs. On the outside, each arc represents the identity of each gene list, using the following color code: Blue, *In vitro* primary neonatal microglia; red, *in vivo*-treated adult microglia. On the inside, each arc represents a gene list, where each gene member of that list is assigned a spot on the arc. Dark orange color represents the genes that are shared by multiple lists and light orange color represents genes that are unique to that gene list. Purple lines link the same gene that are shared by multiple gene lists. For the plot in the right, blue lines link the genes that, although different, fall under the same ontology term (the term has to be statistically significantly enriched and with size no larger than 100). Blue lines indicate the amount of functional overlap among the input gene lists.

C Heatmap depicting the top 20 statistically enriched terms (GO/KEGG, canonical pathways, etc.) hierarchically clustered into a tree based on Kappa-statistical similarities among their gene memberships. The term with the best p-value within each cluster is shown as its representative term in the dendrogram. Heatmap cells are colored by their p-values, while cells indicate the lack of enrichment for that term in the corresponding gene list.

D,E. Hierarchically clustered heatmaps depicting gene expression changes in glycolytic pathways in *in vitro* cultivated mouse microglia (D) and in isolated microglia from LPS-treated mice (depicted as *in vivo*) (E). Color key corresponds to row Z-score.

Following up on the reviewer's comment, we have also performed xCELLigence real-time impedance measurements with various LPS concentrations, and demonstrated that in our cultures, LPS concentrations from 10-1000ng/mL show similar robust microglial responses in terms of Normalized Cell Index with and without FBS in murine microglia (Supplementary Figure 1). For this reason, we decided to keep with the concentration 250ng/ml reported in previous studies in our group^{7,8}. We therefore seeded and treated mouse and human microglia under the same conditions and further assess metabolic outcomes such as oxygen consumption, extracellular acidification rate, lactate production, and ATP measurements.

Supplementary Figure 1. LPS response to mouse microglia. A, B. Mouse cells were seeded in 96-well E-plates at a density of 15,000 cells/well and monitored by a real-time impedance-based xCELLigence system. Mouse microglia

or iMGL cells were challenged with increasing concentrations of LPS (10-1000ng/ml) in the absence (A) or presence (B) of serum for 24hr. C. IBA1 immunostaining of control and LPS-treated mouse microglia with and without FBS. D. Heatmap depicting the top 100 most significant differentially expressed genes of 4h LPS treatment compared with the control. Color key corresponds to row Z-score. E. Volcano plot depicting fold changes and $-\text{Log}_2$ of the adjusted p value per gene comparing responses against LPS 24h and 4h.

5. In addition to effects on metabolism, multiple of the pathways and gene regulated have potential or reported additional effects on inflammation, lipid biosynthesis and metabolism. These effects on both model systems should be assessed. For instance, (line 358-9) PFKFB3 and 4 have differentially regulated expression in human microglia and have opposite effects on glycolysis with PFKFB3 shunting glucose towards glycolysis, whereas PFKFB4 has more FBPase-2 activity redirecting glucose into the pentose phosphate pathway, and providing reducing power for lipid biosynthesis and scavenging reactive oxygen species. Assays of lipid biosynthesis or metabolism, or reactive oxygen species at the LPS timepoints discussed would significantly strengthen the findings.

We have performed additional experiments on targeted proteomic analysis for pentose phosphate pathway (PPP) enzymes following LPS stimulation and observed a significant increase in Phosphogluconate dehydrogenase (PGD) in human microglia but not in mouse microglia. These data strongly suggest that glucose is shunted towards PPP in human microglia. We have included these data as Supplementary Figure 7.

Results descriptions are placed in results subsection **Glucose metabolism is differentially altered in mouse and human microglia challenged with LPS.**

“Furthermore, we performed targeted proteomic analysis for pentose-phosphate pathway (PPP) enzymes following LPS stimulation and observed a significant increase in Phosphogluconate dehydrogenase (PGD) in human microglia but not in mouse microglia (Fig S7). These data strongly suggest that glucose is shunted towards PPP in human microglia.

Supplementary Figure 7. Proteomic characterization of pentose-phosphate pathway enzymes in mouse microglia and human iMGLs. A, B. Absolute proteomic abundances of pentose phosphate pathway (PPP) enzymes measured by LC-MS/MS with internal standards. Mouse microglia are depicted in (A) and mouse iMGLs are depicted in (B). C, D. Relative proteomic abundances of (A) and (B) respectively. All data are presented as mean \pm S.E.M. p values were determined by paired two-tailed t-test. * $p < 0.05$

6. Similarly, assays of mitochondrial effects such as on NADH/NAD⁺ ratio would provide validation for transcriptomic findings delineated by the authors. Some differential genes such as *Dld* (lines 236-238) encode mitochondrially localized protein, serving as a NAD⁺ oxidoreductases in the pyruvate dehydrogenase multienzyme complex (*Pdhc*, *Pdc*, *Pdh*). Downregulation of NADH/NAD⁺ ratio with LPS would validate their findings.

We agree with the reviewer that most dysregulated enzymes that we have presented have dehydrogenase activity and may factor into the NADH/NAD⁺ ratio. However, the electron transport chain has strong dehydrogenase activity and factors into the NADH/NAD⁺ ratio as well. Following the recommendation of the reviewer, we performed NADH and NAD⁺ measurements of LPS treated human and mouse microglia. We observed significant changes upon LPS treatment in mouse microglia only at 4 hours (Revision Figure 1A), but not at 24 hours (Rev Figure 1B). We believe that this data is preliminary and should be interpreted carefully due to the pool of NAD⁺ producing and consuming mechanisms in microglia. Linking such findings directly to an increase of glycolysis may be insufficient to demonstrate these increases. These data are only shared with the reviewer.

Revision Figure 1. NAD⁺/NADH ratio measurements in mouse microglia and human microglia-like cells. A, B. Changes in NAD⁺/NADH ratio measurements in mouse microglia at 4 hours (A) and 8 hours (B) of stimulation with LPS C, D. Changes in NAD⁺/NADH ratio measurements in human microglia-like cells at 4 hours (C) and 8 hours (D) of stimulation with LPS. n=2-3. **p<0.001 by t-test.

7. In figure 6 the authors find that human microglia have an elevated OCR after 24 hours of LPS treatment and decrease after 4 hours of LPS. However, in figure 7 they state that the TCA cycle is increased in iMGLs at all timepoints. Measurements of ATP production or other external validation would confirm the Seahorse findings, and discussion of how these disparate findings are unified in their models is needed.

We agree with the reviewer, measurements of ATP production may correlate with the Oxygen consumption findings upon LPS stimulation. Therefore, we performed additional experiments on lactate and ATP measurements. These data were added to Figure 6, along with the oxygen consumption ratio and extracellular acidification rate data:

“Human microglia showed significance increase in basal respiration and ATP-linked OCR following 24h LPS treatment compared to the control (Fig 7F-G). The basal respiration and maximal OCR was decreased after 4h LPS treatment in human microglia (Fig 7F,H), while in the mouse microglia it was reduced (Fig 7C,E). We then measured ATP concentrations in LPS-treated microglia and observed that ATP levels were decreased at all timepoints for mouse

microglia (Fig 7I) and for human iMGLs (Fig 7J). On the other hand, the extracellular acidification rate (ECAR), an index of glycolysis feeding into lactate, was significantly increased after 24h LPS stimulation in both species (Fig 7K-L). Basal glycolysis and glycolytic capacity were increased after 24h LPS stimulation in mouse (Fig 7M-N) and human microglia (Fig 7P-Q), however, the glycolytic reserve was increased just in mouse microglia (Fig 7O,R).

To further characterize the glycolytic metabolism of these cells, we performed lactate determinations of the supernatant of LPS-treated cells observing significant increases of lactate at all timepoints for mouse microglia (Fig 7S) and for human iMGLs (Fig 7T). This indicates that the capability of microglia to respond to an energetic demand, as well as, how close the glycolytic function is to their maximum capacity depends on the species. Moreover, we measured the non-glycolytic acidification, and we did not observe differences between mouse and human microglia (Supplementary Fig 10). These data indicate that LPS stimulation affects mitochondrial respiration in both mouse and human cells, and the overall response in basal and glycolytic capacity is similar between the two species. Importantly, the metabolic reprogramming towards an increased glycolytic activity was demonstrated in both mouse and human microglia.”

Figure 7. Mitochondrial respiration analysis of mouse and human microglia. Representative experiment showing OCR (oxygen consumption rate) in A. mouse and B human microglial cells treated with LPS for 4 and 24h. Measurements were obtained by an extracellular flux analyzer (Seahorse Bioscience). C-E. Bar graph showing different mitochondrial respiration parameters represented as mean \pm SEM of the OCR values in mouse and F-H in human microglia. I. ATP measurements of mouse and J microglia treated with LPS for 4, 8 and 24h expressed as a percentage of the untreated control for the same timepoint. Data from individual experiments have the same color. K. Representative experiment showing ECAR (Extracellular acidification rate) in K. mouse and L. human microglia cells treated with LPS for 4 and 24h. M-R. Bar graph showing different mitochondrial respiration parameters represented as mean \pm SEM of the ECAR values in mouse and P-R in human. S. Lactate measurements of the supernatants of mouse

microglia treated with LPS for 4, 8 and 24h expressed as a percentage of the untreated control for the same timepoint in mouse and T in human microglia. Data from individual experiments have the same color. Data are shown as mean \pm SEM, n = 3–6 technical replicates. All experiments were repeated at least three times for each cell line (Ctr 8.2 & Gibco). *p*-values indicating statistically significant differences between the mean values are defined as follows: **p* < 0.05, ***p* < 0.01 (compared with ctr), #*p* < 0.05, ##*p* < 0.01 (LPS 4h vs LPS 24h). Data for lactate and ATP measurements are showing 4-5 different experiments where each color point represented an experiment.

8. L402-408: This study uses lactate determination with an enzymatic lactate assay at a single timepoint (8 hr) to assess the changes in gene expression levels. However, static metabolic measurements do not reliably align with the activity of metabolic pathways. While detailed flux analysis is outside the scope of this study, having measurements of lactate concentration at more timepoints, especially those used in gene expression analysis, could provide stronger assessment of whether gene expression changes were translated into functional data. This may also be helpful by explicitly linking these results to the ECAR data as well. Additionally, cell survival at LPS dosing and time points should be assessed.

We thank the reviewer for their suggestions about lactate measurements across time. We agree that metabolic pathways are dynamic processes, and although fluxomics may not be required for the present study, we can parallel our transcriptomic, proteomic and metabolic data with lactate measurements. Following the reviewer's suggestions, we profiled lactate production in human and mouse microglia at 4, 8, and 24 hours.

Consistent with our data, we observe an increase in lactate concentrations in microglia supernatants, for both species. We have added this data, along with the ATP measurements to the extracellular acidification data of Figure 7. (See comment above). To address the last sentence of this comment, we want to refer to our Supplementary Figure 1 (from comment 1), where we assessed cell viability within an LPS range in our cells.

9. Overlap in the differentially expressed genes between the 4- and 24-hour timepoints and the pathways involved should be presented.

To address this comment, we included in figure S3 more information, and we have profiled the transcriptomic responses across time and species. In figure S3C and S3G, we graphically represented with circos plots the overlaps between the lists of DEGs (purple lines) and the overlap with genes that enrich for the same biological processes (blue lines). These overlaps were more numerous in mouse microglia compared to human microglia. In figures S3D and S3H we compared the relationships between the magnitudes of Fold Changes upon LPS 4 and 24 hours observing a slope of 0.93 in the mouse microglia and 0.77 in the human.

To further this comment from the reviewer, we have performed Venn diagrams to determine which DEGs overlap between 4 and 24 hour time points in mouse and human microglia observing that 402 genes are shared and ~670 genes are unique per condition. Furthermore, we performed GSEA and observed that the 402 genes that are shared across both conditions enrich for processes such as innate immune response, response to lipopolysaccharide, and interferon signaling. It is worth noting that the shared enrichment for these pathways is already noted in the main figure 1.

*These new insights complement the findings in Figure S3, and we incorporated them in this supplementary figure and incorporated this text results subsection **Temporal transcriptomic responses to LPS are different in mouse and human models:***

“We analyzed the overlap of DEGs between 4- and 24-hour time points in mouse and human microglia observing that in mouse microglia, 402 genes are shared and ~670 genes are unique per condition. Furthermore, we performed GSEA and observed that the 402 (Supplementary

Figure 2A) genes that are shared across both conditions enrich for processes such as innate immune response, response to lipopolysaccharide, and interferon signaling (Supplementary Figure S2B).

Similarly, we performed the same approach to the human microglia-like cells observing a distinct pattern than murine microglia. We found that the number of DEGs at 4 hours LPS was 481 and for 24 hours was 124, sharing 59 genes (Supplementary Figure S2E). When we input the 59 overlapping genes in Metascape, we obtained enrichment of processes such as leukocyte migration, inflammatory response, and mononuclear cell migration among others (Supplementary Figure S2F)

We find that these data support our conclusions on how the human microglia-like cells have a strong TLR4 activation at 4 hours but decreased its enrichment at 24 hours. However, we did not see such an effect in mouse microglia, where TLR4 signaling maintained the transcriptomic enrichment of inflammatory pathways across timepoints.”

Supplementary Figure 3. Cross-species comparison of transcriptomic responses against LPS over time. A, E. Overlap of DEGs in mouse microglia (A) and human iMGLs (E) at 4 and 24 hours after LPS stimulation. B, F. Top enriched biological processes after performing GSEA with the metascape tool with the overlapped gene lists for mouse microglia (B) and human iMGLs (F). C, G. Circos plot depicting how genes from the lists of DEGs overlap for mouse microglia (C) and human iMGLs (G) for the two timepoints. On the outside, each arc represents the identity of each gene list. On the inside, each arch represents a gene list, where each gene member of that list is assigned a spot on the arc. Dark orange color represents the genes that are shared by multiple lists and light orange color represents genes that are unique to that gene list. Purple lines link the same gene that are shared by multiple gene lists (notice a gene that appears in two gene lists will be mapped once onto each gene list, therefore, the two positions are purple linked). Blue lines link the genes, although different, fall under the same ontology term (the term has to be statistically significantly enriched and with size no larger than 100). The greater the number of purple links and the longer the dark orange arcs imply greater overlap among the input gene lists. Blue links indicate the amount of functional overlap among the input gene lists. D, H. Scatterplots of the genes that were differentially regulated by LPS 4h and 24h

treatment (red dots) with an adjusted $p < 0.05$ in mouse **microglia (D)** and **human iMGLs (H)**. A linear model was fitted among the genes that were differentially regulated and the R squared, and the equation of the line are depicted within the scatterplot.

10. A more robust discussion of in vitro versus in vivo findings in murine microglia are needed. Multiple genes in the glycolytic pathway (lines 193-194) differ between environmental conditions.

We agree with the reviewer that our findings require deeper discussion regarding the nature of our experiments as it is easy to misinterpret and mislabel in vitro culture systems. Microglial primary cultures can be performed in many ways and we would like to make a distinction between ex vivo cultured microglia and in vitro cultured microglia. Taking these into consideration, we have decided to add the following text and table to the discussion section:

“In the studies of Gosselin and colleagues^{5,9}, primary microglia was cultured *ex vivo* by obtaining microglia enriched fractions by percoll gradient from mice that were 8-9 weeks of age. Subsequently, these fractions were purified by cell sorting and selecting for live/DAPI- CD11b+ CD45Low. These microglia were maintained in culture for 6h, 24h or 7 days with DMEM/F12 supplemented with 5% FBS. The aim of their studies was to assess how environmental disruptions modify the transcriptome of macrophage populations.

In the current study, primary microglial protocol makes use of pups of 1-3 days of age. Minced brain tissue was dissociated for 25 mins with trypsin and DNase and plated in several flasks. Following 10-14 days of in vitro co-cultivation with astrocytes, flasks were mechanically shaken for 60 mins, 150rpm to yield microglia released in the supernatant which were subcultured into uncoated plates with conditioned medium from astrocytes.

The study of Bohlen¹⁰ and colleagues showcases the importance of astrocytic support to define microglial phenotypes in vitro. In light of the nature of the acute *ex vivo* cultures, where the astrocytic support is lost, the strength of our approach is that we employ astrocyte-derived conditioned media that supports microglial growth and homeostatic phenotype. It is crucial to note that the study of Bohlen demonstrated that primary cultured microglia with their protocol do not show hallmarks of overt inflammation and it was comparable to freshly isolated microglia. Similarly, LPS- transcriptomic responses of in vitro-cultured microglia were comparable between serum-supplemented and serum-deprived cultures as long as the cues from astrocyte conditioned medium are supplied¹⁰.”

Supplementary Table 11. Studies that employed primary microglia and their experimental considerations

Study	Microglia culture type	From adult mice or pups?	Was FBS included?
Gosselin et al, 2014 ⁹	Acutely isolated ex vivo , percoll gradient with FACS purification CD11b+ CD45Low	8-9 weeks	Yes
Nike et al., 2012 ¹¹	Acutely isolated ex vivo , percoll gradient	Young (1-2 month old) and Aged (14-16 month old)	Yes
Gosselin et al., 2017 ⁵	Acutely isolated ex vivo , percoll gradient with FACS purification CD11b+	8-9 weeks	Yes

	CD45Low		
Geric et al., 2019 ¹²	Brain dissociation and plating with posterior subculturing by mechanical shaking.	Post-natal day 0-1	Yes
Bohlen et al., 2017 ¹⁰	Acutely isolated ex vivo , myelin depletion and with MACS purification for CD11b+	3-5 weeks	Yes
Chhor et al., 2013 ¹³	Brain dissociation and plating with posterior subculturing by mechanical shaking.	Post-natal day 0-1	Yes
Dolga et al., 2012 ⁷	Brain dissociation and plating with posterior subculturing by mechanical shaking.	Post-natal day 1-3	Yes
This study	Brain dissociation and plating with posterior subculturing by mechanical shaking.	Post-natal day 1-3	Yes

Minor concerns:

1. A header should be added to figures 2 A,B and 2 D,E to delineate which pathway is investigated.

Headers for Glycolysis and TCA cycle have been added to this figure.

2. Line 247 should read “human microglia” not just microglia.

We modified this sentence.

3. Line 370, please define gene names.

We defined the genes.

4. The significance of cellular impedance changes in supplementary figure 5 is not well described. *We explained better the interpretation of the changes seen by the cellular impedance in subsection **Differential response to LPS through time in mouse and human microglia:***

“To investigate and compare the potential metabolic reprogramming response of mouse and iMGLs to an immune stimulus, we challenged these cells with LPS (250ng/ml) and monitored their morphological changes using the xCELLigence system. This system allows continuous monitoring of alterations in cell morphology by real-time cell impedance measurements which are expressed as normalized cell indexes.”

5. L180-182: Do the authors postulate any significance to the downregulation of Gapdh in mouse microglia with the 4h of LPS challenge given its key role in glycolysis?

We addressed this point in the discussion.

6. Fig 4, E-F: What is the significance of mouse microglia having significantly higher lactate measurements (Ctr:0.3 mM → LPS 0.4 mM) than human microglia (Ctr: 0.02 mM → LPS: 0.06mM).

We assume that the serum presence in the culture medium of mouse microglia can facilitate an increase in the response to the LPS. Since, human iPSC-differentiated microglia are not coping well with the serum in the culture medium, we think that the lactate levels are related to the response to the LPS. Overall, we see a decrease in the glycolytic response in the human cells, although glycolysis occurs in both species. Due to reviewer comments, we did new lactate measurements for 2 more time points, 4 and 24 hours, together with the measurements at 8 hours, now the differences are represented in percentage of change compared with internal control (Figure 7Q, S)

7. Are ECAR and OCR in Fig 6 normalized to live cell number or ug of protein? Individual data points should be shown in figure 7?

In these experiments, same number of cells were seeded per well. ECAR and OCR data are normalized to ug of protein at the end of the assay. Individual data-points are shown in figure 7.

8. Fig 7: The metabolic pathway for human IMGLs also shows robust upregulation of the GLS gene at LPS 4h unlike murine microglia, but I do not recall this being discussed. Can this impact TCA dysregulation? Could this have any confounding effects on analysis of metabolic differences with four hours of LPS given glutamine's role as an important alternate energy source and its supplementation in Seahorse base media?

In our analyses, the upregulation of GLS only occurred at 4 hours in human iMGLs and not in mouse microglia. More elegant experiments such as isotopically-labeled carbon tracing will provide the answers to these questions. With the current data, we believe it is wise to report that species-specific Gls and GLS modulation occurs with TLR4 agonism. We have added one sentence in the figure legend of figure 8 to point this out.

"In our analyses, the upregulation of GLS only occurred at 4 hours in human iMGLs and not in mouse microglia (Supplementary tables 1-6)"

9. Fig 4/Fig 5: Inconsistent capitalization of gene names between mouse and human bar graphs
This was revised and corrected.

Reviewer #2 (Remarks to the Author):

The manuscript by Sabogal-Guaqueta, Marmolejo-Garza and colleagues investigates the metabolic reprogramming in mouse primary microglia as well as iPSC-derived microglia (iMLG) in basal conditions after LPS stimulation. The topic is certainly interesting and relevant as microglial metabolic reprogramming has been suggested to be linked responsive states to disease or damage. However, except for the last figure, this manuscript does not explore metabolic alterations in microglia, but uses transcriptomic changes as a proxy for potential metabolic changes. Although I agree this could be a very valid starting, hypothesis generating point, the absence of any confirmation using metabolic analyses makes the conclusions of this work a bit shallow or incomplete. In addition to that, there are a number of other aspects that I believe can be improved:

1) Figure 1 is not novel. Transcriptomic analysis of mouse primary microglia in response to LPS has been done multiple times before. I suggest transferring this to supplemental data, and begin the manuscript with the more exciting data on iMGL

We partially agree with the reviewer. Human and mouse innate immune responses to LPS have been profiled previously without studying temporal aspects of their responses. In this study, we have profiled the transcriptomic responses of LPS after 4 and 24 hours in murine microglia (Main Figure 1) and human (Main Figure 3) and we demonstrate that the top dysregulated genes belong to inflammatory pathways. However, when we performed gene set enrichment analysis (GSEA) of downregulated genes, we observed metabolic processes such as metabolism of lipids, carbohydrate metabolic processes, autophagy among others. These data set the stage to investigate further metabolic signatures in murine microglia and later profile them in human microglia. All in all, transcriptomic responses over time have not been profiled in microglia, this is why we included in Figure S3 a comparison of the DEGs between 4 and 24 hours, similarly a comparison with adult microglia was included in Figure 2. To further characterize these temporal discrepancies in mouse and human models, we have added these data to Figure 2 A-C. We feel that Figure 1 shows data that are instrumental to the understanding of the manuscript, should be visible and therefore need to be in the main text.

Figure 2. Transcriptomic changes in LPS-treated primary mouse microglia and *in vivo* microglia.

A. Overlap of DEGs *in vitro* primary neonatal microglia at 4 hours after LPS stimulation and *in vivo* treated adult microglia 3 hours after intraperitoneal LPS injection.

B. Circos plots depicting the overlap in the lists of DEGs. On the outside, each arc represents the identity of each gene list, using the following color code: Blue, *In vitro* primary neonatal microglia; red, *in vivo*-treated adult microglia. On the inside, each arc represents a gene list, where each gene member of that list is assigned a spot on the arc. Dark orange color represents the genes that are shared by multiple lists and light orange color represents genes that are unique to that gene list. Purple lines link the same gene that are shared by multiple gene lists. For the plot in the right, blue lines link the genes that, although different, fall under the same ontology term (the term has to be statistically significantly enriched and with size no larger than 100). Blue lines indicate the amount of functional overlap among the input gene lists.

C. Heatmap depicting the top 20 statistically enriched terms (GO/KEGG, canonical pathways, etc..) hierarchically clustered into a tree based on Kappa-statistical similarities among their gene memberships. The term with the best p-value within each cluster is shown as its representative term in the dendrogram. Heatmap cells are colored by their p-values, while cells indicate the lack of enrichment for that term in the corresponding gene list.

D,E. Hierarchically clustered heatmaps depicting gene expression changes in glycolytic pathways in *in vitro* cultivated mouse microglia (D) and in isolated microglia from LPS-treated mice (depicted as *in vivo*) (E). Color key corresponds to row Z-score.

2) The authors use neonatal primary microglia for their analysis. How does this relate to adult microglia? I think experiments should be also performed in adult and old mice, as this can be very relevant for age related diseases

We thank the reviewer for pointing out the relevance of microglia in age-related diseases. Accordingly, it has been proposed that with age there is an increase in microglial markers in the brain such as CD68 and HLA-DR. Extensive work from the Eggen¹⁴ and the Boddeke¹⁵ labs have profiled the transcriptomes of aged microglia. The study from Galatro and colleagues¹⁴ reported that the overlap in the aging microglia transcriptomic signatures between mouse and human is limited and strongly suggests that is different. Whether this age-dependent transcriptomic signature would affect LPS responsiveness in old mice or old humans remains to be investigated and falls out of the scope of the present study.

*In the present manuscript, we compared primary neonatal microglia data with the acutely isolated adult microglia from Zhang and colleagues⁶ in Figure 2. We performed a four-way plot to observe how these microglia recapitulate the glycolytic and TCA geneset disturbances (Figure 2F and 2I). We concluded that primary neonatal and primary adult microglia react similarly to LPS stimulation regarding transcripts from the glycolytic and TCA pathways. To address the reviewer's comment, we compared the *in vivo* and *in vitro* datasets further. The *in vivo* adult dataset contained 1742 DEGs, while our *in vitro* neonatal dataset contained 1751 DEGs, of which 406 were shared across both conditions. (See figure 2 from previous comment)*

3) What is the evidence of transcriptome as a proxy for metabolic state? In many cases there is even a poor correlation of RNA sequencing with proteomic data, so relying only on transcriptomics seems insufficient.

It has been reported that transcriptome and metabolome correlate in certain conditions. In our study we have paralleled transcriptomic signatures and metabolic signatures. However, we agree with the reviewer that this sort of evidence requires further strengthening. The nature of the transcriptomic and proteomic analyses is fundamentally different, sensitivity and specificity are not comparable. Therefore, when comparing them side by side, one should be careful with their interpretation.

To address the reviewer's concern regarding RNA correlation with protein expression, we performed additional experiments and used two proteomic analysis mass spectrometry methods: i) label-free (termed LFQ) and ii) targeted proteomics for glycolysis, and TCA. We performed LC-MS proteomic analysis instead of semi-quantitative methods such as Western blotting, as the WB technique is limited by antibody-specificity and cross-species reactivity. This information is

included in new figures (Figure 5 and supplementary Figure 6) and the text can be found in the results section in subsection **Glucose metabolism is differentially altered in mouse and human microglia challenged with LPS**:

“To further investigate the effects of LPS on mouse microglia and human microglia-like cells at the protein levels, we performed label-free (LFQ) and targeted proteomics by mass spectrometry. Next, we compared the transcriptomic data with the proteomic data, focusing on the key enzymes of glycolysis: Hexokinases (HKs), Phosphofructokinases (PFKs), 6-phosphofructo-2-kinase/fructose-2,6-biphosphatases (PFKFBs) and Phosphoenolpyruvate kinases (PKs). HKs, PFKs and PKs are classically regarded as the “rate-limiting” steps of glycolysis. PFKFBs are bifunctional enzymes that regulate glycolysis by modulating the levels of fructose 2,6 biphosphate (F2,6BP). PFKFB3, among the PFKFB isoenzymes, has a higher kinase activity compared to phosphatase activity, resulting in increased concentrations of F2,6BP. Notably, F2,6BP serves as an allosteric activator of PFKs.

We assessed the species-specific transcriptomic and proteomic signatures of these enzymes and observed that Hexokinase isoform gene and protein abundance was species-specific: *Hk3* transcripts were more abundant than other isoforms in mouse microglia (Fig 5A), while *HK1* and *HK2* were more abundant than *HK3* transcripts in human microglia-like cells. (Fig 5B). LFQ proteomic data demonstrated that HK2 was the most abundant isoform in mouse microglia (Fig 5C), while HK1 was the most abundant isoform in human microglia-like cells (Fig 5D). These differences were recapitulated by targeted proteomic analysis for mouse (Fig 5E) and human (Fig 5F) cells. It is worth noting that upon LPS treatment, mouse cells upregulated *Hk2* and *Hk3* transcripts (Fig 5A) but proteomic differences were not striking (Figs 5C, 5E). Human cells downregulated *HK1* transcripts in response to the LPS challenge, however (Fig 5B), this was not captured by the proteomic analysis (Figs 5D, and 5F).

At the transcriptional level, it was clear that both mouse and human cells exhibited *Pfkl* and *PFKL* transcripts, respectively, as the most abundant isoforms (Fig 5G and 5H). Similarly, we observed that phosphofructokinase liver (PFKL) was the most abundant isoform of the three forms of PFKs both in mice and in human microglia at the gene and protein level (Fig 5I and Fig 5J). In mouse microglia, PFKs gene and protein abundance did not significantly change following LPS stimulation (Figs 5I, 5K). In human microglia, LPS stimulation increased the abundance of PFKL protein (Fig 5J and 5L) but not the PFKL transcripts (Fig 5H). Additionally, we observed an increased abundance of the isoforms PFKM and PFKP by our targeted approach in human but not mouse microglia (Fig 5L). We observed transcriptomic changes of the **phosphoenolpyruvate kinase isoform M (Pkm)** in human microglia at 24 hours after LPS stimulation (Supplementary Table 6).

The allosteric activator of PFK is fructose-2,6-bisphosphate (Fructose 2,6BP), which is a product of PFKFB3. Conversely, PFKFB2 and PFKFB4 catalyze the degradation of Fructose 2,6BP to Fructose 6P. *Pfkfb4* and *Pfkfb2* were downregulated upon LPS at 4 and 24 hours in mouse microglia (Fig 5M). *PFKFB3* was upregulated upon LPS at 4 and 24 hours in human microglia while *PFKFB2* and *PFKFB4* were slightly decreased (Fig 5N). Based on this analysis, we observed that the mouse transcripts, such as *Hk2*, *Hk3* and *Pfcp* increased in inflammatory conditions, while proteins did not exhibit striking alterations upon LPS challenge, as evidenced by the targeted proteomic analysis (Fig 5O). The human microglial cells exhibited a clearer effect on PFKs, and PFKBs (particularly PFKB3) in conditions of LPS challenge both at the transcript and protein levels (Fig 5J,L,P).

Taken together, it is evident that the metabolic switch and the increase in glycolytic activity in microglia challenged with LPS is conserved across species. However, the enzymes that drive this upregulation of glycolysis are modulated in a species-specific manner. Moreover, the baseline abundance of glycolytic enzymes, such as Hexokinases are different across species.”

Figure 5. Transcriptomic and proteomic characterization of key glycolytic enzymes in mouse microglia and human microglia-like cells. Mouse and human microglial cells were *in vitro* stimulated with LPS 250ng/mL and collected 4hr after stimulation for proteomic analysis by LC-MS/MS. A,B. Normalized counts depicting transcript abundance of Hk1, Hk2, and Hk3 (A) and HK1, HK2, HK3 (B) C,D. Label-free quantitation values from proteomic analysis depicting relative abundance of HK1, HK2, and HK2 in mouse (C) and human (D) cells. E,F. Proteomic abundances of hexokinases measured by LC-MS/MS with internal standards. (E) Depicts plots of absolute and relative abundances on mouse microglia and (F) depicts human cells. G,H. Normalized counts depicting transcript abundance of Pfk1, Pfkp, and PfkM (G) and PFKL, PFKP, and PFKM (H) I,J. Label-free quantitation values depicting relative abundance of PFKL, PFKP, and PFKM in mouse (I) and human (J) cells. K,L. Proteomic abundances of phosphofructokinases measured by LC-MS/MS with internal standards. (K) Depicts plots of absolute and relative abundances on mouse microglia and (L) depicts human cells. (M,N) Normalized counts depicting transcript abundance of Pfkfb1, Pfkfb2, Pfkfb3, and Pfkfb4 (M) and PFKFB2, PFKFB3 and PFKFB4 (N) O,P. Proteomic abundances of PFKFBs measured by LC-MS/MS with internal standards. (O) Depicts relative abundances on mouse microglia and (P) depicts human cells. For all panels, error bars represent SEM. Every biological replicate is depicted as a dot. For

mouse RNAseq n=3, for human RNAseq n=4, for mouse proteomics n=4 and for human proteomics n=5. n indicates biological replicates. For RNAseq data p values were determined by the Wald test and multi testing corrected in DESEQ2. For proteomic data, p values were determined by unpaired two-tailed t-test. *p<0.05, **p<0.01, and ***p<0.001

By adding these data to our manuscript, we have shown that Hexokinase isoform abundance in microglia is species-specific, PFKFB involvement is species-specific at the transcriptomic and proteomic level. Based on these new findings we discussed in the last paragraph of our discussion section:

“Most transcriptomic changes in glycolysis occurred in hexokinase and phosphofructokinase in mouse microglia, while transcriptomic and proteomic changes occurred in the PFKs and PFKFBs in human iMGLs. The direct interrogation of mRNA and protein changes is a topic that remains to be further investigated as a plethora of mechanisms which are species-specific govern transcription, translation, and protein homeostasis.”

2. In the TCA cycle section, citrate synthase, isocitrate dehydrogenase, and α -ketoglutarate dehydrogenase are the key enzymes. Why do they show differences in both mouse and human microglia, for instance citrate synthase is decreased in the mouse microglia but increased in the human microglia? α -ketoglutarate dehydrogenase is only altered in the mouse microglia but not in the human microglia, how do authors address these different changes due to the LPS treatment? It would be helpful to validate these statistically significant changes in the TCA cycle described in figure 5 by western.

We thank the reviewer for their relevant insight on the expression of key TCA cycle enzymes. To address this concern, we have performed additional experiments and conducted LC-MS proteomic analysis, revisited our transcriptomic data, and interpreted it in light of the proteomic analysis of microglia. This data is included in the manuscript as Supplementary Figure 6. All in all, these results strongly suggest that a downregulation of gene transcription may not be paralleled by a decreased abundance of protein. This suggests that in order to observe a downregulation in such a short period of time (i.e. 4 hours), protein should not only decrease its transcription/translation, but increase degradation parallelly. Conversely, inducible changes such as upregulation of FH in human but not mouse microglia could be recapitulated with our targeted proteomic approach, strengthening our transcriptomic findings on LPS responses across species. This issue is also addressed in the discussion section following remarks of the previous comment of the reviewer. The following text was added in the manuscript:

*“Next, we compared the transcriptomic data with the proteomic data, focusing on the key enzymes of the TCA cycle. It was observed that transcripts of *Idh1* and *IDH1* were the most abundant isoforms in mouse (Fig S6A) and human cells (Fig S6B), respectively. With the proteomic analysis, we could determine that *IDH2* is the most abundant isoform in mouse (Fig S6C, Fig S6D) microglia, while both *IDH1* and *IDH2* have comparable abundances in human microglia. The transcriptomic profile of the transcripts of TCA cycle enzymes showed a significant downregulation of *Idh1* for mouse (Fig S6E) and upregulation of *ACO1*, *FH*, *CS* transcripts with downregulation of *IDH1* and *IDH2*. in human microglia (Fig S6F). Overall, we observed no significant differences in the TCA cycle enzymes after LPS stimulation by label-free proteomic quantitation (Fig S6G, and Fig S6H). However, our targeted TCA proteomics approach yielded in discrete upregulations that did not reach statistical significance in mouse microglia (Fig S5I), but yielded significant upregulation of *ACO2*, *IDH2*, and *MDH2* (Fig S5J) in human iMGLs. All in all, these results strongly suggest that TCA cycle activity may be enhanced in human cells, but not in mouse cells. Both species exhibit downregulation of *Idhs* and *IDHs* at the transcriptional level. However, proteomic differences become evident for upregulations of TCA cycle enzymes at the 4hr LPS-treatment timepoint.”*

Supplementary Figure 6. Transcriptomic and proteomic characterization of TCA cycle enzymes in mouse microglia and human iMGLs. Mouse and human microglial cells were in vitro stimulated with LPS 250ng/mL and collected 4 hours after stimulation for proteomic analysis by LC-MS/MS. A,B. Normalized counts depicting transcript abundance of genes that code for Isocitrate dehydrogenase enzymes in mouse (A) and human (B) cells. C,D. Label-free quantitation values from proteomic analysis depicting relative abundance of *IDH1*, *IDH2*, and *IDH3A* and *IDH3B* in mouse (C) and human (D) cells. E,F. Normalized counts depicting transcript abundance of genes that code for TCA cycle enzymes in mouse (E) and human (F) cells. G,H. Label-free quantitation values depicting relative abundance of TCA cycle enzymes in mouse (G) and human (H) cells. I,J. Proteomic abundances of TCA cycle enzymes measured by LC-MS/MS with internal standards. (I) Depicts plots of relative abundances on mouse microglia and (J) depicts human cells. For all panels, data are presented as mean \pm S.E.M.. Every biological replicate is depicted as a dot. For mouse RNAseq $n=3$, for human RNAseq $n=4$, for mouse proteomics $n=4$ and for human proteomics $n=5$. n indicates independent biological experiments. For RNAseq data p values were determined by the Wald test and multi testing corrected in DESEQ2. For proteomic data, p values were determined by unpaired two-tailed t-test. * $p<0.05$, ** $p<0.01$, and *** $p<0.001$

4) What do the authors mean with “substantial collection-dependent effect on the transcriptomes”? This is not clear.

By “substantial collection-dependent effect” we were aiming to convey the message that there was a visible batch effect per collection. Therefore, we needed to include the collection number as a covariate. In the following figure, it is clear that samples cluster also by collection. We have added these data to Supplementary Figure 2 and mention it briefly in the results and discussion sections.

Supplementary Figure 2. Human iMGLs response to LPS

A, B. Principal Component Analysis (PCA) of the transcriptomes of untreated, and LPS-treated iMGLs for 4h and 24h. In (A), samples are colored by condition. In (B), samples are colored by collection/biological replicate. C. Heatmap depicting the top 100 most significant differentially expressed genes of 4h LPS treatment compared with the control. Color key corresponds to row Z-score. D. Volcano plot depicting fold changes and $-\log_2$ of the adjusted p value per gene comparing responses against LPS 24h and 4h.

5) A key point when assessing differences between species is that both microglia are not cultures in the same conditions. Given that the media (specially serum) can have critical effects on microglia, I do not think the data should be compared.

Murine microglia have been reported to acquire an activated phenotype upon serum withdrawal¹⁰. This is the reason why we decided to culture primary microglia with serum-containing media. We have performed control experiments where we stimulated murine microglia with LPS in the absence of serum and we observed a similar pattern to those cultured with serum (Supplementary Figure 1). Additionally, we added serum to human microglia-like cell cultures and observed a significant decrease in viability (Supplementary Figure 9). This is corroborated by Washer and colleagues¹⁶, where the inclusion of FBS in the maturation medium for microglia progenitors was detrimental. Therefore, experiments could not be performed with serum-containing media for the human microglial model. We have added these important data to Supplementary Figures 1 and 9 (see below).

Supplementary Figure 1. LPS response to mouse microglia. A, B. Mouse cells were seeded in 96-well E-plates at a density of 15,000 cells/well and monitored by a real-time impedance-based xCELLigence system. Mouse microglia or iMGL cells were challenged with increasing concentrations of LPS (10-1000ng/ml) in the absence (A) or presence (B) of serum for 24hr. C. IBA1 immunostaining of control and LPS-treated mouse microglia with and without FBS. D. Heatmap depicting the top 100 most significant differentially expressed genes of 4h LPS treatment compared with the control. Color key corresponds to row Z-score. E. Volcano plot depicting fold changes and $-\text{Log}_2$ of the adjusted p value per gene comparing responses against LPS 24h and 4h.

Supplementary Figure 9. Effect of FBS on viability of iMGLs. A. IBA1 immunostaining of untreated and LPS-treated human iMGLs with and without serum.

6) How many iPSC lines were used in this study? And how many biological replicates? It seems from the plot it is only one line, n=3 replicates. This does not seem enough, unless orthogonal experiments (e.g., metabolomics) are performed.

For functional experiments, we used at least 3 lines from different subjects (Gibco, Control 8.2, and isogenic control) with at least 3 independent batches of experiments. We aimed to profile the effect of the classic TLR4 agonist LPS on microglial activation, therefore we decided to use many collections from one cell line for the RNAseq experiments and the other cell lines for functional validation. This approach has served other studies such as the study from Geric and colleagues¹² where only one cell line was employed to profile metabolic shifts in microglia.

For RNA-seq experiments, we believe it is crucial to show to the scientific community the reproducibility of the iPSC-differentiation towards microglia and that there are important batch effects even within an experiment/differentiation. Although we observed robust effects of LPS on lactate, ATP, OCR and ECAR measurements, the high sensitivity of RNA-seq demonstrates that collections behave differently between each other. We have decided to include these considerations in our discussion. Similarly, when the microglia transcriptome is compared to

another cell type, such as iPSCs, these collection differences become minimal given that the dimensionality reduction nature of PCA would detect the differences between iPSCs and iMGLs as top principal components.

7) Along the text, many genes are highlighted, but it is unclear to this reviewer the reason behind the selection of some genes but not others. Perhaps it could be more relevant to explore pathways first, and then discuss the genes differentially expressed across conditions in relevant biological processes?

In the present study we aimed to profile the transcriptome and the metabolism that are dysregulated upon LPS-induced inflammatory processes across species. As a high-throughput technique, the amount of data we obtain is substantial. In the main text, as a descriptive feature, we mentioned several genes that are observable in the volcano plots of figures 1 and 2. These genes are classically-regarded as inflammatory genes. In order to have a more systems-medicine-oriented view, we performed GSEA to understand how these genes enrich or are related to various biological processes.

The type of analysis that was performed for GSEA with the metascape tool requires the over-representation of data to calculate enrichment scores for biological processes. This type of analysis does not rank genes according to their p value or fold change.

In the manuscript, we have mentioned that most of the dysregulated biological processes of figure 1F, for instance, are mainly inflammatory. Highlighting the nature of LPS-responses: inducible gene changes. Afterwards, we performed the overrepresentation analysis of the downregulated genes from our comparisons and observed that many metabolic pathways were enriched at both timepoints. Of interest, we mentioned in the manuscript that pathways such as lipid modification, carbohydrate metabolic process and lysosome are also enriched.

To make sure this is clear, we have modified the results in page 5 to explicitly mention that the highlighted genes are mere examples (i.e.) of genes that are comprised in such processes. Associated data of this analysis is available upon request.

8) Figure 5 is very interesting. However, as said before, the fact that both in vitro systems use completely different media compositions probably introduce too many sources of variation and bias the results. I suggest this is repeated using the same culturing conditions for both species.

We thank the reviewer for their constructive comment. We want to note that previous studies employing iPSC-derived microglia-like cells and mouse microglia have already acknowledged this serum presence discrepancy¹². From a practical point of view, in our hands mouse microglia with and without serum have comparable responses to LPS. However, human microglia are not cultured with FBS and upon FBS addition, these cells die. We believe this is a starting point for cross-species comparison and our data should be mentioned in the discussion for the microglial community to address this discrepancy for future research. Please see Supplementary Figures 1 and 9 from previous comment. We agree with this point of the reviewer and included this discussion in the “Limitations of the study”

“Our results should be interpreted in the light of the wide plethora of the experimental setups of primary microglia and iMGLs that prevails in the field (i.e. FBS presence, LPS concentrations, etc..).”

Reviewer #3 (Remarks to the Author):

The manuscript by Sabogal-Guaqueta and colleagues compares the metabolic reprogramming of human and mouse microglial cells upon inflammatory stimuli. The topic is of high interest for the importance of microglial roles in many inflammatory conditions, and the comparison among human- and mice-derived cells is fundamental for the translatability of data. Importantly, the authors identified different species-specific mechanisms of LPS-induced reprogramming and described different kinetics of gene modulation. They demonstrated that:

1) LPS promotes an enhanced glycolytic metabolism in both species, but this is the result of increased expression of hexokinase in mice and phosphofructokinase in humans; 2) LPS induces a downregulation of oxidative metabolism in both species but associated to a differential time profile; 3) LPS promotes glycolysis coupled to mitochondrial dysfunction only in mouse microglia not in human iMGL. Some limits of this work are acknowledged by the authors, and also by this reviewer.

1) Comparison of human iMGL with cultured mouse microglia is not straightforward. It would be necessary to compare the effects of LPS on the same kind of cells, i.e. iMGL from humans and mice. In absence of these experiments, the novelty of the results is mitigated by previous demonstration of transcriptional differences among human and rodent microglia (Smith and Dragunow, Trends in Neurosciences 2014; Galatro et al., Nat. Neurosci. 2017; Gosselin et al., Science 2017; Burns et al., Eur. J. Pharmacol. 2015 and others).

We partially agree with the reviewer. Multiple efforts from the lab of Marco Prinz have focused on the identification of a core gene program of orthologous genes from rodents to humans¹⁷. This work has also led to the identification of notable divergences in microglial pathways across evolution. From these, metabolic pathways such as glycolysis and TCA cycle were not studied. Additionally, their study did not include the response to LPS. iPSC-derived model systems have emerged as an effort to model human diseases given that cross-species differences may confound positive therapeutic outcomes in humans.

While mouse iPSC systems may be valuable for many aspects of research, shifting towards human iPSC systems, specifically for the study of microglia in disease states, can offer greater translational potential and clinical relevance. It is important to assess the specific research objectives and consider the most appropriate model system based on the scientific goals and the desired impact on human health.

A resource on mouse iPSC-differentiated microglia-like cells published by Pandya and colleagues¹⁸ demonstrated that the transcriptome of iMGLs resembles the fetal but not the adult microglia. However, caution must be taken when interpreting their results and ours. The protocol they employed for generation of iMGLs is substantially different to ours. Conversely, the transcriptomic technique they employed was micro-array. So, our data cannot be compared. There are no more resources on iPSC-differentiated microglia-like cells from mouse origin.

*Although the point of the reviewer is relevant when comparing side-by-side iPSC-derived microglial generation of mouse and human origin, the specific factors that are required during the patterning and differentiation are very different in mouse and in humans. The timeline is also very different during the developmental stages. To establish a mouse iPSC microglial culture would take more than a year. In addition, we spent more than three years for a proper validation and characterization of the human microglial cells. The aim of our study was to compare a generally well-known and standard mouse microglial culture model and compare it with the emerging iPSC-derived human models. We did not intend to establish a new mouse iPSC-derived model. Having mentioned this, and in order to address the reviewer's concern, we would like to highlight these considerations in the **Limitations and considerations** section of this manuscript.*

“Mouse iPSC-derived microglia were described and characterized in 2017¹⁸, concluding that their gene expression is consistent with primary neonatal microglia. Although in this study, iPSC-derived mouse microglia were not studied, we compared the LPS responses of primary neonatal microglia with the human iMGL cells, which may also exhibit young phenotypes.”

2) The authors should consider that human and mouse responsiveness to LPS could differ for a different expression of TLR4 in the two species (Vaure and Liu, Front. Immunol. 2014; Parajuli et al., J. Neuroinflamm. 2019; Carpentier et al., BBI 2008; Jack et al., J. Immunol 2005; Casula et al., 2011).

We thank the reviewer for raising this concern and citing relevant literature. Although it is not straightforward to address, TLR4 expression differences across species are more complex than one might expect. We start by noting that in human CNS, microglia and astrocytes express TLR4, while in mice only microglia do¹⁹. Interestingly the review from Vaure and Liu highlights that many disparate features across species may be attributed to different experimental settings and materials. They propose the cross-analysis of humans with results from animal studies using a system biology approach to further refine the animal models used for non-clinical assessment of TLR4 agonists. We are confident that our transcriptomic and proteomic screenings contribute to this view on responses of TLR4 agonism in the microglial field.

To address the reviewer’s concern, we surveyed our transcriptomic data and compared the TPM values across human and mouse microglia as it is worth it to be mentioned in the manuscript. We included this to Supplementary Figure 3I in our manuscript.

Supplementary Figure 3. Cross-species comparison of transcriptomic responses against LPS over time. A, E. Overlap of DEGs in mouse microglia (A) and human iMGLs (E) at 4 and 24 hours after LPS stimulation. B, F. Top enriched biological processes after performing GSEA with the metascape tool with the overlapped gene lists for mouse microglia (B) and human iMGLs (F). C, G. Circos plot depicting how genes from the lists of DEGs overlap for mouse microglia (C) and human iMGLs (G) for the two timepoints. On the outside, each arc represents the identity of each gene list. On the inside, each arch represents a gene list, where each gene member of that list is assigned a spot on the arc. Dark orange color represents the genes that are shared by multiple lists and light orange color represents genes that are unique to that gene list. Purple lines link the same gene that are shared by multiple gene lists (notice a gene that appears in two gene lists will be mapped once onto each gene list, therefore, the two positions are purple linked). Blue lines link the genes, although different, fall under the same ontology term (the term has to be statistically significantly enriched and with size no larger than 100). The greater the number of purple links and the longer the dark orange arcs imply greater overlap among the input gene lists. Blue links indicate the amount of functional overlap among the input gene lists. D, H. Scatterplots of the genes that were differentially regulated by LPS 4h and 24h

treatment (red dots) with an adjusted $p < 0.05$ in mouse **microglia (D)** and human **iMGLs (H)**. A linear model was fitted among the genes that were differentially regulated and the R squared, and the equation of the line are depicted within the scatterplot. **I. Gene expression of the *Tlr4* and *TLR4* genes depicted by the normalized measure of TPMs per species.**

3) Limits of iPSC-derived microglia must be also considered in transcriptomics: transcriptomic and phenotypic deficiencies due to the lack of interaction with other brain parenchyma cells (Abud et al., Neuron 2017: 3-dimensional brain organoid (BORG) co-cultured with iPSC-derived microglia; Hasselmann et al., Neuron 2019: xenotransplantation approaches).

We agree with the reviewer that 2D and 3D setups modify the transcriptomic profile of microglia. We have performed preliminary experiments where we co-cultured iPSC-derived microglia with brain organoids, with subsequent CD11b sorting and RNA sequencing. Unfortunately, the RNA yield for such an experiment was low, not allowing for a comprehensive analysis of the transcriptomes. We have included this data here, only visible for the reviewer. However, we believe that 2D and 3D comparisons are a next step in the characterization of microglial modeling and should be addressed in future research. We have included these considerations in our discussion section.

Revision Figure. iMGLs-co-cultured with brain organoids. 30K iMGLs were co-cultured with brain organoids. After 1 week, CD11b positive cells were sorted after disruption of the brain organoid. RNA was isolated using RNeasy micro kit. Minibulk RNAseq analysis was performed with cells per sample between 11 and 25. PCA denotes differences in the analyzed transcriptomes of 2D, plastic-grown microglia, and the transcriptomes of microglia that were in the organoid. Samples are labeled as follows: Differentiated microglia: SCB01 and SCB73, differentiated microglia that have been in organoid (SCB20, SCB18, SCB16), and organoid samples (SCB30, SCB31, SB31).

4) Fig:5: the authors should discuss the results on aconitase, which are opposite for human and mouse, and at least mention why FH data are only shown for humans.

To address the reviewer’s concern regarding RNA expression of FH, and to strengthen our understanding of the changes that occur within the TCA pathway, we performed additional

experiments and conducted two proteomic mass spectrometry analysis methods: i) label-free (termed LFQ) and ii) targeted proteomics on TCA and glycolytic proteins. This data is included in the manuscript as Supplementary Figure 6 and described under results subsection **Main enzymes of the TCA cycle are dysregulated in mouse and human microglia after LPS challenge:**

“Next, we compared the transcriptomic data with the proteomic data, focusing on the key enzymes of the TCA cycle. It was observed that transcripts of *Idh1* and *IDH1* were the most abundant isoforms in mouse (Fig S6A) and human cells (Fig S6B), respectively. With the proteomic analysis, we could determine that IDH2 is the most abundant isoform in mouse (Fig S6C, Fig S6D) microglia, while both IDH1 and IDH2 have comparable abundances in human microglia. The transcriptomic profile of the transcripts of TCA cycle enzymes showed a significant downregulation of *Idh1* for mouse (Fig S6E) and upregulation of *ACO1*, *FH*, *CS* transcripts with downregulation of *IDH1* and *IDH2* in human microglia (Fig S6F). Overall, we observed no significant differences in the TCA cycle enzymes after LPS stimulation by label-free quantitation (Fig S6G, and Fig S6H). However, our targeted proteomics approach yielded in discrete upregulations that did not reach statistical significance in mouse microglia (Fig S5I), but yielded significant upregulation of *ACO2*, *IDH2*, and *MDH2* (Fig S5J) in human iMGLs.

All in all, these results strongly suggest that TCA cycle activity may be enhanced in human cells, but not in mouse cells. Both species exhibit downregulation of *Idhs* and *IDHs* at the transcriptional level. However, proteomic differences become evident for upregulations of TCA cycle enzymes at the 4hr LPS-treatment timepoint.”

Supplementary Figure 6. Transcriptomic and proteomic characterization of TCA cycle enzymes in mouse microglia and human iMGLs. Mouse and human microglial cells were in vitro stimulated with LPS 250ng/mL and collected 4 hours after stimulation for proteomic analysis by LC-MS/MS.

A,B. Normalized counts depicting transcript abundance of genes that code for Isocitrate dehydrogenase enzymes in mouse (A) and human (B) cells. C,D. Label-free quantitation values from proteomic analysis depicting relative abundance of IDH1, IDH2, and IDH3A and IDH3B in mouse (C) and human (D) cells. E,F. Normalized counts depicting transcript abundance of genes that code for TCA cycle enzymes in mouse (E) and human (F) cells. G,H. Label-free quantitation values depicting relative abundance of TCA cycle enzymes in mouse (G) and human (H) cells. I,J. Proteomic abundances of TCA cycle enzymes measured by LC-MS/MS with internal standards. (I) Depicts plots of relative abundances on mouse microglia and (J) depicts human cells. For all panels, data are presented as mean \pm S.E.M.. Every biological replicate is depicted as a dot. For mouse RNAseq n=3, for human RNAseq n=4, for mouse proteomics n=4 and for human proteomics n=5. n indicates biological replicates. For RNAseq data p values were determined by the Wald test and multi testing corrected in DESEQ2. For proteomic data, p values were determined by unpaired two-tailed t-test. * $p < 0.05$, ** $p < 0.01$, and *** $p < 0.001$

5) Authors should limit the use of “interesting” while presenting their data (used several times in the text), and/or elaborate the implication of the data.

We are thankful for the remark and expanded the discussion, we limited the use of “interestingly”.

6) Fig.6: the authors should better discuss the seahorse results showing that OCR is different between humans and mice while ECAR is similar.

We thank the reviewer for voicing this concern. We performed additional experiments by including ATP, lactate and NAD/NADH measurements. Measurements of ATP production may correlate with the Oxygen consumption findings upon LPS stimulation while lactate may correlate with glycolysis. Therefore, new experiments on lactate and ATP measurements were added to Figure 6 with the oxygen consumption ratio and extracellular acidification rate data. Discussion was added in the manuscript too.

To better understand the glycolytic shift that these cells undergo, we profiled lactate production in human and mouse microglia at 4, 8, and 24 hours. Consistent with our data, we observe an increase in lactate concentrations in microglia supernatants, for both species. We have added this data, along with the ATP measurements to the extracellular acidification data in Figure 7Q,R,S and T.

Figure 7. Mitochondrial respiration analysis of mouse and human microglia. Representative experiment showing OCR (oxygen consumption rate) in A. mouse and B human microglial cells treated with LPS for 4 and 24h. Measurements were obtained by an extracellular flux analyzer (Seahorse Bioscience). C-E. Bar graph showing different mitochondrial respiration parameters represented as mean \pm SEM of the OCR values in mouse and F-H in human microglia. I. ATP measurements of mouse and J microglia treated with LPS for 4, 8 and 24h expressed as a percentage of the untreated control for the same timepoint. Data from individual experiments have the same color. K. Representative experiment showing ECAR (Extracellular acidification rate) in K. mouse and L. human microglia cells treated with LPS for 4 and 24h. M-R. Bar graph showing different mitochondrial respiration parameters represented as mean \pm SEM of the ECAR values in mouse and P-R in human. S. Lactate measurements of the supernatants of mouse

microglia treated with LPS for 4, 8 and 24h expressed as a percentage of the untreated control for the same timepoint in mouse and T in human microglia. Data from individual experiments have the same color. Data are shown as mean \pm SEM, n = 3–6 technical replicates. All experiments were repeated at least three times for each cell line (Ctr 8.2 & Gibco). *p*-values indicating statistically significant differences between the mean values are defined as follows: **p* < 0.05, ***p* < 0.01 (compared with ctr), #*p* < 0.05, ##*p* < 0.01 (LPS 4h vs LPS 24h). Data for lactate and ATP measurements are showing 4-5 different experiments where each color point represented an experiment.

Minor points:

1) Line 258: “human iMGC cells resembles the adult or fetal human brain regarding their transcriptome.” How do the authors reach this conclusion? Do they compare TMEM119, P2Y12 receptor and IBA 1 at protein levels (immunofluorescence) or they compare their PCR analysis with literature data? It is not clear.

We have defined our microglial identity by direct interrogation of TMEM119, P2YR12 and IBA1 in experimental setups. We have based our remarks of Line 258 on the whole-transcriptome analysis of iPSCs, HPCs, macrophages, primary microglia and previously-published iPSC-derived microglia-like cells. This approach does not only consider the previously mentioned markers, but the whole set of expressed genes. Given that the microglia samples cluster with other microglia, both primary and iPSC-derived, we state that they “resemble” each other. We have modified such statements to make it clear that we are based on our RNA-seq meta-analysis.

Have they also analyzed microglia from a functional point of view in addition to phagocytosis, i.e. patrolling, migration or others?

No, for this manuscript to keep the focus on metabolic pathways we did not discuss them, although we can clearly detect them in the RNAseq data.

2) Line 459 Cytochalasin D is not a phagocytosis inhibitor.

It is an actin polymerization inhibitor. We have modified it in the text.

3) In a number of experiments the number of repeats seems too low considering sample variability (i.e. some transcriptomic results in figure 3 and 4).

For functional experiments, we used at least 3 lines from different subjects (Gibco, Control 8.2, and isogenic control) with at least 3 independent batches of experiments. We aimed to profile the effect of the classic TLR4 agonist LPS on microglial activation, therefore we decided to include many collections from one cell line for the RNAseq experiments and the other cell lines for functional validation. Transcriptomics was performed in parallel experiments with microglia derived from Gibco iPSC line and control 8.2 iPSC line.

For RNA-seq experiments, we believe it is crucial to show to the scientific community the reproducibility of the iPSC-differentiation towards microglia and that there are important batch effects even within an experiment/differentiation. Although we observed robust effects of LPS on lactate, ATP, OCR and ECAR measurements, the high sensitivity of RNA-seq demonstrates that collections behave differently between each other. We have decided to include these considerations in our discussion. Similarly, when the microglia transcriptome is compared to another cell type, such as iPSCs, these collection differences become minimal given that the dimensionality reduction nature of PCA would detect the differences between iPSCs and iMGLs as top principal components.

We used n=3-5 number of independent experiments depending on the setup.

4) The authors state that “Studies of human brain microglia have been performed on isolated microglia from fresh *post-mortem* samples from potentially neuropathologically affected

individuals, which might be hindered by a high interindividual variation.” However, this limitation also applies to human iMGL cells, as in Fig, 3C, and should be acknowledged.

We agree with the reviewer. A primary microglial culture may be influenced by the innate immune memory determined by the life-time of immune experiences that the individual has been through. The plethora of microglial phenotypes of postmortem tissue has been studied extensively, with post-mortem delay having an added confounding effect on the microglial transcriptome. However, culture systems offer a more stringent control on culture conditions and collection timing. We highlight the inter-collection variability which is inherent to our differentiation protocol.

Previous studies performed RNA-seq of iPSC-derived microglia and compare it with the transcriptome of iPSCs and HPCs. However, when only iPSC-derived microglia are analyzed, we can observe these collection differences. We show in our meta-analysis PCA that our 4 control samples cluster relatively together. This is based on the recommendations of the consensus paper on the field on microglia to exploit the use of -omics²⁰. However, when we only take into account the transcriptomes of samples of our study, we have a higher resolution to detect differences across our differentiation.

5) Two different animal Protocols are provided, in two paragraphs named “animals”. Which one is used? If both, differences should be mentioned.

We have revisited our methods section and corrected this discrepancy.

6) Line 250-252: “Microglial progenitors were further differentiated with IL-34, TFG- β , GM-CSF, CD200, and CX3CR1 to assure a full maturation profile.” CX3CL1 should substitute CX3CR1

This is noted and modified.

References

1. Alasoo, K. *et al.* Transcriptional profiling of macrophages derived from monocytes and iPSC cells identifies a conserved response to LPS and novel alternative transcription. *Sci Rep* **5**, 12524 (2015).
2. Hasselmann, J. *et al.* Development of a Chimeric Model to Study and Manipulate Human Microglia. *Neuron* **103**, 1016-1033.e10 (2019).
3. Pulido-Salgado, M., Vidal-Taboada, J. M., Barriga, G. G.-D., Solà, C. & Saura, J. RNA-Seq transcriptomic profiling of primary murine microglia treated with LPS or LPS + IFN γ . *Scientific Reports* **8**, 16096 (2018).
4. Lasselin, J. *et al.* Comparison of bacterial lipopolysaccharide-induced sickness behavior in rodents and humans: Relevance for symptoms of anxiety and depression. *Neuroscience & Biobehavioral Reviews* **115**, 15–24 (2020).
5. Gosselin, D. *et al.* An environment-dependent transcriptional network specifies human microglia identity. *Science (1979)* (2017) doi:10.1126/science.aal3222.
6. Zhang, X. *et al.* Epigenetic regulation of innate immune memory in microglia. *J Neuroinflammation* **19**, 1–19 (2022).
7. Dolga, A. M. *et al.* Activation of KCNN3/SK3/KCa2.3 channels attenuates enhanced calcium influx and inflammatory cytokine production in activated microglia. *Glia* (2012) doi:10.1002/glia.22419.
8. Oun, A. *et al.* Characterization of Lipopolysaccharide Effects on LRRK2 Signaling in RAW Macrophages. *International Journal of Molecular Sciences* vol. 24 Preprint at <https://doi.org/10.3390/ijms24021644> (2023).
9. Gosselin, D. *et al.* Environment drives selection and function of enhancers controlling tissue-specific macrophage identities. *Cell* **159**, 1327–1340 (2014).
10. Bohlen, C. J. *et al.* Diverse Requirements for Microglial Survival, Specification, and Function Revealed by Defined-Medium Cultures. *Neuron* (2017) doi:10.1016/j.neuron.2017.04.043.
11. Njie, eMalick G. *et al.* Ex vivo cultures of microglia from young and aged rodent brain reveal age-related changes in microglial function. *Neurobiology of Aging* **33**, 195.e1-195.e12 (2012).
12. Geric, I. *et al.* Metabolic Reprogramming during Microglia Activation. *Immunometabolism* **1**, (2019).
13. Chhor, V. *et al.* Characterization of phenotype markers and neuronotoxic potential of polarised primary microglia in vitro. *Brain, Behavior, and Immunity* **32**, 70–85 (2013).
14. Galatro, T. F. *et al.* Transcriptomic analysis of purified human cortical microglia reveals age-associated changes. *Nature Publishing Group* (2017) doi:10.1038/nn.4597.
15. Janssen, L. *et al.* Aging, microglia and cytoskeletal regulation are key factors in the pathological evolution of the APP23 mouse model for Alzheimer's disease. *Biochimica et Biophysica Acta (BBA) - Molecular Basis of Disease* **1863**, 395–405 (2017).
16. Washer, S. J. *et al.* Single-cell transcriptomics defines an improved, validated monoculture protocol for differentiation of human iPSC to microglia. *Scientific Reports* **12**, 19454 (2022).
17. Geirsdottir, L. *et al.* Cross-Species Single-Cell Analysis Reveals Divergence of the Primate Microglia Program. *Cell* **179**, 1609-1622.e16 (2019).
18. Pandya, H. *et al.* Differentiation of human and murine induced pluripotent stem cells to microglia-like cells. *Nat Neurosci* **20**, 753 (2017).

19. Vaure, C. & Liu, Y. A Comparative Review of Toll-Like Receptor 4 Expression and Functionality in Different Animal Species . *Frontiers in Immunology* vol. 5 Preprint at (2014).
20. Paolicelli, R. C. *et al.* Microglia states and nomenclature: A field at its crossroads. *Neuron* **110**, 3458–3483 (2022).

REVIEWERS' COMMENTS

Reviewer #1 (Remarks to the Author):

The authors have extensively revised the manuscript and performed numerous follow-up experiments as requested. This includes proteomics and bioinformatic analysis in comparison to other datasets. Overall the comments were reasonably addressed, a few points remain.

1. In figure 5, the group colors are inconsistent, and the point formatting is also inconsistent. Especially Fig. 5F and 5I among others.
2. The authors show in supplementary figure 6 that multiple TCA cycle genes are up regulated by proteomics after four hours of LPS, but do not present a theory for why OCR and ATP then decrease in human microglia in the Seahorse at four hours, the same timepoint undertaken in their proteomics?
3. For Fig 2A venn diagram, a hypergeometric p value would be helpful.
4. In Figure 2B the circus plots are turned such that the text is not readable, presumably by accident.
5. The authors note that mice and humans have different specific activities for key enzymes in glycolysis and the TCA cycle, but overall, this different expression or activity does not affect glycolytic capacity, which has been tested to be similar in mice and humans. Are there any different metabolic pathways between mice and humans?

Reviewer #2 (Remarks to the Author):

The authors have satisfactorily addressed most of my comments. The only point missing that I still feel is not completely addressed is the fact that mouse and human cultures are performed in different conditions, likely having an impact on the cell metabolism. Presence of serum has been shown to dramatically affect microglia (<https://www.ncbi.nlm.nih.gov/pmc/articles/PMC5523817/>) producing transcriptomic changes (even in metabolic pathways) that are long lasting. I feel this is not entirely address, and either additional key experiments (e.g. assessing changes in gene expression of the pathways highlighted in the manuscript in mouse microglia cultured in the presence of serum, or in serum free media), or a much more thorough discussion needs to be added.

Reviewer #3 (Remarks to the Author):

This revised version of the manuscript answered my previous concerns. The limitations of the paper have been significantly reduced or acknowledged and the manuscript is now very much improved.

Reviewer #4 (Remarks to the Author):

My review focuses on data in this manuscript derived from proteomics. I agree with other reviewers that investigating metabolic reprogramming requires investigation at the protein level, which authors have now included, both with discovery, label free proteomics and targeted proteomics. However, the following points need to be addressed:

1. I do not see anything in the methods or data availability section that states where the raw proteomics data is available. It is essential that the raw mass spectrometry/ proteomics data is shared to a public repository such as PRIDE/ Proteome Xchange. Proteomics data is also not present in the supplementary files included.
2. Figure 5. Figure legend poorly describes the data, I find it very hard to understand each graph and what it is meant to be showing. Sometimes Figure letters are enclosed within brackets, sometimes not. Full stops are missing. Please be consistent.
3. Figure 5. Please state in figure legend and on graphs that Figures E,F,K,L, O and P are derived from data from targeted proteomics rather than LFQ. You do this in Sup Fig 6 but not in main Fig 5. I am also confused by the statement of 'absolute and relative abundancies' to describe this data as it is displayed as fold change only. Please amend.
4. Figure 5F. Not sure why a darker shade of green is used instead of blue for LPS4hr.
5. Would be good to state how many proteins were identified by discovery proteomics in both mouse and human samples. This will give an overall idea of the quality and coverage gained from the methods.

Response to Reviewers (*Responses in Italics*)

Thank you very much for the time and for reviewing our manuscript NCOMMS-23-04080. We thank the editor and the reviewers for their valuable comments to improve our manuscript.

Reviewer #1 (Remarks to the Author):

The authors have extensively revised the manuscript and performed numerous follow-up experiments as requested. This includes proteomics and bioinformatic analysis in comparison to other datasets. Overall the comments were reasonably addressed, a few points remain.

1. In figure 5, the group colors are inconsistent, and the point formatting is also inconsistent. Especially Fig. 5F and 5I among others.

Response: Thank you for this observation. We have corrected it.

2. The authors show in supplementary figure 6 that multiple TCA cycle genes are up regulated by proteomics after four hours of LPS, but do not present a theory for why OCR and ATP then decrease in human microglia in the Seahorse at four hours, the same timepoint undertaken in their proteomics?

Response: We thank the reviewer for pointing out this missing remark. Given the discrepant relation between decreased OCR and ATP with gene and protein expression of TCA cycle enzymes for human and mouse cells at LPS 4h, we hypothesize that the decreased OCR in human cells may be confounded by biochemical properties such as reaction speeds, half-lives, cellular environments, feedback loops and degradation rates of TCA cycle genes at this time point. Such changes have been described during development¹. A difference in such parameters would mechanistically imply that upon an increased gene and protein expression, OCR and ATP would remain unchanged or decreased but not increased in human cells at 4hr. Such biochemical characterizations and their relation to species-specificity remain to be investigated. We have included the following statements in our discussion:

Last paragraph of discussion:

“Given the discrepant relation between decreased OCR and ATP with gene and protein expression of TCA cycle enzymes for human and mouse cells at LPS 4h, we hypothesize that the decreased OCR in human cells may be confounded by biochemical properties such as reaction speeds, half-lives, cellular environments, feedback loops and degradation rates of TCA cycle genes at this time point. Such changes have been described during development¹. A difference in such parameters would mechanistically imply that upon an increased gene and protein expression, OCR and ATP would remain unchanged or decreased but not increased in human cells at 4hr. Such biochemical characterizations and their relation to species-specificity remain to be investigated.”

3. For Fig 2A venn diagram, a hypergeometric p value would be helpful.

We have taken the suggestion of the reviewer and have performed calculations of the statistical significance of the overlaps between gene sets based on the hypergeometric distribution. We have added the calculated p values and their enrichment level which is defined as the ratio of the number of observed overlaps over that of expected overlaps.

Response: We have modified Fig2A and added the following text in the statistical analysis subsection of the methods section.

“The statistical significance of the overlaps between gene lists is calculated based on the hypergeometric distribution. More specifically, given a total of N genes, if gene sets A and B contain m and n genes, respectively, and k of them are in common, then the p value of enrichment is calculated as follows:

$$p = \sum_{i=k}^{\min(m,n)} \binom{m}{i} \binom{N-m}{n-i} / \binom{N}{n}$$

Let us use Fig2A overlap as an example: $N=11852$, $A=1334+406$, $B=1344+406$, $k=406$. The numerical computation of these P values was performed in R. The enrichment level of the overlap is calculated by $R= kN/(mn)$ ”

4. In Figure 2B the circos plots are turned such that the text is not readable, presumably by accident.

Response: We have corrected this issue.

5. The authors note that mice and humans have different specific activities for key enzymes in glycolysis and the TCA cycle, but overall, this different expression or activity does not affect glycolytic capacity, which has been tested to be similar in mice and humans. Are there any different metabolic pathways between mice and humans?

Response: We thank the reviewer for pointing out this missing remark. And now we introduce the following paragraphs in the discussion section:

“In the pursuit of robust models for biomedical research, murine models have been commonly used, to ensure the translation of data to humans. However, there are some aspects that remain as central concerns: i) The disparities in metabolic pathways between these two species can potentially impact the reliability of extrapolations. Recent research by Matsuda et al. in 2020, showed that human presomitic mesoderm cells exhibit slower metabolism compared to their murine cells when they evaluated the transcription factor HES7. Similar results have been shown in generation of motor neurons² and biochemical reaction speeds of the p53-Mdm2 network³, highlighting the intricate interplay between species and their metabolic dynamics.

ii) These differences are not limited to inter-species disparities alone. Over the past few decades, metabolic differences have been identified within various strains of mice, not only in liver metabolism as noted by Bulfield et al. in 1977⁴ but also in brain and other tissues, as described by Burlikowska et al. in 2020⁵.

iii) Expanding the scope of the study, intriguing distinctions could emerge even during cellular reprogramming, as it was shown a notable divergence in bioenergetic remodeling between induced pluripotent stem cells and fibroblasts⁶. Collectively, it is fundamental to underline that both inter-species divergence and specific experimental conditions play a significant role over metabolic processes.”

Reviewer #2 (Remarks to the Author):

The authors have satisfactorily addressed most of my comments. The only point missing that I still feel is not completely addressed is the fact that mouse and human cultures are performed in different conditions, likely having an impact on the cell metabolism. Presence of serum has been shown to dramatically affect microglia (<https://www.ncbi.nlm.nih.gov/pmc/articles/PMC5523817/>) producing transcriptomic changes (even in metabolic pathways) that are long lasting. I feel this is not entirely address, and either additional key experiments (e.g. assessing changes in gene expression of the pathways highlighted in the manuscript in mouse microglia cultured in the presence of serum, or in serum free media), or a much more thorough discussion needs to be added.

Response: We thank the reviewer for their feedback and voicing their thoughts regarding culture conditions of our models. We created this table where we recognize the importance of serum in the experimental set-ups. It is important to underline that although the serum is not used in the experiments, it is used for isolation and maintenance of the cells used in these studies. Effect of the serum in the isolation and maintenance should be addressed in future studies.

Study Author, year	Microglia culture conditions	Effect on microglia
Koyama et al., 2000⁷	Mixed glial culture from cerebrum P1-P2 wistar rats (10% FBS)	Serum deprivation caused cell death and nuclei condensation. Addition of a specific inhibitor of p38MAPK SB2035800, prevented microglial cell death after serum deprivation.
Yenari & Giffard, 2001⁸	Murine glial culture from whole brains P1-P3 from Swiss Webster mice (10% FBS)	Serum deprivation caused cell death (apoptosis)
Negishi et al., 2003⁹	Fetal brains for cynomolgus monkey (Macaca fascicularis) were isolated and dissociated with serum	Serum deprivation after 14 days in vitro. Addition of M-CSF to promote microglial cell growth
Etemad et al., 2012¹⁰	Human blood mononuclear cells (PBMC) were isolated from buffy coats of healthy donors; fresh adherent cells, (monocytes>90%) were used for the generation of microglia	Removal of serum for generation of microglia. It was substituted for a cocktail of cytokines M-CSF, GMCSF, NGF and CCL-2 that promoted primary and secondary processes in the microglia.
Bohlen et al., 2017¹¹	Sprague-Dawley rats brain were removed for microglia cell culture. Astrocytes condition medium (ACM) was obtained from P1 rat cortices (10% FCS)	Removal of serum in microglia cultures was replaced with ACM to prevent microglial death. The reactivity of microglia was similar to the addition of serum when medium was supplemented with CSF-1/IL-34, TGF-b2, and cholesterol.
Collins & Bohlen, 2018¹²	Juvenile rat brains were used to generate microglia cultures	Use of TGF-β2/IL-34/cholesterol in culture to have serum-free culture conditions. These conditions support high levels of microglial viability over time. Microglia exhibit elaborate ramified processes and dynamic surveillance behavior

Yao & Yuan, 2020 ¹³	BV2 cells. Medium conditions: DMEM, 10% FBS. Primary microglia. Cortexes from ICR mice. 10% FBS	Serum deprivation produces changes in cell morphology and the expression levels of p-p38 and p-ERK in primary microglia and BV-2 cells.
Goshi et al., 2020 ¹⁴	Primary cortical cells taken from Sprague Dawley rats P0 to get a “tri-culture” (neurons, astrocytes, microglia) 10% Heat inactive horse serum used as base media	To eliminate serum, the authors add to the co-culture medium IL-34, TGF-B and cholesterol. The result of this tri-culture free serum was a model that mimics neuroinflammatory responses more realistically
Montilla et al., 2020 ¹⁵	Primary mixed glial cultures were prepared from the cerebral cortex of neonatal rats (P0-P2). FBS 10%	Microglia isolation from their environment and the presence of serum could alter their function and lead to a rapid loss of their signature gene expression
Tewari et al., 2021 ¹⁶	Human brain tissue was processed from de-identified neurosurgical patient	Removal of serum and addition of 3 components (TGF- β , IL-34, and cholesterol). Although limited proliferation of microglia was shown, they observed microglia ramified morphology and active in its surveillance and phagocytic capacity.
Dorion et al., 2022 ¹⁷	Human brain tissues were obtained from non-malignant cases of temporal lobe epilepsy at sites distant from suspected primary epileptic foci	Removal of FBS and its replacement with TIC4: TGF- β 1, IL-34, CSF1, CD200, CX3CL1, and cholesterol. Human microglia survive in serum-free condition, however FBS enhances survival and proliferation

Based on these reported findings, we have decided to include the following text in our manuscript:

“Cultivation and differentiation of primary murine microglia cultures provided crucial cues on the importance of experimental conditions, such as medium composition, or serum presence. Serum deprivation in primary microglia led to a significant cell death, nuclei condensation^{7,8} and cell death markers, such as p-p38 and p-ERK¹³. Therefore, several (growth) factors tested to replace the use of serum, such as M-CSF⁹, M-CSF, GMCSF, NGF and CCL-2¹⁰ could lead to viable microglia with primary and secondary processes. More recently, Bohlen et al., 2017 proposed the addition of three important factors: CSF-1/IL-34, TGF- β 2, and cholesterol to prevent microglial cell death¹¹. These cytokines have been used for murine primary microglia cultures^{12,14} and human primary microglia obtained from brain patients¹⁶. Extra addition of CD200, CX3CL1 was also proposed to support survival and human microglial response¹⁷. However, it has been shown that the use of serum further enhances survival and proliferation¹⁵.”

While uncertainties persist regarding the optimal culture models for recapitulating the properties and functions of microglia in the intact central nervous system (CNS), we have chosen the most used protocol for generating primary microglia, which involves the inclusion of 10% Fetal Bovine Serum (FBS). This approach allows us to establish a basis for comparison with a majority of prior studies. Although, Dorion et al. 2022 have suggested that the media formulation exerts only a minor influence on culture-induced alterations in the human microglial transcriptome; we recognize that the addition of serum or other reagents within the medium could potentially give rise to significant changes in the metabolic profile.”

Reviewer #3 (Remarks to the Author):

This revised version of the manuscript answered my previous concerns. The limitations of the paper have been significantly reduced or acknowledged and the manuscript is now very much improved.

Response: We appreciate the kind comments on our work.

Reviewer #4 (Remarks to the Author):

My review focuses on data in this manuscript derived from proteomics. I agree with other reviewers that investigating metabolic reprogramming requires investigation at the protein level, which authors have now included, both with discovery, label free proteomics and targeted proteomics. However, the following points need to be addressed:

1. I do not see anything in the methods or data availability section that states where the raw proteomics data is available. It is essential that the raw mass spectrometry/ proteomics data is shared to a public repository such as PRIDE/ Proteome Xchange. Proteomics data is also not present in the supplementary files included.

Response:

We have deposited our raw data in public repositories prior to submission of our revision.

This is for the reviewer and editor only:

The untargeted proteomics data have been deposited to the ProteomeXchange Consortium via the PRIDE partner repository. The dataset identifier is PXD043836 (the data can be accessed for review using the following information:

Username: reviewer_pxd043836@ebi.ac.uk; Password: sWhUGGRf).

The targeted mass spectrometry proteomics data have been deposited to the PASSEL server with the data identifier PASS05835 (the data can be accessed for review using the following information:

Username: PASS05835; Password: JD8364y).

Additionally, all data required to generate the plots of proteomic analyses are provided within our Source File as a supplement to our manuscript.

We have made sure that the following statement is incorporated in our data availability subsection:

Data availability statement

The datasets generated through (Abud et al., 2017) and re-analyzed for this study are available through GEO Series accession number GSE89189. The datasets generated through (Brownjohn et al., 2018) and re-analyzed for this study are available through GEO Series accession number GSE110952. We have generated RNAseq data on human and mouse microglia (iPSC-derived microglia and primary microglia) that are available using the accession number GSE221013. **The untargeted proteomics data have been deposited to the ProteomeXchange Consortium via the PRIDE partner repository. The dataset identifier is PXD043836. The targeted mass spectrometry proteomics data have been deposited to the PASSEL server with the data identifier PASS05835.**

2. Figure 5. Figure legend poorly describes the data, I find it very hard to understand each graph and what it is meant to be showing. Sometimes Figure letters are enclosed within brackets, sometimes not. Full stops are missing. Please be consistent.

Response: We thank this reviewer for raising this issue. We have revisited the legend for Figure 5. Letters are enclosed within brackets now only to refer to a previously mentioned panel.

Figure 5. Transcriptomic and proteomic characterization of key glycolytic enzymes in mouse microglia and human microglia-like cells.

Mouse and human microglial cells were in vitro stimulated with LPS and collected 4 hours after stimulation for proteomic analysis by LC-MS/MS.

A,B. Normalized counts depicting transcript abundance of *Hk1*, *Hk2*, and *Hk3* (A) and *HK1*, *HK2*, *HK3* (B). C,D. Label-free quantitation values from proteomic analysis depicting relative abundance of HK1, HK2, and HK2 in mouse (C) and human (D) cells. E,F. Proteomic abundances of hexokinases measured by LC-MS/MS with internal standards. (E) Depicts plots of relative abundances on mouse microglia and (F) depicts human cells. G,H. Normalized counts depicting transcript abundance of *Pfkl*, *Pfkp*, and *Pfkm* (G) and *PFKL*, *PFKP*, and *PFKM* (H). I,J. Label-free quantitation values depicting relative abundance of PFKL, PFKP, and PFKM in mouse (I) and human (J) cells. K,L. Proteomic abundances of phosphofructokinases measured by LC-MS/MS with internal standards. (K) Depicts plots of relative abundances on mouse microglia and (L) depicts human cells. M,N. Normalized counts depicting transcript abundance of *Pfkfb1*, *Pfkfb2*, *Pfkfb3*, and *Pfkfb4* (M) and *PFKFB2*, *PFKFB3* and *PFKFB4* (N). O,P. Proteomic abundances of PFKFBs measured by LC-MS/MS with internal standards. (O) Depicts relative abundances on mouse microglia and (P) depicts human cells. For all panels, error bars represent SEM. Every biological replicate is depicted as a dot. For mouse RNAseq n=3, for human RNAseq n=4, for mouse proteomics n=4 and for human proteomics n=5. n indicates biological independent experiments. For RNAseq data p values were determined by the Wald test and multi testing corrected in DESEQ2. For proteomic data, p values were determined by unpaired two-tailed t-test. * $p < 0.05$, ** $p < 0.01$, and *** $p < 0.001$

3. Figure 5. Please state in figure legend and on graphs that Figures E,F,K,L, O and P are derived from data from targeted proteomics rather than LFQ. You do this in Sup Fig 6 but not in main Fig 5. I am also confused by the statement of ‘absolute and relative abundancies’ to describe this data as it is displayed as fold change only. Please amend.

Response: We have stated on graphs and figure legend that Figures E,F,K,L, O and P are derived from targeted proteomics. We have also removed the incorrect statement

4. Figure 5F. Not sure why a darker shade of green is used instead of blue for LPS4hr.

Response: We have corrected the color coding for this figure.

5. Would be good to state how many proteins were identified by discovery proteomics in both mouse and human samples. This will give an overall idea of the quality and coverage gained from the methods.

Response: We have added this in our methods section.

Amount of identified proteins for mouse microglia: 2901

Amount of identified proteins for human microglia: 3127

References

1. Matsuda, M. *et al.* Species-specific segmentation clock periods are due to differential biochemical reaction speeds. *Science* **369**, 1450–1455 (2020).
2. Rayon, T. *et al.* Species-specific pace of development is associated with differences in protein stability. *Science* **369**, (2020).
3. Stewart-Ornstein, J., Cheng, H. W. (Jacky) & Lahav, G. Conservation and Divergence of p53 Oscillation Dynamics across Species. *Cell Systems* **5**, 410-417.e4 (2017).
4. Bulfield, G., Moore, E. A. & Kacser, H. GENETIC VARIATION IN ACTIVITY OF THE ENZYMES OF GLYCOLYSIS AND GLUCONEOGENESIS BETWEEN INBRED STRAINS OF MICE. *Genetics* **89**, 551–561 (1978).
5. Burlikowska, K. *et al.* Comparison of metabolomic profiles of organs in mice of different strains based on SPME-LC-HRMS. *Metabolites* **10**, 1–15 (2020).
6. Folmes, C. D. L. *et al.* Metabolome and metaboproteome remodeling in nuclear reprogramming. *Cell Cycle* **12**, 2355–2365 (2013).
7. Koyama, Y. *et al.* Serum-Deprivation Induces Cell Death of Rat Cultured Microglia Accompanied With Expression of Bax Protein. *The Japanese Journal of Pharmacology* **83**, 351–354 (2000).
8. Yenari, M. A. & Giffard, R. G. Ischemic vulnerability of primary murine microglial cultures. *Neuroscience Letters* **298**, 5–8 (2001).
9. Negishi, T., Ishii, Y., Kyuwa, S., Kuroda, Y. & Yoshikawa, Y. Primary culture of cortical neurons, type-1 astrocytes, and microglial cells from cynomolgus monkey (*Macaca fascicularis*) fetuses. *Journal of Neuroscience Methods* **131**, 133–140 (2003).
10. Etemad, S., Zamin, R. M., Ruitenber, M. J. & Filgueira, L. A novel in vitro human microglia model: Characterization of human monocyte-derived microglia. *Journal of Neuroscience Methods* **209**, 79–89 (2012).
11. Bohlen, C. J. *et al.* Diverse Requirements for Microglial Survival, Specification, and Function Revealed by Defined-Medium Cultures. *Neuron* (2017) doi:10.1016/j.neuron.2017.04.043.
12. Collins, H. Y. & Bohlen, C. J. Isolation and culture of rodent microglia to promote a dynamic ramified morphology in serum-free medium. *Journal of Visualized Experiments* **2018**, 1–9 (2018).
13. Yao, Y. & Fu, K. Y. Serum-deprivation leads to activation-like changes in primary microglia and BV-2 cells but not astrocytes. *Biomedical Reports* **13**, 1–8 (2020).
14. Goshi, N., Morgan, R. K., Lein, P. J. & Seker, E. A primary neural cell culture model to study neuron, astrocyte, and microglia interactions in neuroinflammation. *Journal of Neuroinflammation* **17**, 1–16 (2020).
15. Montilla, A., Zabala, A., Matute, C. & Domercq, M. Functional and Metabolic Characterization of Microglia Culture in a Defined Medium. *Frontiers in Cellular Neuroscience* **14**, 1–11 (2020).
16. Tewari, M. *et al.* Physiology of Cultured Human Microglia Maintained in a Defined Culture Medium. *ImmunoHorizons* **5**, 257–272 (2021).
17. Dorion, M. F. *et al.* Systematic comparison of culture media uncovers phenotypic shift of primary human microglia defined by reduced reliance to CSF1R signaling. *Glia* **71**, 1278–1293 (2023).